# Understanding Accelerated Gradient Methods: Lyapunov Analyses and Hamiltonian-Assisted Interpretations

**Penghui Fu**                                                          *penghui@cuhk.edu.cn*
*School of Data Science*
*The Chinese University of Hong Kong, Shenzhen*

**Zhiqiang Tan**                                                        *ztan@stat.rutgers.edu*
*Department of Statistics*
*Rutgers University*

**Reviewed on OpenReview:** *https://openreview.net/forum?id=0jvg4M1W4O*

## Abstract

We formulate two classes of first-order algorithms more general than previously studied for minimizing smooth and strongly convex or, respectively, smooth and convex functions. We establish sufficient conditions, via new discrete Lyapunov analyses, for achieving accelerated convergence rates which match Nesterov's methods in the strongly convex and convex (not necessarily strongly convex) settings. Our results identify, for the first time, a concrete sufficient condition on gradient correction for accelerated convergence. Next, we study the convergence of limiting ordinary differential equations (ODEs), including high-resolution ODEs, and point out currently notable gaps between the convergence properties of the corresponding algorithms and ODEs, especially regarding the role of gradient correction. Finally, we propose a new class of discrete algorithms, called the Hamiltonian-assisted gradient method, directly based on a Hamiltonian function and several interpretable operations, and then provide specific interpretations of our acceleration conditions in terms of the momentum variable updates.[1]

## 1 Introduction

Optimization plays a vital role in machine learning, statistics, and many other fields. In the optimization literature, there exists a striking phenomenon where after suitable modifications of a first-order method,[2] the convergence guarantee can be improved, often attaining the complexity lower bound, with a similar computational cost as before. Such acceleration has been widely studied since the seminal work of Nesterov (1983), which improves gradient descent for minimizing smooth convex functions. Examples include constrained optimization (Nesterov, 2018), mirror descent with a non-Euclidean norm (Krichene et al., 2015), composite optimization with a proximable function (Beck & Teboulle, 2009), primal-dual splitting (Chambolle & Pock, 2011), stochastic gradient methods (Zhang & Lin, 2017; Allen-Zhu, 2018), and others.

Despite extensive research, the scope and mechanism of acceleration remain to be fully understood, even including the original acceleration of gradient descent for smooth convex optimization. Typically, a class of algorithms is constructed, and their convergence proofs are provided using suitable techniques. Among them, Lyapunov analysis has been a prevalent approach for studying convergence properties of discrete algorithms, once those algorithms are defined. In fact, Nesterov (1983) used a potential function or a

---

[1]Code for numerical results is available at: `https://github.com/ffpphh/acc_grad`.
[2]First-order methods refer to methods using function values and gradients only, whereas second-order methods additionally rely on the Hessian matrices or their approximations.

Lyapunov function to establish the convergence of an accelerated algorithm, although Nesterov's later work turned to the technique of estimate sequences for studying gradient methods (Nesterov, 1988; 2018). See Bansal & Gupta (2019) and d'Aspremont et al. (2021) for overviews of Lyapunov-based proofs for gradient methods. The central step in Lyapunov analysis is to construct an appropriate Lyapunov function that is used to deduce the desired convergence. Recently, a promising framework for constructing Lyapunov functions systematically is via linear matrix inequalities (LMI) and semidefinite programming (SDP) techniques. The core idea is to transform the existence of a quadratic Lyapunov function into the feasibility of an SDP, which involves solving for a positive definite matrix subject to linear matrix inequalities. Currently, there are two distinct methodologies under this framework, namely the performance estimation problem (PEP) and integral quadratic constraints (IQC). The PEP method is motivated by finding an objective function for which the given optimization algorithm has a worse performance, and has been adapted for Lyapunov analysis (Taylor et al., 2018; Taylor & Bach, 2019; Upadhyaya et al., 2025). On the other hand, the IQC method is motivated from a control-theoretic perspective of optimization algorithms, as originally proposed in Lessard et al. (2016) and later extended by Fazlyab et al. (2018). More recently, the IQC method has been leveraged to analyze a family of Nesterov optimization methods by Sanz Serna & Zygalakis (2021), whose results are further improved by Dobson et al. (2025).

Although being a major technical tool for establishing convergence rates, Lyapunov analysis itself may not explain conceptually when and how acceleration can be achieved. Various efforts have been made for the latter purpose. For example, geometric formulations are proposed by coupling gradient descent and mirror descent (Allen-Zhu & Orecchia, 2017), and by averaging two minimizers of an upper and a lower quadratic bound for the objective function (Bubeck et al., 2015; Drusvyatskiy et al., 2018). Another useful approach is to relate discrete algorithms to their continuous limits as ordinary differential equations (ODEs), obtained by letting the stepsize in the discrete algorithms tend to zero. ODEs are usually more tractable to study than their discrete counterparts by exploiting a rich set of analytical tools from continuous-time dynamical systems and control theory. The analyses and properties of the limiting ODEs can provide insights about those of the original discrete algorithms, for example, to facilitate the construction of Lyapunov functions for the discrete algorithms (Qian, 1999; Su et al., 2016; Yang et al., 2018; Sun et al., 2020; Shi et al., 2022). Recently, stochastic differential equations (SDEs) are leveraged to understand the interplay between the learning rate, gradient noise, and gradient compression in distributed stochastic gradient methods (Compagnoni et al., 2025a;b). Conversely, ODEs can be directly formulated, and then their numerical discretizations are studied (Wibisono et al., 2016; Wilson et al., 2021; Sanz Serna & Zygalakis, 2021). It has been argued that acceleration can be attributed to suitable discretizations of ODEs, such as the Runge–Kutta integrator and its variants (Zhang et al., 2018; Dobson et al., 2025), and the symplectic integrators (Shi et al., 2019; França et al., 2020; Muehlebach & Jordan, 2021).

However, the approach based solely on ODEs is insufficient to account for whether acceleration is achieved. For example, for minimizing strongly convex functions, both Nesterov's accelerated gradient method (NAG-SC) and Polyak's heavy-ball method (HB) admit the same limiting ODE, but only NAG-SC achieves accelerated convergence while HB *provably* does not (Goujaud et al., 2025). To address this limitation, Shi et al. (2022) proposed a new approach based on high-resolution ODEs. These ODEs are derived by retaining certain terms from the discrete algorithms that would otherwise vanish in the continuous-time limit as the stepsize tends to zero. In particular, the high-resolution ODE for NAG-SC includes an additional *Hessian* term, which is absent in the ODE for HB. The Hessian term stems from the *gradient correction* term in the NAG-SC algorithm, which is defined as linear in the difference between the current and previous gradients. By translating the Lyapunov analysis of high-resolution ODEs back to their discrete counterparts, Shi et al. (2022) established accelerated convergence for NAG-SC and non-accelerated convergence for HB. Their analysis suggests that gradient correction plays a crucial role in achieving acceleration. This raises two interesting questions. First, can we quantify the magnitude of gradient correction for achieving acceleration by discrete algorithms? Second, can high-resolution ODEs themselves provide insight into such a quantification?

We take a direct approach to studying the scope and mechanism of gradient correction for acceleration of discrete algorithms. Our work has three main contributions, summarized as follows. See Appendix A for a detailed comparison of our work and the literature, including LMI/SDP techniques, Sanz Serna & Zygalakis (2021), and Shi et al. (2022).

- We formulate two general classes of algorithms and establish sufficient conditions for when the algorithms achieve accelerated as well as non-accelerated convergence for the strongly convex and convex settings, respectively (see Sections 2 and 3). Our results, *for the first time*, identify a concrete sufficient condition on the magnitude of the gradient correction term to achieve acceleration, thereby answering the first question raised above. Our proofs are based on hand-crafted Lyapunov analysis relying on neither connections with ODEs nor the LMI/SDP framework. See Section 6 for a comparison of our and existing Lyapunov analyses.

- We study high-resolution ODEs derived from our classes of algorithms and compare convergence rates of these ODEs with those of their discrete counterparts. We find that high-resolution ODEs from our classes converge at the same rate, even though the corresponding discrete algorithms exhibit different acceleration behaviors. This provides new evidence that even high-resolution ODEs may be insufficient for explaining acceleration, thereby suggesting a negative answer to the second question above. See Section 4 for a more detailed summary.

- To provide further understanding of the sufficient conditions on gradient correction, we propose a new class of discrete algorithms called the Hamiltonian-assisted gradient method (HAG), which leads to specific interpretations for these conditions in terms of momentum variable updates in both the strongly convex and convex settings. See Section 5.2 for a more detailed summary.

Before discussing our main technical findings, we introduce necessary notation that is largely adopted from Nesterov (2018). For a smooth and convex function $f : \mathbb{R}^n \to \mathbb{R}$, consider the minimization problem

$$\min_{x \in \mathbb{R}^n} f(x). \tag{1}$$

Throughout this paper, we assume that the minimum in (1) is finite and attained, i.e., $\arg\min_{x \in \mathbb{R}^n} f(x) \neq \emptyset$. Denote by $x^\star \in \arg\min_{x \in \mathbb{R}^n} f(x)$ one of the minimizers, and $f^\star = f(x^\star)$. When $f$ is strongly convex, $x^\star$ uniquely exists, so that $f^\star$ is always finite and attained. For $m \geq 1$, let $\mathcal{F}^m$ be the set of functions that are convex and $m$ times continuously differentiable on $\mathbb{R}^n$. Moreover, we define $\mathcal{F}_L^m \subset \mathcal{F}^m$ which further requires that $f$ is $L$-smooth, that is, the gradient $\nabla f$ is $L$-Lipschitz continuous, $\|\nabla f(y) - \nabla f(x)\| \leq L\|y - x\|$ for $x, y \in \mathbb{R}^n$, where $L > 0$ is the Lipschitz constant and $\|\cdot\|$ denotes the Euclidean norm. For $\mu > 0$ we define $\mathcal{S}_\mu^m \subset \mathcal{F}^m$ which further requires that $f$ is $\mu$-strongly convex, that is, $f(y) \geq f(x) + \langle \nabla f(x), y - x \rangle + \frac{\mu}{2}\|y - x\|^2$ for $x, y \in \mathbb{R}^n$. Let $\mathcal{S}_{\mu,L}^m = \mathcal{F}_L^m \cap \mathcal{S}_\mu^m$. For additional properties of smoothness and strong convexity, readers are referred to, for example, Appendix A in d'Aspremont et al. (2021). Throughout this paper, we use $s$ to denote a fixed step size in discrete algorithms.

Our main technical findings are highlighted by the following two propositions, stated in a self-contained manner. Further interpretations of the technical results are provided in Section 5.2. In the strongly convex setting, Proposition 1 can be deduced from Corollary 1 to give sufficient conditions for a simplified class of algorithms to achieve an accelerated convergence bound matching NAG-SC.

**Proposition 1.** *Suppose that $f : \mathbb{R}^n \to \mathbb{R}$ is a function in $\mathcal{S}_{\mu,L}^1$ for $0 < \mu \leq L$. Let $\{x_k\}$ be the iterates from the following algorithm:*

$$x_{k+1} = x_k - s\nabla f(x_k) + (1 - c_1\sqrt{\mu s})(x_k - x_{k-1}) - \tilde{c}_2 \cdot s(\nabla f(x_k) - \nabla f(x_{k-1})), \quad \text{for } k \geq 1,$$

*where $x_1 = x_0 - s\nabla f(x_0)$; $c_1$ and $\tilde{c}_2$ are scalar parameters. Then there exist constants $C_0, C_1 > 0$, depending only on $c_1$ and $\tilde{c}_2$, such that for $0 < s \leq C_0/L$, the iterates $\{x_k\}$ satisfy the bound*

$$f(x_k) - f^\star = O\left(L\|x_0 - x^\star\|^2(1 - C_1\sqrt{\mu s})^k\right),$$

*provided that $c_1 > 2$ and $\tilde{c}_2 \geq \frac{1}{2}$.*

The conditions in Proposition 1 are in terms of $c_1$ and $\tilde{c}_2$, which controls the momentum term $x_k - x_{k-1}$ and the gradient correction term $s(\nabla f(x_k) - \nabla f(x_{k-1}))$, respectively. To our knowledge, this is the first time that explicit conditions have been established for the acceleration of a general class of algorithms in

the strongly convex setting. In addition, Corollary 1 indicates that when $0 \leq \tilde{c}_2 < \frac{1}{2}$, an algorithm in the above form is proved to only have a convergence rate matching the vanilla gradient descent. Previously, such comparative results were only known for two specific algorithms, NAG-SC and HB (Shi et al., 2022). For a numerical illustration, Figure 1 (a) presents the optimality gap by applying the iterative algorithm in Proposition 1 to an ill-conditioned quadratic function, with $s = 0.1$, $x_0 = c(1, 1)$, $c_1 = 3 > 2$, and varying $\tilde{c}_2$. It can be seen that using $\tilde{c}_2 \geq 1/2$ significantly improves convergence.

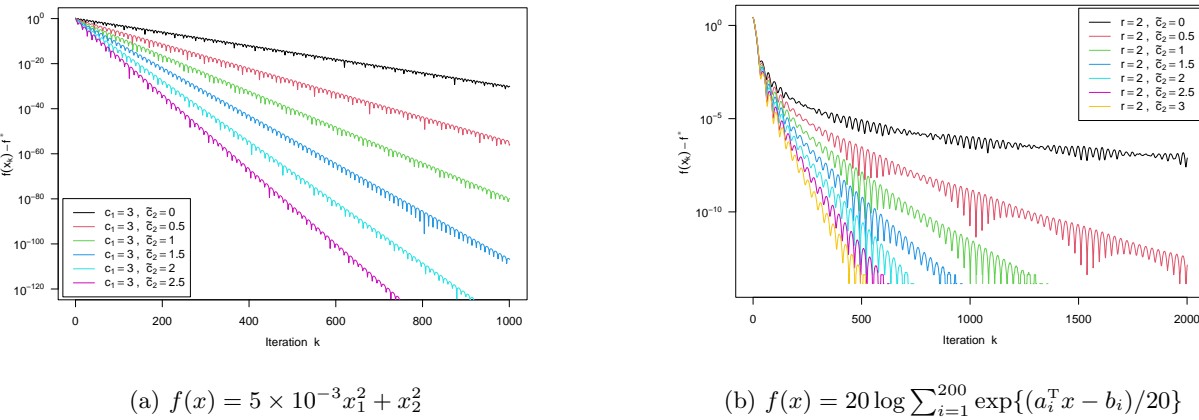

(a) $f(x) = 5 \times 10^{-3} x_1^2 + x_2^2$  (b) $f(x) = 20 \log \sum_{i=1}^{200} \exp\{(a_i^{\mathsf{T}} x - b_i)/20\}$

Figure 1: Optimality gaps in minimizing strongly convex and convex objectives with varying coefficients of gradient correction. Parameters $a_i \in \mathbb{R}^{200}$ and $b_i \in \mathbb{R}$ in the log-sum-exp function are i.i.d. draws from the standard Normal distribution. The minimum $f^\star$ of the log-sum-exp function is approximated by the minimum value that the algorithm achieves.

In the convex setting, Proposition 2 can be deduced from Theorem 4, to give sufficient conditions for a simplified class of algorithms to achieve an accelerated convergence bound matching NAG-C (the analogue of NAG-SC in the convex setting).

**Proposition 2.** *Suppose that $f \colon \mathbb{R}^n \to \mathbb{R}$ is a function in $\mathcal{F}_L^1$ with $L > 0$. Let $\{x_k\}$ be the iterates from the following algorithm*

$$x_{k+1} = x_k - s\nabla f(x_k) + \sigma_{k+1}(x_k - x_{k-1}) - \sigma_{k+1}\tilde{c}_2 \cdot s(\nabla f(x_k) - \nabla f(x_{k-1})), \quad \text{for } k \geq 1,$$

*where $x_1 = x_0 - s\nabla f(x_0)$; $\tilde{c}_2$ is a scalar parameter; $\sigma_{k+1} = (\alpha_k - 1)/\alpha_{k+1}$, and $\{\alpha_k > 0\}$ with $\alpha_0 = 1$ is a scalar sequence such that $\alpha_k = \Omega(k)$, $\alpha_{k+1}(\alpha_{k+1} - 1) \leq \alpha_k^2$, $\{\alpha_{k+1}/\alpha_k\}$ is monotone (either non-increasing or non-decreasing) in $k$ and $\lim_k \alpha_{k+1}/\alpha_k = 1$. Then for $0 < s \leq C_0/L$ with $C_0 = \frac{1}{4} \wedge \frac{2\tilde{c}_2 - 1}{2\tilde{c}_2^2}$, the iterates $\{x_k\}$ satisfy the bound*

$$f(x_k) - f^\star = O\left(\frac{\|x_0 - x^\star\|^2}{sk^2}\right),$$

*provided that $\tilde{c}_2 > \frac{1}{2}$.*

Proposition 2 is more general than the related result in Shi et al. (2022) corresponding to specific choices of $\alpha_k = \frac{k+r}{r}$ and $\sigma_{k+1} = \frac{k}{k+r+1}$ with $r \geq 2$. For a numerical illustration, Figure 1 (b) presents the optimality gap by applying the iterative algorithm in Proposition 2 to the log-sum-exp function, which is convex but not strongly convex, with $s = 1$, $x_0 = 0$, $r = 2$, and varying $\tilde{c}_2$. Similar to the strongly convex case, it is observed that using $\tilde{c}_2 > 1/2$ significantly improves convergence.

By comparing Propositions 1 and 2, we observe a common condition requiring $\tilde{c}_2$, the coefficient for gradient correction $s(\nabla f(x_k) - \nabla f(x_{k-1}))$, to (asymptotically) surpass an explicit threshold of $\frac{1}{2}$. Such a unified condition on gradient correction is identified for the first time for acceleration. It remains an open question to study whether this condition is also necessary for acceleration. After presenting the main results in Sections 2 and 3, we present further numerical results to illustrate different performances of the algorithms whose

parameters either satisfy or violate the sufficient conditions for acceleration. It is found that algorithms with parameters satisfying (or lying on the boundary of) these conditions exhibit better performance. Overall, the numerical results align well with our theoretical findings. All proofs are deferred to Appendix C–F.

**Asymptotic Notation.** Given two sequences $\{a_k\}$ and $\{b_k\}$, we write $a_k = O(b_k)$ if there exist constants $C > 0$ and $K \geq 1$, such that $a_k \leq Cb_k$ for $k \geq K$. Similarly, we write $a_k = \Omega(b_k)$ if $a_k \geq Cb_k$ for $k \geq K$. We write $a_k = \Theta(b_k)$ if both $a_k = \Omega(b_k)$ and $a_k = O(b_k)$, and write $a_k \sim b_k$ if $\lim_k a_k/b_k = 1$. More generally, for a set of pairs $(g, h)$, we write $g = O(h)$ or $g \lesssim h$ if there exists a constant $C > 0$ such that $g \leq Ch$ for $(g, h)$. Similarly, we write $g = \Omega(h)$ or $g \gtrsim h$ if $g \geq Ch$ for $(g, h)$. We write $g \asymp h$ if both $g \lesssim h$ and $g \gtrsim h$.

For convenience, a summary of the notation used is provided in Table 3 (Appendix B).

## 2 Acceleration for strongly convex functions

In this section, we first give background on several algorithms for minimizing smooth and strongly convex functions. Then we study a general class of algorithms.

### 2.1 Review of accelerated gradient methods

A basic method for solving (1) is the gradient descent (GD):

$$x_{k+1} = x_k - s\nabla f(x_k), \quad \text{for } k \geq 0, \tag{2}$$

with an initial point $x_0$ and a stepsize $s$. In this section, we study the case where $f \in \mathcal{S}_{\mu,L}^1$ is smooth and strongly convex. See Section 3 for the case where $f$ is smooth and convex.

For $f \in \mathcal{S}_{\mu,L}^1$, GD can be exponentially convergent in $k$, and the convergence rate depends on the condition number $\kappa = L/\mu$. In fact, the GD iterates (2) satisfy that for $0 < s \leq 2/(\mu + L)$ (Nesterov, 2018),

$$f(x_k) - f^\star \leq \frac{L\|x_0 - x^\star\|^2}{2}\left(1 - \frac{2\mu s}{1 + 1/\kappa}\right)^k.$$

When $s = 1/L$, the rate is $(\frac{1-1/\kappa}{1+1/\kappa})^k \sim (1 - \frac{1}{\kappa})^{2k}$ for $\kappa \to \infty$. When $s = 2/(L + \mu)$, the rate improves to $(\frac{1-1/\kappa}{1+1/\kappa})^{2k} \sim (1 - \frac{1}{\kappa})^{4k}$. For both choices of the stepsize, the iteration complexity for $f(x_k) - f^\star \leq \epsilon$ is $O(\kappa \log(\frac{1}{\epsilon}))$, which has a linear dependency on $\kappa$.

Nesterov (1988) proposed an accelerated gradient method (NAG-SC) of the following form with an additional extrapolation step: for $k \geq 0$,

$$y_{k+1} = x_k - s\nabla f(x_k), \tag{3a}$$
$$x_{k+1} = y_{k+1} + \sigma(y_{k+1} - y_k), \tag{3b}$$

with $x_0 = y_0$ and the momentum coefficient $\sigma = \frac{1-\sqrt{q}}{1+\sqrt{q}}$, where $q = \mu s$, a shorthand to be used throughout this paper. Equivalently, (3) can be expressed in a single-variable form:

$$x_{k+1} = \underbrace{x_k - s\nabla f(x_k)}_{\text{gradient descent}} + \underbrace{\sigma(x_k - x_{k-1})}_{\text{momentum}} - \underbrace{\sigma s(\nabla f(x_k) - \nabla f(x_{k-1}))}_{\text{gradient correction}}, \quad \text{for } k \geq 1, \tag{4}$$

with $x_0$ and $x_1 = x_0 - \frac{2s\nabla f(x_0)}{1+\sqrt{q}}$. Compared with GD, the iterate (4) involves two additional terms, called the momentum and gradient correction. For $0 < s \leq 1/L$, the NAG-SC iterates satisfy $f(x_k) - f^\star = O((1 - \sqrt{q})^k)$. When $s = 1/L$, the bound reduces to $O((1 - 1/\sqrt{\kappa})^k)$, and the iteration complexity is lowered to $O(\sqrt{\kappa} \log(\frac{1}{\epsilon}))$, with a square-root dependency on $\kappa$. Drori & Taylor (2022) established that a lower bound for minimizing general $f \in \mathcal{S}_{\mu,L}^1$ is $f(x_k) - f^\star = \Omega((1 - 1/\sqrt{\kappa})^{2k})$, where $\{x_k\}$ are iterates of any black-box

first-order method.[3] Therefore, NAG-SC is optimal up to a constant factor of 2 in terms of the iteration complexity.

There are several first-order methods *exactly* reaching the lower bound $(1 - 1/\sqrt{\kappa})^{2k}$, for instance, the information-theoretic exact method (ITEM) (Taylor & Drori, 2023) and triple-momentum method (TMM) (Van Scoy et al., 2017). While ITEM involves time-dependent coefficients, TMM is defined with time-independent coefficients (d'Aspremont et al., 2021): for $k \geq 0$,

$$y_{k+1} = x_k - s\nabla f(x_k), \tag{5a}$$

$$z_{k+1} = \sqrt{q}\left(x_k - \frac{1}{\mu}\nabla f(x_k)\right) + (1 - \sqrt{q})z_k, \tag{5b}$$

$$x_{k+1} = \frac{2\sqrt{q}}{1+\sqrt{q}}z_{k+1} + \left(1 - \frac{2\sqrt{q}}{1+\sqrt{q}}\right)y_{k+1}, \tag{5c}$$

with $x_0 = z_0$. The sequence $\{z_k\}$ in TMM (5) is auxiliary. See Appendix C.1 for an equivalent form of (5) containing $\{y_{k+1}\}$ and $\{x_{k+1}\}$ only, where $z_{k+1}$ can be recovered by

$$z_{k+1} = \frac{1+\sqrt{q}}{2\sqrt{q}}x_{k+1} + \left(1 - \frac{1+\sqrt{q}}{2\sqrt{q}}\right)y_{k+1}, \tag{6}$$

Nevertheless, $\{z_k\}$ plays a vital role in the existing analysis of TMM. Van Scoy et al. (2017) showed that $\{z_k\}$ achieves the lower bound, i.e., $f(z_k) - f^\star = O((1 - 1/\sqrt{\kappa})^{2k})$ when $s = 1/L$. To our knowledge, it remains an open question whether $\{x_k\}$ also achieves the lower bound.

In the Lyapunov analysis of NAG-SC in Bansal & Gupta (2019), a similar auxiliary sequence $\{z_k\}$ as above is introduced by defining

$$z_{k+1} = \frac{1+\sqrt{q}}{\sqrt{q}}x_{k+1} + \left(1 - \frac{1+\sqrt{q}}{\sqrt{q}}\right)y_{k+1}, \tag{7}$$

and the NAG-SC iterates in (3) together with (7) are equivalently reformulated as: for $k \geq 0$,

$$y_{k+1} = x_k - s\nabla f(x_k), \tag{8a}$$

$$z_{k+1} = \sqrt{q}\left(x_k - \frac{1}{\mu}\nabla f(x_k)\right) + (1 - \sqrt{q})z_k, \tag{8b}$$

$$x_{k+1} = \frac{\sqrt{q}}{1+\sqrt{q}}z_{k+1} + \left(1 - \frac{\sqrt{q}}{1+\sqrt{q}}\right)y_{k+1}, \tag{8c}$$

with $x_0 = z_0$. See Appendix C.1 for a derivation, provided for completeness. By comparing TMM (5) with NAG-SC (8), they differ only in how $x_{k+1}$ is defined in (5c) and (8c). Bansal & Gupta (2019) constructed a new Lyapunov function to simplify the convergence proof for NAG-SC, and showed that $\{y_k\}$ achieves the bound $f(y_k) - f^\star = O((1 - 1/\sqrt{\kappa})^k)$ when $s = 1/L$. By some additional arguments, similar convergence bounds can also be deduced for $\{z_k\}$ and $\{x_k\}$.

For comparison, we also mention the heavy-ball (HB) method (Polyak, 1964), defined as

$$x_{k+1} = x_k - s\nabla f(x_k) + \sigma(x_k - x_{k-1}), \quad \text{for } k \geq 1, \tag{9}$$

with $\sigma = \frac{1-\sqrt{q}}{1+\sqrt{q}}$ as in (3). Note that (9) differs from NAG-SC (4) only in the absence of gradient correction. Following Shi et al. (2022), algorithm (9) is a slight modification of the original method in Polyak (1964) where the momentum coefficient $\sigma = (1 - \sqrt{q})^2$. If $s$ is small, the two coefficients $(1 - \sqrt{q})^2$ and $\frac{1-\sqrt{q}}{1+\sqrt{q}}$ are close. The original heavy-ball method, with the specific $s = 4/(\sqrt{L} + \sqrt{\mu})^2$, achieves an accelerated convergence rate $(\frac{1-1/\sqrt{\kappa}}{1+1/\sqrt{\kappa}})^{2k}$ for quadratic $f$ (Polyak, 1987). However, when minimizing functions in $\mathcal{S}_{\mu,L}^1$ that are not necessarily quadratic, it is shown in Goujaud et al. (2025) that for any $s$ and $\sigma$, the worst-case

---

[3]Black-box means that no prior knowledge of $f$ (e.g., $f$ is quadratic) is available except for the class $f$ belongs to. For $\mathcal{F}_L^1$, the available information is only the Lipschitz constant $L$.

Table 1: Existing algorithms covered by our analysis

| Method | Formulation | Convergence |
|---|---|---|
| NAG-SC | (10) or (11) with $\eta = \nu = \tau = 1$ | Accelerated rate by Theorem 1 (ii-d) |
| | (14) with $c_0 = 1$, $c_1 = 2$, and $c_2 = 3/2$ | Accelerated rate by Corollary 1 (ii) as a limiting case when $c_1 \downarrow 2$ |
| TMM | (10) or (11) with $\eta = \nu = 1$, $\tau = 2$ | Accelerated rate by Theorem 1 (ii-a) |
| | (14) with $c_0 = 2$, $c_1 = 3$, and $c_2 = \sqrt{2}$ | Accelerated rate by Corollary 1 (ii) |
| HB | (14) with $c_0 = 1$, $c_1 = 2$, and $c_2 = 1/2$ | Non-accelerated rate by Corollary 1 (i) as a limiting case when $c_1 \downarrow 2$ |

convergence rate of HB on $\mathcal{S}_{\mu,L}^1$ is no better than $O((1 - O(1/\kappa))^k)$. Therefore, HB provably does not reach an accelerated convergence rate on smooth and strongly convex problems. Goujaud et al. (2025) further shows that adding more regularity conditions to $f$ (e.g., restricting $f$ within those with Lipschitz continuous Hessians) does not result in acceleration either. The failure of HB to achieve acceleration indicates that the gradient correction plays a vital role in achieving acceleration.

## 2.2 Main results

We formulate a broad class of algorithms, including NAG-SC and TMM as special cases, and establish sufficient conditions for when the algorithms in the class achieve acceleration, which is defined as reaching an objective gap of $O((1 - C/\sqrt{\kappa})^k)$ at iteration $k$, for a constant $C > 0$. Our work does not aim to find a sharp value of $C$ or address the question of whether these algorithms *exactly* achieve the complexity lower bound corresponding to $C = 2$ (Drori & Taylor, 2022). For an overview of prior work on acceleration covered by our analysis of the general class of algorithms, see Table 1.

To unify and extend NAG-SC and TMM, we consider the following class of algorithms: for $k \geq 0$,

$$y_{k+1} = x_k - \eta s \nabla f(x_k), \tag{10a}$$

$$z_{k+1} = \nu \sqrt{q} \left( x_k - \frac{1}{\mu} \nabla f(x_k) \right) + (1 - \nu \sqrt{q}) z_k, \tag{10b}$$

$$x_{k+1} = \frac{\tau \sqrt{q}}{1 + \sqrt{q}} z_{k+1} + \left( 1 - \frac{\tau \sqrt{q}}{1 + \sqrt{q}} \right) y_{k+1}, \tag{10c}$$

with $x_0 = z_0$, where $\eta, \nu, \tau \geq 0$ are three parameters which may depend on $q$. NAG-SC (3) or (8) is recovered by setting $(\eta, \nu, \tau) = (1, 1, 1)$, and TMM (5) is recovered by $(\eta, \nu, \tau) = (1, 1, 2)$. In this way, the two algorithms differ only in the choice of $\tau$.

Equivalently, algorithm (10) can be put into a single-variable form in terms of $\{x_k\}$, similarly to (4) for NAG-SC: for $k \geq 1$,

$$x_{k+1} = x_k - \frac{\nu(\tau + \zeta\eta\sqrt{q})}{1 + \sqrt{q}} s \nabla f(x_k) + \frac{\zeta(1 - \nu\sqrt{q})}{1 + \sqrt{q}} (x_k - x_{k-1}) - \frac{\zeta\eta(1 - \nu\sqrt{q})}{1 + \sqrt{q}} s(\nabla f(x_k) - \nabla f(x_{k-1})), \tag{11}$$

with $x_0$ and $x_1 = x_0 - \frac{\zeta\eta + \nu\tau}{1 + \sqrt{q}} s \nabla f(x_0)$, where $\zeta = 1 + (1 - \tau)\sqrt{q}$, a shorthand used throughout this paper. The coefficients in the three terms for $s\nabla f(x_k)$, $x_k - x_{k-1}$, and $s(\nabla f(x_k) - \nabla f(x_{k-1}))$ are highly structured due to the translation from (10).

The following result gives sufficient conditions for the convergence of algorithm (10) in the scenario where $(\eta, \nu, \tau)$ are constants free of $q (= \mu s)$.

**Theorem 1.** *Let $f \colon \mathbb{R}^n \to \mathbb{R}$ be a function in $\mathcal{S}_{\mu,L}^1$ with $0 < \mu \leq L$. Assume that $\eta = \eta_0$, $\nu = \nu_0$, and $\tau = \tau_0$ for some constants $(\eta_0, \nu_0, \tau_0)$, free of $q (= \mu s)$.*

*(i) There exist constants $C_0, C_1 > 0$, depending only on $(\eta_0, \nu_0, \tau_0)$, such that for $0 < s \le C_0 \frac{\mu}{L^2}$, the iterates of (10) satisfy*

$$f(x_k) - f^\star = O\left(L\|x_0 - x^\star\|^2(1 - C_1\sqrt{\mu s})^k\right), \tag{12}$$

*provided that one of the following conditions holds:*

    *(i-a) $\nu_0, \tau_0 > 0$, $\nu_0 \ne \tau_0$, and $0 \le \eta_0 < \nu_0\tau_0/2$;*

    *(i-b) $\nu_0 = \tau_0 > 2$ and $\eta_0 = \tau_0^2/2$.*

*(ii) There exist constants $C_0, C_1 > 0$, depending only on $(\eta_0, \nu_0, \tau_0)$, such that for $0 < s \le \frac{C_0}{L}$, the iterates of (10) satisfy (12), provided that one of the following conditions holds:*

    *(ii-a) $\nu_0, \tau_0 > 0$, $\nu_0 \ne \tau_0$, and $\eta_0 \ge \nu_0\tau_0/2$;*

    *(ii-b) $\nu_0 = \tau_0 \ge 2$ and $\eta_0 > \tau_0^2/2$;*

    *(ii-c) $1 < \nu_0 = \tau_0 < 2$ and $\eta_0 > \tau_0$;*

    *(ii-d) $0 < \nu_0 = \tau_0 \le 1$ and $\eta_0 \ge \tau_0$.*

The constants $C_0$ and $C_1$ in Theorem 1 can be explicitly specified, although the expressions are complicated and thus suppressed. See the proofs in Appendix C for details. Similarly, the explicit expressions of constants $C_0$ and $C_1$ appearing in other formal statements in this subsection are suppressed as well. The convergence bound (12) exhibits two types of dependency on $\kappa$, determined by how large the stepsize $s$ is allowed. A similar phenomenon occurs in the comparison of NAG-SC and HB. Under the conditions in Theorem 1(i), algorithm (10) achieves a usual convergence bound $O((1 - C/\kappa)^k)$ for $s \asymp \frac{\mu}{L^2}$, resulting in an iteration complexity $O(\kappa \log(\frac{1}{\epsilon}))$. Under the conditions in Theorem 1(ii), algorithm (10) reaches an accelerated convergence bound $O((1 - C/\sqrt{\kappa})^k)$ for $s \asymp \frac{1}{L}$, resulting in an iteration complexity $O(\sqrt{\kappa}\log(\frac{1}{\epsilon}))$. In particular, TMM with $(\eta_0, \nu_0, \tau_0) = (1, 1, 2)$ is covered by condition (ii-a), whereas NAG-SC with $(\eta_0, \nu_0, \tau_0) = (1, 1, 1)$ is covered by condition (ii-d). It is interesting that for both TMM and NAG-SC, the choice of $(\eta_0, \nu_0, \tau_0)$ lies on boundaries of the acceleration regions identified in Theorem 1(ii).[4] In addition, although no algorithm in class (10) leads to exactly HB, there are close variations of HB which are in the class (10) with constant $(\eta, \nu, \tau)$ and covered by the non-acceleration regions in Theorem 1(i).[5]

We next extend the constant case to a more general scenario where $(\eta, \nu, \tau)$ are analytical functions of $\sqrt{q}$, free of negative exponents like $q^{-1/2}$, as stated in Assumption 1. In particular, $(\eta, \nu, \tau)$ that are polynomials of $\sqrt{q}$ satisfy Assumption 1.

**Assumption 1.** There exist non-negative, analytic functions $\tilde{\eta}(\cdot)$, $\tilde{\nu}(\cdot)$ and $\tilde{\tau}(\cdot)$ in a neighborhood of 0 such that $\eta = \tilde{\eta}(\sqrt{q})$, $\nu = \tilde{\nu}(\sqrt{q})$ and $\tau = \tilde{\tau}(\sqrt{q})$. Then $\eta$, $\nu$ and $\tau$ admit (convergent) Taylor expansions:

$$\eta = \sum_{i=0}^{\infty} \eta_i(\sqrt{q})^i, \quad \nu = \sum_{i=0}^{\infty} \nu_i(\sqrt{q})^i, \quad \tau = \sum_{i=0}^{\infty} \tau_i(\sqrt{q})^i, \tag{13}$$

when $0 \le q \le q_0$ for some constant $q_0 > 0$.

The following result gives sufficient conditions on the convergence of the algorithm (10), in terms of only the constant coefficients, $(\eta_0, \nu_0, \tau_0)$, in the expansions (13).

---

[4]The TMM choice, $\eta_0 = \nu_0 = 1$ and $\tau_0 = 2$, satisfies $\nu_0 \ne \tau_0$ and $\eta_0 = \nu_0\tau_0/2$, lying on the boundary of condition (ii-a) in Theorem 1. The NAG-SC choice, $\eta_0 = \nu_0 = \tau_0 = 1$, satisfies $\nu_0 = \tau_0 = 1$ and $\eta_0 = \tau_0$, lying on the boundary of condition (ii-d) in Theorem 1.

[5]For any constant $0 < \nu_0 \ne 1$, taking $(\eta, \tau, \nu) = (0, 1, \nu_0)$ in (11) yields the update $x_{k+1} = x_k - \frac{\nu_0}{1+\sqrt{q}}s\nabla f(x_k) + \frac{1-\nu_0\sqrt{q}}{1+\sqrt{q}}(x_k - x_{k-1})$, which closely resembles HB (9) as $\nu_0 \to 1$ and achieves non-accelerated convergence by Theorem 1(i-a).

**Theorem 2.** *Suppose that $f \colon \mathbb{R}^n \to \mathbb{R}$ is a function in $\mathcal{S}_{\mu,L}^1$ with $0 < \mu \le L$ and Assumption 1 holds.*

*(i) For $0 < s \le C_0 \frac{\mu}{L^2}$ (non-accelerated convergence), the iterates of (10) satisfy (12), with $C_0, C_1 > 0$ depending only on the functions $(\tilde{\eta}, \tilde{\nu}, \tilde{\tau})$ (independent of $q$), provided that condition (i-a) in Theorem 1 holds.*

*(ii) For $0 < s \le C_0 \frac{1}{L}$ (accelerated convergence), the iterates of (10) satisfy (12), with $C_0, C_1 > 0$ depending only on the functions $(\tilde{\eta}, \tilde{\nu}, \tilde{\tau})$ (independent of $q$), provided that condition (ii-a) in Theorem 1 holds.*

Theorem 2 serves as a generalization to Theorem 1, cases (i-a) and (ii-a). When $(\eta, \nu, \tau)$ depend on $q$ with leading constants $\nu_0 = \tau_0 > 0$ and $\eta_0 \ge \tau_0^2/2$, which is not addressed by Theorem 2, the following result gives sufficient conditions on convergence of algorithm (10), involving the coefficients $(\eta_1, \nu_1, \tau_1)$ of linear terms in the expansions (13). However, in the degenerate case of constant parameters $(\eta, \nu, \tau)$, the sufficient conditions in Theorem 3 are more restrictive than those in Theorem 1.

**Theorem 3.** *Suppose that $f \colon \mathbb{R}^n \to \mathbb{R}$ is a function in $\mathcal{S}_{\mu,L}^1$ with $0 < \mu \le L$ and Assumption 1 holds with $\nu_0 = \tau_0 > 0$ and $\eta_0 \ge \tau_0^2/2$.*

*(i) For $0 < s \le C_0 \frac{\mu}{L^2}$ (non-accelerated convergence), the iterates of (10) satisfy (12), with constants $C_0, C_1 > 0$ depending only on the functions $(\tilde{\eta}, \tilde{\nu}, \tilde{\tau})$ (independent of $q$), provided that the following holds:*

$$\text{(i-a) } \eta_0 = \tau_0^2/2, \ \nu_1 - \tau_1 < \tau_0 \left( \tfrac{\tau_0}{2} - 1 \right), \text{ and } 2\eta_1 < \nu_1 \tau_1 + \tfrac{\tau_0^2}{2} \left( \tfrac{5}{2} \tau_0 - 2 \right).$$

*(ii) For $0 < s \le C_0 \frac{1}{L}$ (accelerated convergence), the iterates of (10) satisfy (12), with constants $C_0, C_1 > 0$ depending only on the functions $(\tilde{\eta}, \tilde{\nu}, \tilde{\tau})$ (independent of $q$), provided that one of the following conditions holds:*

$$\text{(ii-a) } \eta_0 = \tau_0^2/2, \ \nu_1 - \tau_1 < \tau_0 \left( \tfrac{\tau_0}{2} - 1 \right), \text{ and } 2\eta_1 \ge \nu_1 \tau_1 + \tfrac{\tau_0^2}{2} \left( \tfrac{5}{2} \tau_0 - 2 \right);$$

$$\text{(ii-b) } \eta_0 > \tau_0^2/2 \text{ and } \nu_1 - \tau_1 < (\tau_0 - 1)\tau_0 - \eta_0;$$

$$\text{(ii-c) } \eta_0 > \tau_0^2/2 \text{ and } (\tau_0 - 1)\tau_0 - \eta_0 < \nu_1 - \tau_1 < \eta_0 - \tau_0.$$

All the preceding results are applicable to algorithm (10) or equivalently its single-variable form (11) with structured coefficients. To facilitate comparison and interpretation, we translate Theorem 2 in terms of a single-variable form (14) below with unstructured coefficients. It remains an open question to directly analyze (14), for example, via the LMI/SDP framework.

**Corollary 1.** *For $f \colon \mathbb{R}^n \to \mathbb{R}$ in $\mathcal{S}_{\mu,L}^1$ with $0 < \mu \le L$, consider the following algorithm:[6]*

$$x_{k+1} = x_k - (c_0 + R_1)s\nabla f(x_k) + (1 - c_1\sqrt{q} + R_2)(x_k - x_{k-1}) - \left(c_2\sqrt{c_0} - \frac{c_0}{2} + R_3\right)s(\nabla f(x_k) - \nabla f(x_{k-1})),$$

$$(14)$$

*where $x_1 = x_0 - h_1 s\nabla f(x_0)$; $c_0, c_1, c_2 > 0$ are constants independent of $q$; $R_1 = O(\sqrt{q})$, $R_2 = O(q)$, $R_3 = O(\sqrt{q})$, and $h_1$ are analytic functions of $\sqrt{q}$ around 0.*

*(i) For $0 < s \le C_0 \frac{\mu}{L^2}$ (non-accelerated convergence), the iterates of (14) satisfy (12), with $C_0, C_1 > 0$ depending only on $(c_0, c_1, c_2)$ and the forms of $(R_1, R_2, R_3)$ (independent of $q$), provided $c_1^2 > 4c_0$ and $c_0/4 \le c_2^2 < c_0$.*

*(ii) For $0 < s \le C_0 \frac{1}{L}$ (accelerated convergence), the iterates of (14) satisfy (12), with $C_0, C_1 > 0$ depending only on $(c_0, c_1, c_2)$ and the forms of $(R_1, R_2, R_3)$ (independent of $q$), provided $c_1^2 > 4c_0$ and $c_2^2 \ge c_0$.*

To the best of our knowledge, Corollary 1, *for the first time*, identifies concrete conditions for accelerated and non-accelerated convergence in a general class of algorithms (14). Moreover, the conditions are defined in

---

[6]The seemingly unnatural parameterization of the gradient correction coefficient is motivated by the HAG algorithm in Section 5.

terms of two parameters $c_1$ and $c_2$, which control the momentum and gradient correction terms, respectively, while the other parameter $c_0$ can be viewed as a non-essential rescaling parameter. We discuss these two conditions in detail.

First, the condition $c_1^2 > 4c_0$ for the momentum coefficient is due to implicit constraints in the coefficients of (14) when translated from Theorem 2 in terms of (11). Although NAG-SC (3) and HB (9) can both be put into (14) with parameters $c_0 = 1$, $c_1 = 2$, $c_2 = 3/2$, and $c_0 = 1$, $c_1 = 2$, $c_2 = 1/2$, respectively, their convergences are not covered by Corollary 1 due to that the condition $c_1^2 > 4c_0$ does not hold. Nonetheless, if viewing NAG-SC (3) and HB (9) as the limiting algorithms when $c_1 \downarrow 2$, then NAG-SC (3) achieves an accelerated convergence by Corollary 1 part (ii) while HB (9) achieves a non-accelerated convergence by part (i). See Table 1 for a summary. We believe that $c_1^2 > 4c_0$ is technical and can be potentially relaxed to $c_1 > 0$. See Appendix A for a detailed discussion of the connection with related work in Sanz Serna & Zygalakis (2021).

Second, the condition $c_2^2 \geq c_0$ or $c_0/4 \leq c_2^2 < c_0$ directly distinguishes between the accelerated or non-accelerated convergence. In other words, the leading constant $c_2\sqrt{c_0} - c_0/2$ in the coefficient of gradient correction needs to exceed a threshold $c_0/2$.

For numerical illustrations of the theoretical results in Corollary 1, we apply (14) to two quadratic functions on $\mathbb{R}^2$, similarly to Su et al. (2016), under different parameters and step sizes. It is particularly of interest to compare the performance of the parameter choices falling inside (or on the boundaries) versus outside the sufficient conditions for achieving accelerated convergence in our theoretical results. Specifically, we take $c_0$ fixed at 1, $c_1 = 1, 2$ (boundary) and $c_2 = 1/2, 1$ (boundary), $3/2$. All remainder terms $R_1$, $R_2$, and $R_3$ in (14) are taken to be 0. Here $c_1 = 1, 2$ corresponds to under-damping ($c_1^2 < 4c_0$) and critical-damping ($c_1^2 = 4c_0$) respectively, and $c_2 = 1/2, 1, 3/2$ corresponds to the leading constant of the gradient correction coefficient in (14) being $0, 1/2, 1$ respectively. For NAG-SC (4), $c_1 = 2$ and $c_2 = 3/2$, and for heavy-ball (9), $c_1 = 2$ and $c_2 = 1/2$, but both with nonzero remainder terms $R_2$ and $R_3$. Hence, the algorithms tested are not exactly NAG-SC or the heavy-ball method.

From Figure 2 we observe that critical-damping ($c_1 = 2$) results in a faster convergence than under-damping ($c_1 = 1$). Increasing $c_2$ (gradient correction) also tends to improve the performance, but to a relatively small extent in the ill-conditioned case once $c_2 \geq 1$. The error plots when $c_2 = 1/2$ in the ill-conditioned case appear to be a sum of two oscillations. Overall, the numerical results are consistent with our theoretical results in the strongly convex setting. Better performances are observed from algorithms with more friction (i.e., a large $c_1$) and larger gradient correction (i.e., a large $c_2$).

## 3 Acceleration for convex functions

In this section, we first give background on several algorithms for minimizing smooth and convex functions. Then we study a general class of algorithms.

### 3.1 Review of accelerated gradient methods

Consider the unconstrained minimization problem (1) with $f \in \mathcal{F}_L^1$. For stepsize $0 < s \leq 1/L$, the GD iterates $\{f(x_k)\}$ is non-increasing and satisfy $f(x_k) - f^\star \leq \|x_0 - x^\star\|^2/(2ks)$.[7] However, the $O(1/k)$ rate is not optimal. Nesterov (1988) proposed an accelerated gradient method in a similar form as NAG-SC (3): for $k \geq 0$,

$$
\begin{aligned}
y_{k+1} &= x_k - s\nabla f(x_k), \\
x_{k+1} &= y_{k+1} + \sigma_{k+1}(y_{k+1} - y_k),
\end{aligned}
\tag{15}
$$

with $x_0 = y_0$ and $\sigma_{k+1} = k/(k+3)$. Compared with NAG-SC (3), the momentum coefficient $\sigma_{k+1}$ varies with $k$, instead of taking a fixed value depending on the strong-convexity parameter $\mu$. For $0 < s \leq 1/L$,

---

[7]For stepsize $1/L < s < 2/L$, the GD iterates $\{f(x_k)\}$ is still non-increasing but satisfies a slightly different bound $O\left(\frac{\|x_0 - x^\star\|^2}{ks(2-Ls)}\right)$ for the objective gap (Nesterov, 2018).

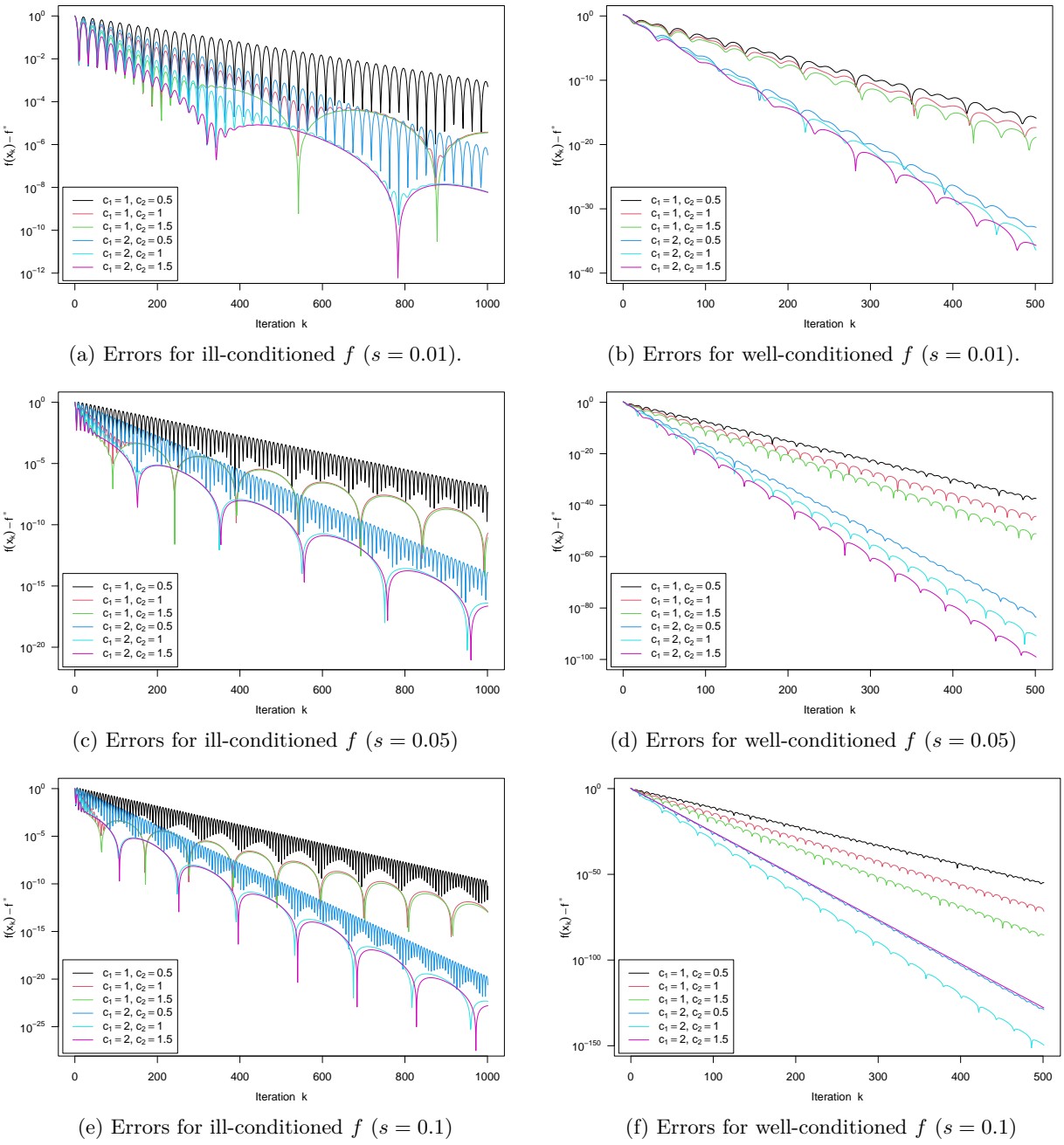

(a) Errors for ill-conditioned $f$ $(s = 0.01)$.

(b) Errors for well-conditioned $f$ $(s = 0.01)$.

(c) Errors for ill-conditioned $f$ $(s = 0.05)$

(d) Errors for well-conditioned $f$ $(s = 0.05)$

(e) Errors for ill-conditioned $f$ $(s = 0.1)$

(f) Errors for well-conditioned $f$ $(s = 0.1)$

Figure 2: Minimizing strongly convex functions by (14). The left column is for ill-conditioned $f(x_1, x_2) = 5 \times 10^{-3} x_1^2 + x_2^2$, and the right column is for well-conditioned $f(x_1, x_2) = 5 \times 10^{-1} x_1^2 + x_2^2$. Fix $c_0 = 1$. The initial iterates are $x_0 = (1, 1)$ and $x_1 = x_0 - 2s\nabla f(x_0)/(1 + \sqrt{\mu s})$.

the NAG-C iterates $\{f(x_k)\}$ may not be non-increasing but satisfy

$$f(x_k) - f^\star = O\left(\frac{\|x_0 - x^\star\|^2}{sk^2}\right). \tag{16}$$

Compared with GD, the iteration complexity for $f(x_k) - f^\star \leq \epsilon$ is reduced from $O(L/\epsilon)$ to $O(\sqrt{L/\epsilon})$ when taking $s = 1/L$. Notably, the convergence bound in (16) matches with the lower bound of black-box first-order methods for minimizing functions in $\mathcal{F}_L^1$ when $n$ (the dimension of $x$) is relatively large compared with $k$ (Nesterov, 2018).

The single-variable form in terms of $\{x_k\}$ for NAG-C (15) is: for $k \geq 1$,

$$x_{k+1} = \underbrace{x_k - s\nabla f(x_k)}_{\text{gradient descent}} + \underbrace{\sigma_{k+1}(x_k - x_{k-1})}_{\text{momentum}} - \underbrace{\sigma_{k+1} \cdot s(\nabla f(x_k) - \nabla f(x_{k-1}))}_{\text{gradient correction}}, \tag{17}$$

with $x_0$, $x_1 = x_0 - s\nabla f(x_0)$, and $\sigma_{k+1} = k/(k+3)$. The iterate (17) involves an additional momentum term and a gradient correction term similarly as in (4) for NAG-SC. Shi et al. (2022) studied NAG-C by relating (17) to a high-resolution ODE (see Section 4.2) and obtained a new result on the squared gradient norm: for stepsize $0 < s \leq 1/(3L)$,

$$\min_{0 \leq i \leq k} \|\nabla f(x_i)\|^2 = O\left(\frac{\|x_0 - x^\star\|^2}{s^2(k+1)^3}\right). \tag{18}$$

The inverse cubic rate (18) cannot be obtained directly from (16). Moreover, Shi et al. (2022) extended the coefficients in the momentum and gradient correction terms in (17) to $\sigma_{k+1} = k/(k+r+1)$ and $\sigma_{k+1}\beta s$ respectively, and showed that the two bounds in (16) and (18) remain valid for any $r \geq 2$ and $\beta > 1/2$.

The form of momentum coefficient $\sigma_{k+1}$ can be made more general. For instance, a popular scheme is to set $\sigma_{k+1} = (\alpha_k - 1)/\alpha_{k+1}$, where $\{\alpha_k\}$ is a scalar sequence to be chosen. It is known (Beck, 2017) that for any sequence $\{\alpha_k\}$ satisfying $\alpha_k = \Omega(k)$ and a recursive condition[8]

$$\alpha_{k+1}(\alpha_{k+1} - 1) \leq \alpha_k^2, \tag{19}$$

the corresponding algorithm (15) achieves the optimal bound (16). For the well-known accelerated proximal gradient method or FISTA (Beck & Teboulle, 2009), $\{\alpha_k\}$ is defined recursively as $\alpha_{k+1} = (1 + \sqrt{1 + 4\alpha_k^2})/2$ with $\alpha_0 = 1$, i.e., (19) holds as equality. It can also be verified that $\{\alpha_k = (k+r)/r\}$ satisfies (19) IFF $r \geq 2$, with corresponding $\sigma_{k+1} = k/(k+r+1)$.

### 3.2 Main results

We formulate a broad class of algorithms including NAG-C (15) and existing variations (Beck, 2017; Shi et al., 2022) and establish sufficient conditions for when the algorithms in the class achieve both the optimal bound (16) for the objective gap and the inverse cubic rate (18) for the squared gradient norm, similar to NAG-C.

To unify and extend existing choices of the momentum and gradient correction terms related to NAG-C, we consider the following class of algorithms: for $k \geq 0$,

$$\begin{aligned} y_{k+1} &= x_k - \beta_k s\nabla f(x_k), \\ x_{k+1} &= x_k - \gamma_k s\nabla f(x_k) + \sigma_{k+1}(y_{k+1} - y_k), \end{aligned} \tag{20}$$

where $x_0 = y_0$, $\sigma_{k+1} = (\alpha_k - 1)/\alpha_{k+1}$, and $\{\alpha_k\}$, $\{\beta_k\}$ and $\{\gamma_k\}$ are three scalar sequences. The equivalent single-variable form of (20) is for $k \geq 1$,

$$x_{k+1} = x_k - (\gamma_k + \sigma_{k+1}(\beta_k - \beta_{k-1}))s\nabla f(x_k) + \sigma_{k+1}(x_k - x_{k-1}) - \sigma_{k+1}\beta_{k-1} \cdot s(\nabla f(x_k) - \nabla f(x_{k-1})), \tag{21}$$

starting from $x_0$ and $x_1 = x_0 - (\gamma_0 + (\alpha_0 - 1)\beta_0/\alpha_1)s\nabla f(x_0)$. For an overview of prior work on acceleration covered by our analysis of the general class of algorithms (20) or (21), see Table 2. To highlight the main points, Proposition 2 was stated in Section 1 for a special case of (21) with constant $\gamma_k \equiv 1$ and $\beta_k \equiv \tilde{c}_2$.

---

[8]The recursive condition (19) implies that $\alpha_{k+1} \leq (1 + \sqrt{1 + 4\alpha_k^2})/2 \leq \alpha_k + 1$, and hence $\alpha_k = \Theta(k)$.

Table 2: Existing algorithms covered by our analysis

| Method | $\beta_k$ | $\gamma_k$ | $\alpha_k$ | $\sigma_{k+1}$ | Convergence |
|---|---|---|---|---|---|
| NAG-C (15) or (17) | 1 | 1 | $(k+2)/2$ | $k/(k+3)$ | Acceleration by Theorem 4 |
| Beck (2017) | 1 | 1 | any $\alpha_k$ satisfying (19) | $(\alpha_k - 1)/\alpha_{k+1}$ | Acceleration by Theorem 4 |
| Shi et al. (2022) | $\beta$ | 1 | $(k+r)/r$ for $r \geq 2$ | $k/(k+r+1)$ | Acceleration by Theorem 4 |

Motivated by the Lyapunov analysis of NAG-C in Su et al. (2016), we introduce the following three-variable form of (20). Define $z_k = \alpha_k x_k + (1 - \alpha_k)y_k$ for $k \geq 0$, which is reminiscent of (5b) and (7) in the strongly convex setting. Then (20) can be equivalently reformulated as: for $k \geq 0$,

$$y_{k+1} = x_k - \beta_k s \nabla f(x_k), \tag{22a}$$

$$z_{k+1} = z_k - \widetilde{\alpha}_k s \nabla f(x_k), \tag{22b}$$

$$x_{k+1} = \frac{1}{\alpha_{k+1}} z_{k+1} + \left(1 - \frac{1}{\alpha_{k+1}}\right) y_{k+1}, \tag{22c}$$

starting from $x_0 = z_0$, where $\widetilde{\alpha}_k = \beta_k \alpha_k + (\gamma_k - \beta_k)\alpha_{k+1}$. See Appendix D for a proof. By definition, given $\{\beta_k\}$ and $\{\alpha_k > 0\}$, there is a one-to-one correspondence between $\{\gamma_k\}$ and $\{\widetilde{\alpha}_k\}$: $\gamma_k = (\widetilde{\alpha}_k - \beta_k(\alpha_k - \alpha_{k+1}))/\alpha_{k+1}$. Hence, algorithm (20) or (22) can be considered to be directly parameterized by the three sequences $\{\alpha_k\}$, $\{\beta_k\}$ and $\{\widetilde{\alpha}_k\}$. The three-variable form (22) is introduced mainly to facilitate our Lyapunov analysis, which is presented in Section 6. The $z_k$ update (22b) appears to differ from (10b) in the strongly convex case, although (22c) resembles (10c) in the sense that the weight of $z_k$ or $y_k$ in the update of $x_k$ goes to 0 or 1 respectively as $\alpha_k \to \infty$ in (22c) or $s \to 0$ in (10c).

Our main result gives sufficient conditions on when algorithm (20) or (22) achieves accelerated convergence in both the objective gap and gradient norm, similar to NAG-C. We discuss interpretations of the conditions from the perspective of HAG in Section 5.

**Theorem 4.** *Let $f \colon \mathbb{R}^n \to \mathbb{R}$ be a function in $\mathcal{F}_L^1$ with $L > 0$. Assume that the following conditions jointly hold:*

*(i) $\lim_k \beta_k = \beta$ and $\lim_k \gamma_k = \gamma$ with $\beta > \gamma/2 > 0$;*

*(ii) $\{\alpha_k > 0\}$ satisfies that $\alpha_k = \Omega(k)$, $\lim_k \alpha_{k+1}/\alpha_k = 1$, $\alpha_{k+1}(\alpha_{k+1} - 1) \leq \alpha_k^2$;*

*(iii) $\{\widetilde{\alpha}_k/\alpha_k\}$ is monotone (either non-increasing or non-decreasing) in $k$.*

*Then for $0 < s \leq C_0/L$ with $C_0 = \frac{2 - \gamma/\beta}{2\beta} \wedge \frac{1}{4\gamma} > 0$, the iterates $\{x_k\}$ from (20), (21) or (22) satisfy the bound (16) for the objective gap and the inverse cubic rate (18) for the squared gradient norm.*

For Theorem 4, the existence of the limits, $\lim_k \beta_k = \beta$, $\lim_k \gamma_k = \gamma$, and $\lim_k \alpha_{k+1}/\alpha_k = 1$, are introduced mainly to simplify the sufficient conditions and may be relaxed even with the same Lyapunov function in our proofs. The monotonicity condition on $\{\widetilde{\alpha}_k/\alpha_k\}$ may also be relaxed by using other Lyapunov functions. In the setting of $\beta_k \equiv \beta$ and $\gamma_k \equiv \gamma$, we have $\widetilde{\alpha}_k/\alpha_k = \beta + (\gamma - \beta)\alpha_{k+1}/\alpha_k$. Then the monotonicity condition is equivalent to either requiring $\beta = \gamma$ without any additional constraint on $\{\alpha_k\}$ or, if $\beta \neq \gamma$, requiring that $\{\alpha_{k+1}/\alpha_k\}$ is monotone. The latter condition is satisfied by both the linear choice $\alpha_k = (k + r)/r$ and the iterative choice $\alpha_{k+1} = (1 + \sqrt{1 + 4\alpha_k^2})/2$ with $\alpha_0 = 1$ in FISTA.

We also point out that the conditions in Theorem 4 are general enough to allow the absence of a limiting ODE for (20). As described in Section 4, a necessary condition for the existence of a limiting ODE of (20) is that $\lim_k k(1 - \sigma_{k+1})$ exists. However, we provide an example where the conditions in Theorem 4 are satisfied, but $\lim_k k(1 - \sigma_{k+1})$ does not exist.

**Lemma 1.** *Consider the sequence $\{\alpha_k\}$ defined by alternating two rules:*

$$\alpha_k = \begin{cases} (k+r)/r, & \text{if } k \text{ is even,} \\ (1 + \sqrt{1 + 4\alpha_{k-1}^2})/2. & \text{if } k \text{ is odd,} \end{cases}$$

*with $\alpha_0 = 1$. Then for any $r \geq 2$, condition (ii) in Theorem 4 holds. But $\lim_k k(1 - \sigma_{k+1})$ does not exist if $r > 2$: along $\{k'\} = \{2k\}$ and $\{k''\} = \{2k+1\}$, we have*

$$\lim_{k' \to \infty} k'(1 - \sigma_{k'+1}) = 3r/2, \quad \lim_{k'' \to \infty} k''(1 - \sigma_{k''+1}) = 2 + r/2.$$

Lemma 1 shows that accelerated convergence can be achieved by algorithm (20), independent of whether a limiting ODE exists.[9] Hence, convergence of discrete algorithms may not always be explained from the ODE perspective.

For numerical illustrations of the theoretical results in Theorem 4, we apply (21) to various objective functions under different parameters and step sizes. It is particularly of interest to compare the performance of the parameter choices falling inside (or on the boundaries) versus outside the sufficient conditions for achieving accelerated convergence in our theoretical results. Specifically, we consider a quadratic objective $f(x) = x^{\mathrm{T}}Ax/2 + b^{\mathrm{T}}x$ and the log-sum-exp objective $f(x) = \rho \log \sum_{i=1}^{200} \exp\{(a_i^{\mathrm{T}}x - b_i)/\rho\}$, following Su et al. (2016). The quadratic function is strongly convex but with $\mu \approx 0$, and the log-sum-exp function is not strongly convex. For algorithm parameters, we set $\gamma \equiv 1$, $\sigma_{k+1} = \frac{k}{k+r+1}$ for $r = 1, 2$ (boundary) and $\beta = 0, 0.5$ (boundary), 1. In particular, NAG-C corresponds to $r = 2$ and $\beta = 1$. In the log-sum-exp case, the minimizer has no closed form, so we approximate $f^\star$ by running NAG-C, which converges fast in hundreds of iterations using a relatively large step size.

Figures 3 and 4 present the traces of the scaled optimality gap and the scaled squared gradient norm. In both Figures 3 and 4, for fixed $\beta$, increasing $r = 1$ to $r = 2$ significantly accelerates the convergence of $f(x_k) - f^\star$ and $\min_{0 \leq i \leq k} \|\nabla f(x_i)\|^2$. In particular, it can be observed in Figure 3 (b), (d), and (f) that the bound $\min_{0 \leq i \leq k} \|\nabla f(x_i)\|^2 = O(1/k^3)$ may fail when $r = 1$ since the product $(k+1)^3 \min_{0 \leq i \leq k} \|\nabla f(x_i)\|^2$ appears to increase in an unbounded way as $k$ increases. From Figure 4, for fixed $r$, increasing $\beta$ markedly improves the performance in decreasing $f(x_k) - f^\star$ and $\min_{0 \leq i \leq k} \|\nabla f(x_i)\|^2$, especially when the step size is large as in Figure 4 (e). A similar beneficial effect of a larger $\beta$ can be observed from Figure 3 in decreasing $\min_{0 \leq i \leq k} \|\nabla f(x_i)\|^2$, but the effect there is less obvious in decreasing $f(x_k) - f^\star$, where all algorithms exhibit oscillations during iterations, and the oscillations become stronger as the step size increases. Overall, the numerical results are consistent with our theoretical results. Better performances are observed from algorithms with more friction (i.e., a larger $r$) and larger gradient correction (i.e., a larger $\beta$).

## 4 ODE connection and comparison

As the stepsize $s$ vanishes to 0 in a discrete algorithm, the limiting ODE (if exists) can be studied to understand the behavior of its discrete counterpart (Su et al., 2016; Shi et al., 2022). We study the convergence rates of limiting ODEs of algorithms in Sections 2 and 3, and compare them with the discrete results.

We briefly review how the limiting ODEs from NAG-SC (4) and NAG-C (17) exhibit interesting differences from that of the vanilla gradient descent (2), as discussed in Su et al. (2016). On one hand, by taking $\Delta t = s$ and $x_k = X(t_k) = X(ks)$ for a continuous-time trajectory $X_t = X(t)$, the limit of gradient descent (2) as $s \to 0$ is the gradient flow

$$\dot{X}_t + \nabla f(X_t) = 0, \tag{23}$$

with $X(0) = x_0$. It is known that when $f \in \cup_{L>0}\mathcal{F}_L^1$, (23) has a unique solution $X_t$, which satisfies that $f(X_t) - f^\star \leq \|x_0 - x^\star\|^2/(2t)$ and when $f \in \cup_{L \geq \mu}\mathcal{S}_{\mu,L}^1$, $f(X_t) - f^\star \leq e^{-2\mu t}(f(x_0) - f^\star)$. By relating $X_t$ to $x_k$ (i.e., taking $t = ks$ for small $s$), the former rate $O(1/t)$ translates into $O(1/(ks))$, which is exactly the discrete rate of gradient descent in the convex setting. The second rate $O(e^{-2\mu t})$ resembles $O((1 - 2\mu s)^k)$

---

[9]A similar conclusion can be found for algorithm (3) or (4) in the strongly convex case. See Appendix A for a detailed discussion.

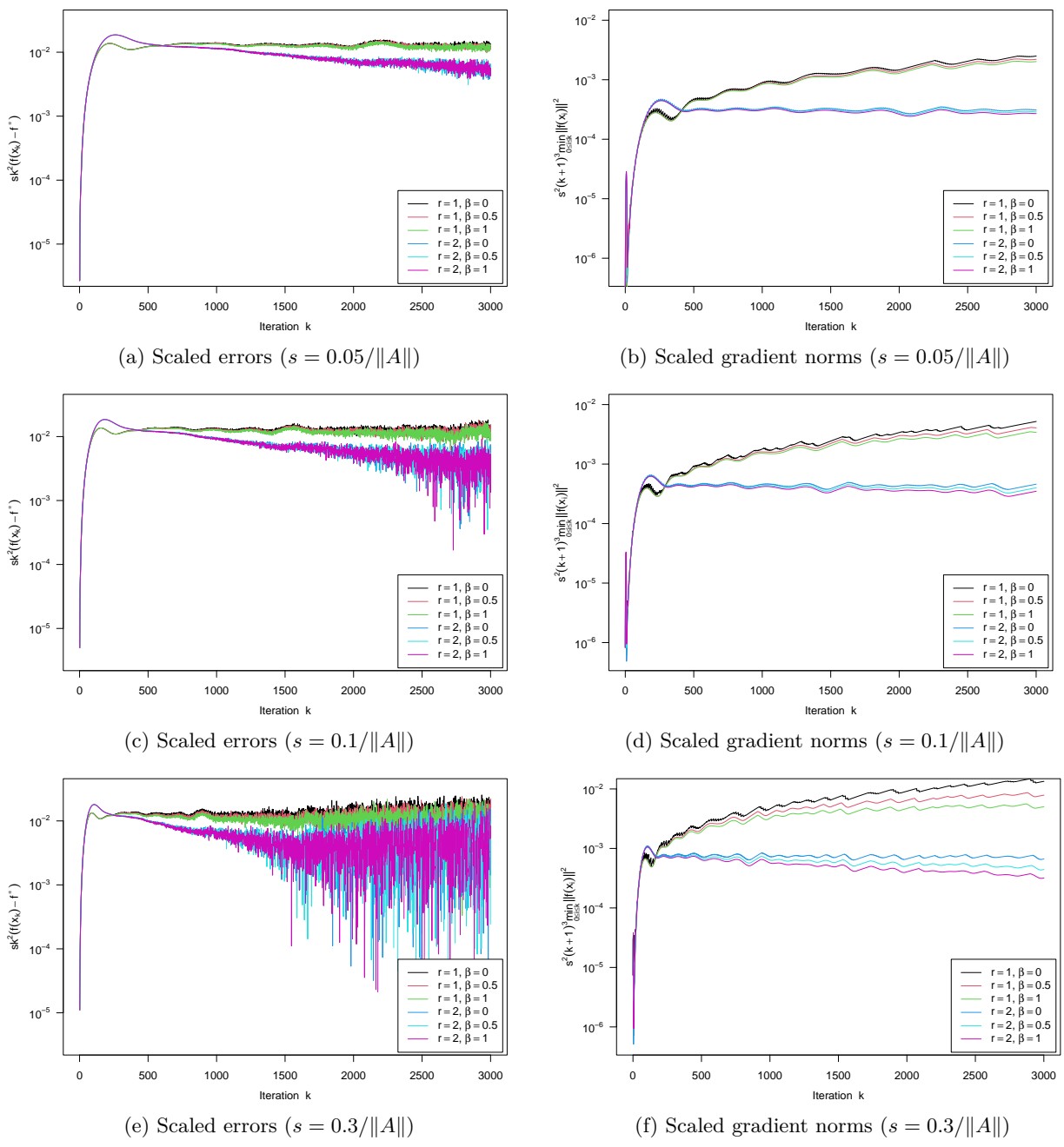

Figure 3: Scaled errors and squared gradient norms in minimizing $f(x) = \frac{1}{2}x^{\mathrm{T}}Ax + b^{\mathrm{T}}x$ by (29) under different step sizes, where $A = B^{\mathrm{T}}B$ for $B \in \mathbb{R}^{500 \times 500}$, $b \in \mathbb{R}^{500}$. All entries in $B$ and $b$ are i.i.d. draws from $U(0,1)$, and $\|A\|$ is the spectral norm of $A$. We take $\gamma = 1$, $\sigma_{k+1} = \frac{k}{k+r+1}$ for $r = 1, 2$ and $\beta = 0, 0.5, 1$ in (29).

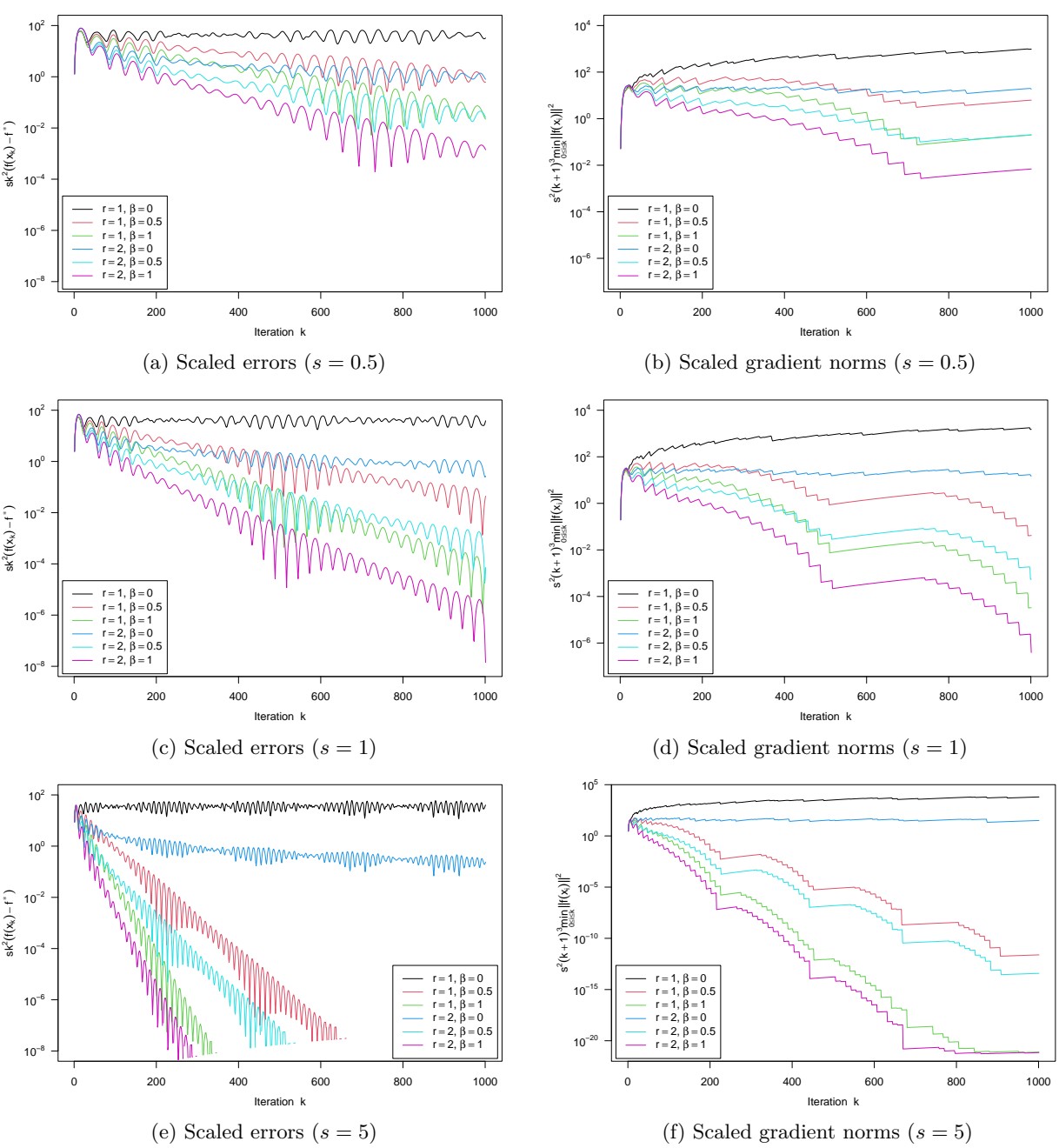

(a) Scaled errors ($s = 0.5$)

(b) Scaled gradient norms ($s = 0.5$)

(c) Scaled errors ($s = 1$)

(d) Scaled gradient norms ($s = 1$)

(e) Scaled errors ($s = 5$)

(f) Scaled gradient norms ($s = 5$)

Figure 4: Scaled errors and squared gradient norms in minimizing $f(x) = \rho \log \sum_{i=1}^{200} e^{\frac{a_i^{\mathrm{T}} x - b_i}{\rho}}$ by (29) under different step sizes, where $A = [a_1, \ldots, a_{200}] \in \mathbb{R}^{50 \times 200}$, $b \in \mathbb{R}^{200}$, and $\rho = 20$. All entries in $A$ and $b$ are i.i.d. draws from $N(0,1)$. We take $\gamma = 1$, $\sigma_{k+1} = \frac{k}{k+r+1}$ for $r = 1, 2$ and $\beta = 0, 0.5, 1$ in (29).

for $s \approx 0$, which is similar to the discrete rate $O((1 - \frac{2\mu s}{1 + 1/\kappa})^k)$ for gradient descent in the strongly convex setting.

On the other hand, by taking $\Delta t = \sqrt{s}$ and $x_k = X(t_k) = X(k\sqrt{s})$ for a continuous-time trajectory $X_t = X(t)$, the limit of NAG-C (17) as $s \to 0$ is for $t > 0$,

$$\ddot{X}_t + \frac{3}{t}\dot{X}_t + \nabla f(X_t) = 0, \tag{24}$$

with initial conditions $X(0) = x_0$ and $\dot{X}(0) = 0$. When $f \in \cup_{L>0}\mathcal{F}_L^1$, (24) has a unique solution $X_t$, which satisfies that $f(X_t) - f^\star \leq 2\|x_0 - x^\star\|^2/t^2$. The limit of NAG-SC (4) is

$$\ddot{X}_t + 2\sqrt{\mu}\dot{X}_t + \nabla f(X_t) = 0, \tag{25}$$

with initial conditions $X(0) = x_0$ and $\dot{X}(0) = 0$. When $f \in \cup_{L\geq\mu}\mathcal{S}_{\mu,L}^1$, (25) has a unique solution $X_t$, which satisfies that $f(X_t) - f^\star \leq 2e^{-\sqrt{\mu}t}(f(x_0) - f^\star)$. For a fixed $t$, $X_t$ can be approximated by the discrete iterate $x_k$ with $t = k\sqrt{s}$ for a small $s$. Using $t = k\sqrt{s}$, the former $O(1/t^2)$ rate translates into $O(1/(sk^2))$, which is the discrete rate for NAG-C. The latter $O(e^{-\sqrt{\mu}t})$ rate matches the discrete rate $O((1 - \sqrt{\mu s})^k)$ for NAG-SC. In these cases, the convergence rate of the continuous-time trajectory $X_t = X(t)$ matches that of the discrete iterates $\{x_k\}$.

The overall findings of our study can be summarized as follows.

- For minimization of strongly convex functions, the ODE convergence bounds do not directly inform the range of feasible stepsizes, which are crucial in determining whether acceleration is achieved.

- For minimization of either strongly convex or convex functions, the gradient correction term vanishes in the (low-resolution) ODEs. Although this term is explicitly retained in high-resolution ODEs, these ODEs can still converge at the same rate with or without it.

In conclusion, the convergence properties of limiting ODEs, including high-resolution ODEs, fall short of informing whether discrete algorithms achieve acceleration. Our work provides explicit convergence bounds for the low- and high-resolution ODEs through new Lyapunov analyses. These results complement and enrich the existing literature on the connections and gaps between discrete algorithms and ODEs.

## 4.1 Strongly convex setting

For strongly convex $f$, we compare the convergence of algorithm (14) and related ODEs. By taking $\Delta t = \sqrt{s}$ and $x_k = X(t_k) = X(k\sqrt{s})$, the limiting ODE of (14) as $s \to 0$ is

$$\ddot{X}_t + c_1\sqrt{\mu}\dot{X}_t + c_0\nabla f(X_t) = 0, \tag{26}$$

with initial conditions $X(0) = x_0$ and $\dot{X}(0) = 0$. This can be viewed as a Newtonian equation of motion in a viscous medium in the potential field $c_0 f$ with $c_0$ as the rescaling constant.[10] The damping coefficient is $c_1\sqrt{\mu}$, resulting from the momentum coefficient $1 - c_1\sqrt{q}$ in (14). For convenience, the three settings $c_1^2 > 4c_0$, $c_1^2 = 4c_0$, or $c_1^2 < 4c_0$ are referred to as over-damping, critical-damping, or under-damping, respectively. The parameter $c_2$ associated with the gradient correction vanishes. Nevertheless, for any smooth and $\mu$-strongly convex $f$ and any $c_0, c_1 > 0$, we show that the solution $X_t$ uniquely exists, and $f(X_t)$ converges to $f^\star$ exponentially fast with a decaying rate proportional to $\sqrt{\mu}$. To state the result, we use $C^p(I; \mathbb{R}^n)$ to denote the class of $p^{\text{th}}$ continuously differentiable maps from an interval $I$ to $\mathbb{R}^n$.

**Proposition 3.** *Let $f: \mathbb{R}^n \to \mathbb{R}$ be a function in $\cup_{L\geq\mu}\mathcal{S}_{\mu,L}^1$ with $\mu > 0$. Then the ODE (26) with initial conditions $X(0) = x_0$ and $\dot{X}(0) = 0$ has a unique solution $X_t \in C^2([0,\infty); \mathbb{R}^n)$. Moreover, for any $c_0, c_1 > 0$, the solution $X_t$ satisfies that $f(X_t) - f^\star = O(e^{-C\sqrt{\mu}t}(f(x_0) - f^\star))$ for a constant $C = \frac{c_1 - \sqrt{(c_1^2 - 4c_0)\vee 0}}{2} > 0$.[11] Otherwise (either $c_0 \leq 0$ or $c_1 \leq 0$), $X_t$ may fail to converge to $x^\star$ as $t \to \infty$.*

---

[10]It suffices to establish convergence for (26) for $c_0 = 1$, then all other cases with $c_0 > 0$ can be recovered by a time rescaling.

[11]The constant $C$ in Proposition 3 can be improved using the IQC methods (Sanz Serna & Zygalakis, 2021). However, our analysis is based on a concise Lyapunov analysis, and our goal is not to find the sharpest constant. See Appendix A for details.

The dependency on $\mu$ in the bound above is improved to $\sqrt{\mu}$, compared with the $O(\mathrm{e}^{-2\mu t})$ bound for gradient flow (23). A direct translation by $t_k = k\sqrt{s}$ suggests that a discrete bound of $O((1-C\sqrt{\mu s})^k)$ may be expected for suitable discretizations of (26), as opposed to $O((1 - C\mu s)^k)$ for gradient descent (2). However, there are notable gaps between such results suggested by ODEs and our results for discrete algorithms. First, the bound in Proposition 3 only requires $c_0, c_1 > 0$, which is much weaker than the over-damping condition $c_1^2 > 4c_0$ in Corollary 1. The condition $c_1^2 > 4c_0$ may be only technical and can be potentially relaxed. Second, the bound $O((1 - C\sqrt{\mu s})^k)$ does not directly inform whether $s \asymp 1/L$ (equivalently $q \asymp 1/\kappa$) or $s \asymp \mu/L^2$ ($q \asymp 1/\kappa^2$), corresponding to different orders of time interval in discretizing the ODE. The former leads to acceleration while the latter does not. Third, the gradient correction term from algorithm (14) vanishes in the limiting ODE (26), so that the ODE does not capture the effect of gradient correction.

To complement the preceding discussion, we study the high-resolution ODEs for NAG-SC (4) and HB (9), which are proposed to reflect the gradient correction or lack of in the two methods (Shi et al., 2022). The high-resolution ODEs are derived by retaining $O(\sqrt{s})$ terms that would otherwise vanish in the limit of $s \to 0$. By taking $\Delta t = \sqrt{s}$ and $x_k = X(t_k) = X(k\sqrt{s})$, the low-resolution ODEs for NAG-SC (4) and HB (9) are the same equation in (25). The high-resolution ODE for NAG-SC is

$$\ddot{X}_t + 2\sqrt{\mu}\dot{X}_t + \sqrt{s}\nabla^2 f(X_t)\dot{X}_t + (1 + \sqrt{\mu s})\nabla f(X_t) = 0, \tag{27}$$

with initial conditions $X(0) = x_0$ and $\dot{X}(0) = -\frac{2\sqrt{s}}{1+\sqrt{\mu s}}\nabla f(x_0)$. The high-resolution ODE for HB is

$$\ddot{X}_t + 2\sqrt{\mu}\dot{X}_t + (1 + \sqrt{\mu s})\nabla f(X_t) = 0, \tag{28}$$

with the same initial conditions. For NAG-SC, the gradient correction results in an additional Hessian term, $\sqrt{s}\nabla^2 f(X_t)\dot{X}_t$, in (27). However, the convergence rates of these two high-resolution ODEs are the same. The following result is qualitatively similar to Theorems 1 and 2 in Shi et al. (2022), but involves a sharper rate due to a new Lyapunov analysis.

**Proposition 4.** *Let $f \colon \mathbb{R}^n \to \mathbb{R}$ be a function in $\cup_{L \geq \mu}\mathcal{S}_{\mu,L}^2$ with $\mu > 0$. Then each of the ODEs (27) and (28) with initial conditions $X(0) = x_0$ and $\dot{X}(0) = -2\sqrt{s}\nabla f(x_0)/(1 + \sqrt{\mu s})$ has a unique solution $X_t \in C^2([0,\infty); \mathbb{R}^n)$. Moreover, for both ODEs, the solutions $X_t$ satisfy that $f(X_t) - f^\star \leq V_0 \mathrm{e}^{-\sqrt{\mu}t}$, where the constant $V_0 = (1 + \sqrt{\mu s})(f(x_0) - f^\star) + \frac{1}{2}\|\sqrt{\mu}(x_0 - x^\star) - \sqrt{s}\frac{1-\sqrt{\mu s}}{1+\sqrt{\mu s}}\nabla f(x_0)\|^2$ for (27) and $V_0 = (1 + \sqrt{\mu s})(f(x_0) - f^\star) + \frac{1}{2}\|\sqrt{\mu}(x_0 - x^\star) - \frac{2\sqrt{s}}{1+\sqrt{\mu s}}\nabla f(x_0)\|^2$ for (28).*

Unfortunately, despite explicitly incorporating the gradient correction (or not), high-resolution ODEs do not exhibit differences in convergence rates and therefore fail to capture the effect of gradient correction.

## 4.2 Convex setting

For convex $f$, we compare the convergence of algorithm (21) and related ODEs in the setting where $\gamma_k = \gamma$, $\beta_k = \beta$, and $\alpha_k = \frac{k+r}{r}$.[12] By rescaling the stepsize $s$ to $s/\gamma$, the parameters $\gamma$ and $\beta$ can be reset to 1 and $\beta/\gamma$ respectively. In this setting, (21) reduces to the sub-class in Shi et al. (2022), with two parameters $r$ and $\beta/\gamma$:

$$x_{k+1} = x_k - s\nabla f(x_k) + \sigma_{k+1}(x_k - x_{k-1}) - \sigma_{k+1}\frac{\beta}{\gamma} \cdot s(\nabla f(x_k) - \nabla f(x_{k-1})), \tag{29}$$

where $\sigma_{k+1} = \frac{\alpha_k - 1}{\alpha_{k+1}} = \frac{k}{k+r+1}$. By taking $\Delta t = \sqrt{s}$ and $x_k = X(t_k) = X(k\sqrt{s})$, the limiting ODE for (29) is for $t > 0$,

$$\ddot{X}_t + \frac{r+1}{t}\dot{X}_t + \nabla f(X_t) = 0, \tag{30}$$

with initial conditions $X(0) = x_0$ and $\dot{X}(0) = 0$. This can be viewed as a Newtonian equation for a particle moving in the potential field $f$ with friction. The damping coefficient is $\frac{r+1}{t}$, resulting from the momentum coefficient $\sigma_{k+1}$. Su et al. (2016) showed that if $f \in \cup_{L>0}\mathcal{F}_L^1$, (30) with the specified initial conditions has

---

[12]For a general sequence $\{\alpha_k\}$, algorithm (21) may not admit a limiting ODE, as shown in Lemma 1.

a unique solution $X_t \in C^2((0,\infty); \mathbb{R}^n) \cap C^1([0,\infty); \mathbb{R}^n)$. Moreover, the convergence of $X_t$ exhibits a phase transition at $r = 2$, the choice in NAG-C (15). If $r \geq 2$, then $f(X_t) - f^\star = O(1/t^2)$. But if $r < 2$, the rate $O(1/t^2)$ may fail as illustrated by counterexamples. For $\alpha_k = \frac{k+r}{r}$, it can be directly verified that the recursive condition $\alpha_{k+1}(\alpha_{k+1} - 1) \leq \alpha_k^2$ in Theorem 4 holds if and only if $r \geq 2$. Therefore, the condition on the momentum term for acceleration of (29) is well captured by the limiting ODE.

The ODE (30), however, does not reflect the parameter $\beta/\gamma$ associated with the gradient correction in algorithm (29), which is similarly observed in the low-resolution ODE (26) for algorithm (14) in the strongly convex setting. The gradient correction term, of order $O(\sqrt{s})$, can be incorporated in a high-resolution ODE (Shi et al., 2022). By taking $t_0 = \frac{(r+1)\sqrt{s}}{2}$ and $t_k = t_0 + k\sqrt{s}$ for algebraic convenience, the high-resolution ODE for (21) is for $t \geq t_0 > 0$,

$$\ddot{X}_t + \frac{r+1}{t}\dot{X}_t + \frac{\beta}{\gamma}\sqrt{s}\nabla^2 f(X_t)\dot{X}_t + \left(1 + \frac{(r+1)\sqrt{s}}{2t}\right)\nabla f(X_t) = 0, \tag{31}$$

with initial conditions $X(t_0) = x_0$ and $\dot{X}(t_0) = -\sqrt{s}\nabla f(x_0)$. When $f \in \cup_{L>0}\mathcal{F}_L^2$ and $r = 2$, Shi et al. (2022) established the existence and uniqueness of a solution $X_t \in C^2([t_0, \infty); \mathbb{R}^n)$. The general case with $r \geq 2$ is beyond the scope of this paper. If a solution exists, the following result gives convergence bounds of (31) for both $r > 2$ and $r = 2$ via a new Lyapunov analysis. Shi et al. (2022) provided similar bounds only explicitly in the case of $r > 2$.

**Proposition 5.** *Let $f \colon \mathbb{R}^n \to \mathbb{R}$ be a function in $\mathcal{F}^2$. Suppose that $X_t$ is a solution to (31) with initial conditions $X(t_0) = x_0$ and $\dot{X}(t_0) = -\sqrt{s}\nabla f(x_0)$. If $r \geq 2$ and $\frac{\beta}{\gamma} > 0$, then there exists a time point $t_1 \geq t_0$,[13] with $t_1/\sqrt{s}$ depending only on $r$ and $\beta/\gamma$, such that $X_t$ satisfies the bounds:*

$$f(X_t) - f^\star = O\left(\frac{V_{t_1}}{t^2}\right), \quad \inf_{t_1 \leq u \leq t}\|\nabla f(X_u)\|^2 = O\left(\frac{\gamma V_{t_1}}{\beta\sqrt{s}(t^3 - t_1^3)}\right).$$

*In addition, if $r \geq 2$ and $\beta = 0$, then the first bound still holds. Here $V_{t_1}$ is the value that the continuous Lyapunov function takes at $t_1$: $V_t = (t + C\sqrt{s})(t + (\frac{r+1}{2} - \frac{\beta}{\gamma})\sqrt{s})(f(X_t) - f^\star) + \frac{t+C\sqrt{s}}{t}\frac{1}{2}\|r(X_t - x^\star) + t(\dot{X}_t + \frac{\beta\sqrt{s}}{\gamma}\nabla f(X_t))\|^2$, where $C = 0$ if $r > 2$ and $C = \frac{\beta}{\gamma}$ if $r = 2$.*

If we replace $t$ by $t_k = t_0 + k\sqrt{s} \sim k\sqrt{s}$, then the first bound above translates into the accelerated bound $O(1/(sk^2))$ on the objective gap as in (16) and the second becomes $O(1/(s^2 k^3))$ on the squared gradient norm as in (18). These two bounds above are exactly the continuous analogs of the associated bounds for algorithm (29). However, compared with the condition $\beta/\gamma > 1/2$ in Theorem 4, the condition in the continuous ODE setting is much weaker: $\beta/\gamma$ is allowed to be arbitrarily small, and even 0 (no Hessian term) if only the $O(1/t^2)$ rate for the objective gap is desired. The latter case corresponds to a discrete algorithm without the gradient correction, similar to HB (9). Therefore, even when a Hessian term is included as a continuous counterpart of the gradient correction, the convergence properties of the high-resolution ODE are currently not fully matched by those of algorithm (29). This remains to be further studied as an open question, as also mentioned in Shi et al. (2022), Section 5.1.

## 5 Hamiltonian-assisted interpretation

As observed in Section 4, the low-resolution and high-resolution ODEs do not fully capture the convergence properties of the algorithms studied. Alternatively, we directly formulate a broad class of discrete algorithms, HAG, based on a Hamiltonian function with a position variable and a momentum variable, and demonstrate that the conditions from our convergence results in Sections 2 and 3 can be interpreted through HAG in a unified manner in both the strongly convex and convex settings. This development is motivated by a related formulation of Hamiltonian-assisted Metropolis sampling (HAMS) (Song & Tan, 2023; 2022).

---

[13]See the proof of Proposition 5 in Appendix E.4 for an explicit expression for $t_1$.

### 5.1 HAG: Hamiltonian-assisted gradient method

For $f \in \mathcal{F}_L^1$, consider the following unconstrained minimization problem:

$$\min_{x,u \in \mathbb{R}^n} H(x,u) = f(x) + \frac{1}{2}\|u\|^2,$$

where $H(x,u)$ can be interpreted as a Hamiltonian function (or total energy), with $x$ a position variable and $u$ a momentum variable. The above problem is equivalent to the original problem (1). If $x$ and $u$ are updated separately, then there is no possible improvement compared with solving (1) directly. However, we show how $x$ and $u$ can be updated in a coupled manner to derive a rich class of first-order algorithms, which include representative algorithms from the classes (14) and (21) studied in Sections 2 and 3.

**Gradient descent with a linear constraint.** Given initial points $(x_0, u_0)$, consider minimizing (or decreasing) $H(x,u)$ subject to a linear constraint $x - \delta u = x_0 - \delta u_0$ for some $\delta > 0$. By substituting $u = \frac{x - x_0 + \delta u_0}{\delta}$, the problem with respect to $x$ becomes minimizing (or decreasing) $\tilde{f}(x; x_0, u_0) = f(x) + \frac{1}{2\delta^2}\|x - x_0 + \delta u_0\|^2$. Consider updating $x_0$ by gradient descent with a stepsize $s = \frac{1}{1 + \frac{1}{\delta^2}} = \frac{\delta^2}{1 + \delta^2}$, which, by the standard characterization of gradient descent, is equivalent to minimizing a quadratic surrogate function of $\tilde{f}$ at $x_0$ (with surrogate smoothness parameter being 1 for $f$) as follows:

$$x_1 = \arg\min_x \langle \nabla f(x_0), x - x_0 \rangle + \frac{1}{2}\|x - x_0\|^2 + \frac{1}{2\delta^2}\|x - x_0 + \delta u_0\|^2.$$

Then $u_0$ is updated by solving $x_1 - \delta u_1 = x_0 - \delta u_0$. The resulting update is

$$x_1 = x_0 - \frac{\delta^2}{1 + \delta^2}\nabla f(x_0) - \frac{\delta}{1 + \delta^2}u_0,$$

$$u_1 = u_0 - \frac{\delta}{1 + \delta^2}\nabla f(x_0) - \frac{1}{1 + \delta^2}u_0.$$

The function $\tilde{f}(x; x_0, u_0)$ has a smoothness parameter $L_{\tilde{f}} = L + \frac{1}{\delta^2}$. If $0 < \delta^2 \leq \frac{1}{L}$, then $L_{\tilde{f}} \leq \frac{2}{\delta^2} < \frac{2}{s}$ and $s < \frac{2}{L_{\tilde{f}}}$, so that the above GD update is well-behaved and $\tilde{f}$ (as well as $H$) is non-increasing (see the footnote 7).

**Momentum negation and extrapolation.** The above update can be repeated on $(x_1, u_1)$, but the constraint $x - \delta u = x_1 - \delta u_1 = x_0 - \delta u_0$ will remain the same, so that all subsequent updates also satisfy the same constraint and, in general, cannot converge to a minimizer of $H(x,u)$. To resolve this issue, we negate $u$ at the beginning of each iteration. In other words, given $(x_0, u_0)$, we first negate $u_0$ to $-u_0$, and then implement the above update. The negation of $u$ does not change the value of $H$, but keeps the constraint changing from iteration to iteration.

An illustration is provided in Figure 5. For 1-dim $x$, the linear constraint becomes a straight line in $\mathbb{R}^2$. When $f(x) = \frac{x^2}{2}$, the level sets of $H(x,u) = \frac{x^2}{2} + \frac{u^2}{2}$ are circles, and the gradient descent update is exactly the minimizer along the line. Then the update without negation gets stuck after the first iteration. Negating the momentum changes the linear constraint, and then the updates keep moving toward the minimizer $(0,0)$. When $f(x) = x^2$, the level sets of $H(x,u) = x^2 + \frac{u^2}{2}$ are ellipses, and the gradient descent update is not exactly the minimizer along the line. Nevertheless, without negation, all the iterates will stay on the same line, hence failing to converge to the minimizer $(0,0)$. The momentum negation is also essential to the HAMS algorithm and related under-damped Langevin sampling algorithms, where a negation of the momentum is required to achieve generalized reversibility (Neal, 2011; Song & Tan, 2022).

In addition, we introduce an extrapolation step before the negation of $u_0$ for $\rho \geq 0$. This leads to the following update given the initial points $(x_0, u_0)$:

$$x_1 = x_0 - \frac{(1 + \rho)\delta^2}{1 + \delta^2}\nabla f(x_0) + \frac{(1 + \rho)\delta}{1 + \delta^2}u_0,$$

$$u_1 = -u_0 - \frac{(1 + \rho)\delta}{1 + \delta^2}\nabla f(x_0) + \frac{1 + \rho}{1 + \delta^2}u_0.$$

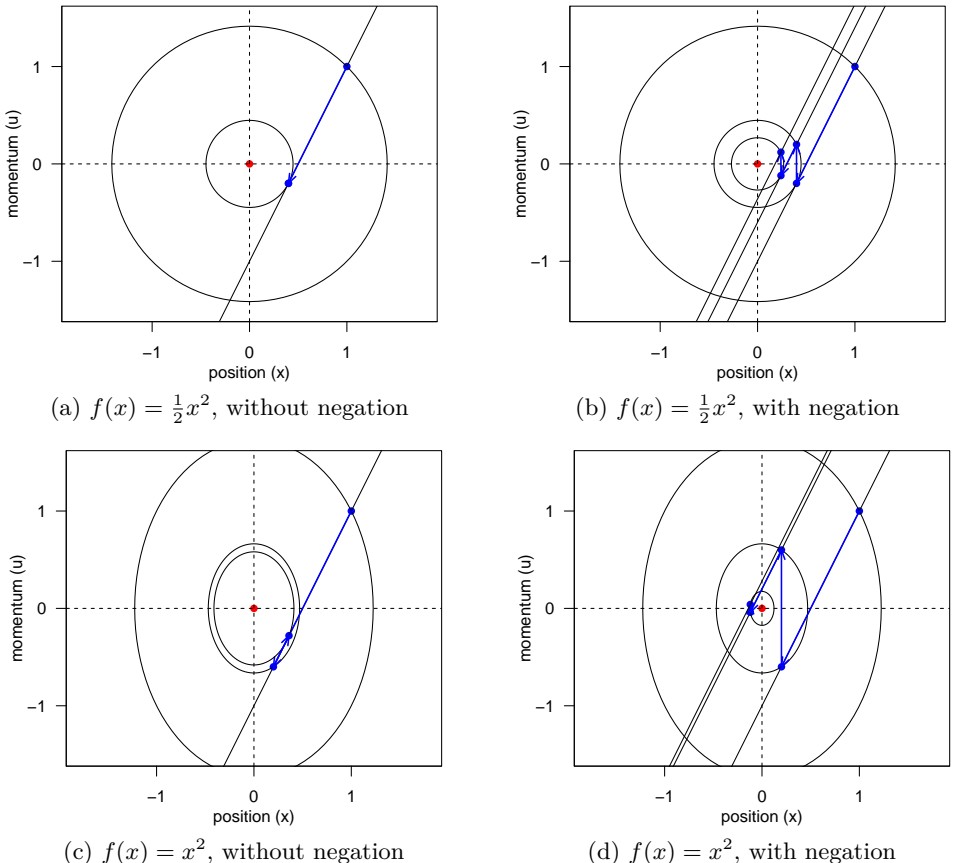

Figure 5: An illustration of the gradient descent with a linear constraint and the effect of momentum negation when $x \in \mathbb{R}$. The first two iterates are plotted without or with negation, for $f(x) = x^2/2$ in the first row and $f(x) = x^2$ in the second row. The initial point is $(1, 1)$ and the global minimizer is $(0, 0)$. We fix $\delta = 1/2$. The black solid circles and ellipses correspond to different level sets of the Hamiltonian functions.

By setting $a = \frac{(1+\rho)\delta^2}{1+\delta^2}$ and $b = \frac{1+\rho}{1+\delta^2}$, the above can be put in a clean form as

$$x_1 = x_0 - a\nabla f(x_0) + \sqrt{ab}\, u_0,$$
$$u_1 = -u_0 - \sqrt{ab}\nabla f(x_0) + bu_0.$$

The parameters $(\delta, \rho)$ can be determined from $(a, b)$ as $\delta = \sqrt{a/b}$ and $\rho = a + b - 1$.

**Gradient correction.** The above updates of $x_1$ and $u_1$ are parallel, so that $u_1$ uses only the gradient information at $x_0$, but not at the newly updated $x_1$. To exploit the gradient information at $x_1$, we incorporate a gradient correction into the update:

$$x_1 = x_0 - a\nabla f(x_0) + \sqrt{ab}\, u_0,$$
$$u_1 = -u_0 - \sqrt{ab}\nabla f(x_0) + bu_0 - \phi(\nabla f(x_1) - \nabla f(x_0)),$$

for some scalar $\phi$. Lastly, we allow $a$, $b$ and $\phi$ to vary from iteration to iteration and consider the following class of algorithms, called Hamiltonian-assisted gradient method (HAG): for $k \geq 0$,

$$x_{k+1} = x_k - a_k\nabla f(x_k) + \sqrt{a_k b_k}\, u_k,$$
$$u_{k+1} = -u_k - \sqrt{a_k b_k}\nabla f(x_k) + b_k u_k - \phi_k(\nabla f(x_{k+1}) - \nabla f(x_k)),$$
(32)

starting from $(x_0, u_0)$. Notably, the overall contribution from gradients remains unchanged in the update for $u_{k+1}$. Instead, they are re-weighted:

$$u_{k+1} = (b_k - 1)u_k - \underbrace{\left((\sqrt{a_k b_k} - \phi_k)\nabla f(x_k) + \phi_k \nabla f(x_{k+1})\right)}_{\text{re-weighted gradient}}. \tag{33}$$

The larger $\phi_k$ is, the larger weight is assigned to the new gradient $\nabla f(x_{k+1})$.

To facilitate comparison with algorithms in Sections 2 and 3, the HAG algorithm (32) can be put into a single-variable form involving $\{x_k\}$ only (see Appendix F): for $k \geq 1$,

$$x_{k+1} = x_k - \left(a_{k-1}\sqrt{\frac{a_k b_k}{a_{k-1} b_{k-1}}} + a_k\right)\nabla f(x_k) + (b_{k-1} - 1)\sqrt{\frac{a_k b_k}{a_{k-1} b_{k-1}}}(x_k - x_{k-1})$$

$$- \left(\phi_{k-1}\sqrt{a_k b_k} - a_{k-1}\sqrt{\frac{a_k b_k}{a_{k-1} b_{k-1}}}\right)(\nabla f(x_k) - \nabla f(x_{k-1})), \quad (34)$$

starting from $x_0$ and $x_1 = x_0 - a_0 \nabla f(x_0) + \sqrt{a_0 b_0} u_0$. The parameter $\{\phi_k\}$ only appears in the gradient correction in (34), and in the case of $a_k b_k$ being constant in $k$, the parameters $\{a_k\}$ and $\{b_k\}$ fully determine the gradient descent and the momentum terms respectively. For convenience, the term of $x_k - x_{k-1}$ is still referred to as the momentum term in single-variable forms, which is not to be confused with the momentum variable $u_k$.

## 5.2 Interpretation from HAG

HAG (32) or (34) are general enough to represent various algorithms studied in Sections 2 and 3, by choosing the parameters $\{a_k\}$, $\{b_k\}$ and $\{\phi_k\}$ accordingly. We examine our convergence results from the HAG perspective and obtain unified interpretations of sufficient conditions for achieving acceleration in both strongly convex and convex settings. Our findings are summarized as follows.

- The parameter $a_k$ acts as a re-scaled stepsize $s$. The parameter $b_k$, which controls the momentum term in the single-variable form, has a leading constant 2, and the gap $2 - b_k$ is $\Omega(\sqrt{q})$ with $q = \mu s$ or $\Omega(\frac{1}{k})$ in the strongly convex or convex setting, respectively.[14] In terms of $(\delta_k, \rho_k)$ in the HAG derivation, this indicates that $\delta_k \sim \sqrt{\frac{a_k}{2}}$ and $1 - \rho_k \sim 2 - b_k$. Therefore, $\delta_k$ is of order $\sqrt{s}$ (comparable to $\Delta t$ in deriving ODEs for NAG-SC and NAG-C), $\rho_k$ has a leading constant 1 (indicating symmetric extrapolation), and the gap $1 - \rho_k$ is $\Omega(\sqrt{q})$ or $\Omega(\frac{1}{k})$ in the strongly convex or convex setting.

- The parameter $\phi_k$ is of order $\sqrt{s}$ and controls the gradient correction, such that in the re-weighted gradient for the momentum update (33), the new gradient $\nabla f(x_{k+1})$ *fully dominates* with its weight greater than the total weight, whereas the old gradient $\nabla f(x_k)$ has a negative weight. The boundary case of a zero weight for $\nabla f(x_k)$ is also allowed in the strongly convex setting. Such heuristic interpretations are not feasible from the single-variable forms.

Currently such interpretations are derived from convergence results for classes of algorithms (14) and (22). It remains an interesting question to analyze HAG directly and study further implications.

**Strongly convex setting.** For $f \in \mathcal{S}_{\mu,L}^1$, we set

$$a_k \equiv a = \left(\frac{c_0}{2} + O(\sqrt{q})\right)s, \quad b_k \equiv b = 2 - c_1\sqrt{q} + O(q), \quad \phi_k \equiv \phi = (c_2 + O(\sqrt{q}))\sqrt{s},$$

where $c_0, c_1, c_2 > 0$. Then HAG (34) reduces to (14) exactly:

$$\begin{aligned} x_{k+1} &= x_k - 2a\nabla f(x_k) + (b-1)(x_k - x_{k-1}) - (\phi\sqrt{ab} - a)(\nabla f(x_k) - \nabla f(x_{k-1})) \\ &= x_k - (c_0 + O(\sqrt{q}))s\nabla f(x_k) + (1 - c_1\sqrt{q} + O(q))(x_k - x_{k-1}) \\ &\quad - (c_2\sqrt{c_0} - \frac{c_0}{2} + O(\sqrt{q}))s(\nabla f(x_k) - \nabla f(x_{k-1})). \end{aligned} \tag{35}$$

---

[14]The two orders, $\sqrt{q}$ and $\frac{1}{k}$, can be seen to match each other, because the accelerated algorithms and their continuous limits are linked via $t = k\sqrt{s}$ in Section 4 and hence $\sqrt{s}$ can be viewed to be in the order of $\frac{1}{k}$, as $s \to 0$ or $k \to \infty$.

In the following, we interpret the conditions in Corollary 1 through HAG.

The parameter $a = (\frac{c_0}{2} + O(\sqrt{q}))s$ plays the role of a re-scaled stepsize $s$. The parameter $b = 2 - c_1\sqrt{q} + O(q)$ controls the momentum term $(x_k - x_{k-1})$ in (35). The leading constant of $b$ is 2, which ensures the existence of a limiting ODE as $s \to 0$. In terms of $c_1$, Corollary 1 requires the condition $c_1^2 > 4c_0$, which as mentioned earlier may be potentially relaxed, for example, to $c_1 > 0$. This would lead to a condition $2 - b = \Omega(\sqrt{q})$ as $s \to 0$. In terms of $(\delta, \rho)$ in the HAG derivation, the preceding discussion also gives

$$\delta = \sqrt{a/b} = \frac{1}{2}\sqrt{c_0 s} + O(s), \quad 1 - \rho = 2 - b - a = \Omega(\sqrt{q}). \tag{36}$$

We observe that $\delta$ is in the order of $\sqrt{s}$, similarly as $\Delta t$ in deriving the ODEs (25) and (26), and $\rho$ is close to 1 (symmetric extrapolation) but with a gap being $\Omega(\sqrt{\mu s})$.

The parameter $\phi = (c_2 + O(\sqrt{q}))\sqrt{s}$ controls the gradient correction $(\nabla f(x_{k+1}) - \nabla f(x_k))$ in (35). The re-weighted gradient in the momentum update (33) for $u_{k+1}$ becomes

$$(\sqrt{c_0} - c_2 + O(\sqrt{q}))\sqrt{s}\nabla f(x_k) + (c_2 + O(\sqrt{q}))\sqrt{s}\nabla f(x_{k+1}). \tag{37}$$

For accelerated convergence in Corollary 1, the condition $c_2^2 \geq c_0$ indicates that the new gradient $\nabla f(x_{k+1})$ fully dominates with its weight, $c_2\sqrt{s}$, no smaller than the total weight $\sqrt{c_0 s}$, whereas the old gradient $\nabla f(x_k)$ has a zero or negative weight. For non-accelerated (sub-optimal) convergence in Corollary 1, the condition $c_0/4 \leq c_2^2 < c_0$ indicates that both the new and old gradients have positive weights, but $\nabla f(x_{k+1})$ contributes no less than $\nabla f(x_k)$. Interestingly, such an interpretation cannot be obtained from the single-variable form (14) or (35), even though a similar re-weighted gradient can also be identified there.[15]

**Convex setting.** For $f \in \mathcal{F}_L^1$, we set

$$b_k = 1 + \sigma_{k+2}, \quad a_k = \frac{c_0}{b_k}s, \quad \phi_k \equiv c_2\sqrt{s}, \tag{38}$$

where $c_0, c_2 > 0$ and, as before, $\sigma_{k+1} = \frac{\alpha_k - 1}{\alpha_{k+1}}$ for some sequence $\{\alpha_k\}$. Then HAG (34) reduces to

$$x_{k+1} = x_k - c_0\left(\frac{1}{b_{k-1}} + \frac{1}{b_k}\right)s\nabla f(x_k) + \sigma_{k+1}(x_k - x_{k-1}) - \left(c_2\sqrt{c_0} - \frac{c_0}{b_{k-1}}\right)s(\nabla f(x_k) - \nabla f(x_{k-1})), \tag{39}$$

which falls in the class (21) with $\gamma_k$ and $\beta_k$ varying in $k$, whose convergence property can be readily deduced from Theorem 4. We choose $a_k$ varying in $k$ such that $a_k b_k$ is constant in $k$, mainly to simplify the coefficients from (34).

Given any $\{\alpha_k\}$ satisfying $\alpha_k = \Omega(k)$ and $\frac{\alpha_{k+1}}{\alpha_k} = 1 + O(\frac{1}{k})$, we have $\sigma_{k+1} = \frac{\alpha_k - 1}{\alpha_{k+1}} = 1 - O(\frac{1}{k})$, $b_k = 1 + \sigma_{k+2} = 2 - O(\frac{1}{k})$, $a_k = \frac{c_0}{b_k}s = (c_0 + O(\frac{1}{k}))s/2$, and the preceding HAG algorithm (39) is further simplified to

$$x_{k+1} = x_k - \left(c_0 + O(\frac{1}{k})\right)s\nabla f(x_k) + \sigma_{k+1}(x_k - x_{k-1}) - \left(c_2\sqrt{c_0} - \frac{c_0}{2} - O(\frac{1}{k})\right)s(\nabla f(x_k) - \nabla f(x_{k-1})), \tag{40}$$

which resembles algorithm (35) in the strongly convex setting, in all the leading constants involved. Algorithm (40) can be put into (21) with $\gamma_k \to \gamma = c_0$ and $\beta_k \to \beta = c_2\sqrt{c_0} - \frac{c_0}{2}$. Note that (39) or (40) is more complex

---

[15]In (14) or (35), it seems reasonable to consider the re-weighted gradient as

$$(c_0 + O(\sqrt{q}))s\nabla f(x_k) + (c_2\sqrt{c_0} - \frac{c_0}{2} + O(\sqrt{q}))s(\nabla f(x_k) - \nabla f(x_{k-1}))$$

$$= (c_2\sqrt{c_0} + \frac{c_0}{2} + O(\sqrt{q}))s\nabla f(x_k) + (-c_2\sqrt{c_0} + \frac{c_0}{2} + O(\sqrt{q}))s\nabla f(x_{k-1}).$$

However, the condition $c_2^2 \geq c_0$ indicates that the weight of $\nabla f(x_k)$ is no smaller than $\frac{3}{2}c_0$, whereas the weight of $\nabla f(x_{k-1})$ is no larger than $-\frac{1}{2}c_0$, with a total weight $c_0 s$. Therefore, the conditions bear no meaningful interpretations.

than (29) with constant $\gamma_k$ and $\beta_k$ studied in Shi et al. (2022). In the following, we interpret conditions (i) and (ii) in Theorem 4 through HAG for the choice $\alpha_k = \frac{k+r}{r}$. The technical monotonicity condition (iii) can be verified for this specific choice (see Appendix F for a proof).

Similarly to the strongly convex setting, the parameter $a_k = (c_0 + O(\frac{1}{k}))s/2$ plays the role of a re-scaled stepsize $s$. Moreover, the parameter $b_k = 1 + \sigma_{k+2}$ controls the momentum term $(x_k - x_{k-1})$ in (40), and has its leading constant being 2, which is necessary (but not sufficient by Lemma 1) for a limiting ODE to exist as $s \to 0$. For the choice $\alpha_k = \frac{k+r}{r}$ and $\sigma_{k+1} = \frac{k}{k+r+1}$, the recursive condition $\alpha_{k+1}(\alpha_{k+1} - 1) \le \alpha_k^2$ is equivalent to $r \ge 2$. Then it implies a condition $2 - b_k = (r+1)/k - O(1/k^2) = \Omega(1/k)$ as $k \to \infty$ (or $s \sim 1/k^2 \to 0$). In terms of the time-dependent $(\delta_k, \rho_k)$ in HAG, the preceding discussion gives

$$\delta_k = \sqrt{a_k/b_k} = \frac{1}{2}\sqrt{c_0 s} + O(s), \quad 1 - \rho_k = 2 - b_k - a_k = \Omega\left(\frac{1}{k}\right). \tag{41}$$

We observe that $\delta_k$ is in the order of $\sqrt{s}$, similarly as $\Delta t$ in deriving the ODE (24), and $\rho_k$ is close to 1 (symmetric extrapolation) but with a gap being $\Omega(\frac{1}{k})$. The conditions, (36) and (41), for the strongly convex and convex settings respectively, are of similar forms, with $\sqrt{\mu s} (= \sqrt{q})$ and $\frac{1}{k}$ exchanged with each other.

The parameter $\phi_k = c_2 \sqrt{s}$ plays a similar role of controlling the gradient correction term $(\nabla f(x_k) - \nabla f(x_{k-1}))$ in (40), as does $\phi = (c_2 + O(\sqrt{q}))\sqrt{s}$ in the strongly convex setting. The re-weighted gradient in the momentum update (33) for $u_{k+1}$ is similar to (37):

$$(\sqrt{c_0} - c_2)\sqrt{s}\nabla f(x_k) + c_2\sqrt{s}\nabla f(x_{k+1}).$$

With $\gamma = c_0$ and $\beta = c_2\sqrt{c_0} - \frac{c_0}{2}$, the condition $\beta > \gamma/2$ in Theorem 4 reduces to $c_2^2 > c_0$, which indicates that the new gradient $\nabla f(x_{k+1})$ fully dominates with its weight, $c_2\sqrt{s}$, greater than the total weight $\sqrt{c_0 s}$, whereas the old gradient $\nabla f(x_k)$ has a negative weight. This is the same as the condition, $c_2^2 \ge c_0$, for accelerated convergence in the strongly convex setting except that the boundary case $c_2^2 = c_0$ is excluded here. As discussed earlier, the single-variable form (40) does not admit a meaningful interpretation for the re-weighted gradient.

## 6  Outlines of Lyapunov analyses

We outline our Lyapunov analyses to prove the convergence results for the discrete algorithms in Sections 2 and 3. See Appendix E for our Lyapunov analyses for the convergence of ODEs in Section 4. Compared with existing ones, our Lyapunov functions are constructed to handle more general algorithms and ODEs or to achieve more concise analysis and sometimes sharper results. Before giving the outlines of our analyses, we summarize the comparison of Lyapunov analyses. The comparison is restricted among analytic Lyapunov functions, excluding those constructed by the LMI/SDP framework, which require additional numerical solvers (Lessard et al., 2016; Fazlyab et al., 2018; Taylor et al., 2018; Taylor & Bach, 2019; Sanz Serna & Zygalakis, 2021; Dobson et al., 2025).

**Strongly convex setting.** We construct the discrete Lyapunov (43) to establish the convergence of algorithm (10) including NAG-SC and TMM as special cases. The auxiliary-energy term $\mu\|z_{k+1} - x^\star\|^2/2$ in (43) is also used in Bansal & Gupta (2019) for NAG-SC and in d'Aspremont et al. (2021) for TMM. However, the potential-energy term in our Lyapunov is in $\{x_k\}$ whereas the one in Bansal & Gupta (2019) is in $\{y_k\}$. The potential-energy term of the Lyapunov in d'Aspremont et al. (2021) is in $\{x_k\}$ like ours, but is not lower-bounded by $f(x_k) - f^\star$, so that their analysis only establishes the convergence for $\{z_k\}$. In addition, compared with the analysis of NAG-SC in Shi et al. (2022), our Lyapunov function (43) has fewer terms, and our analysis is much more concise.

Our continuous Lyapunov function to analyze the class of low-resolution ODEs (26) for Proposition 3 is extended from the one proposed in Wilson et al. (2021) for $(c_0, c_1) = (1, 2)$ (i.e., the low-resolution ODE of NAG-SC). Furthermore, we construct suitable Lyapunov functions to analyze the high-resolution ODEs of NAG-SC and HB. Compared with the ones used in Shi et al. (2022), our Lyapunov functions lead to a sharper convergence bound for NAG-SC and HB (Proposition 4).

**Convex setting.** We construct the discrete Lyapunov (46) to establish the convergence of (20) or equivalently (22), including those in Beck (2017) and Shi et al. (2022) as special cases. The auxiliary-energy term $\|z_{k+1} - x^\star\|^2/2$ in (46) is motivated by Su et al. (2016) for NAG-C and Shi et al. (2022) for (29), a sub-class of (20) with $\gamma_k = 1$, $\beta_k = \beta/\gamma$, and $\alpha_k = (k+r)/r$ for $r \geq 2$. However, the potential-energy term in our Lyapunov (46) differs from the related ones in Su et al. (2016) and Shi et al. (2022). Moreover, Shi et al. (2022) analyzed the three cases, $r = 2$ and $\beta \leq 1$, $r = 2$ and $\beta > 1$, and $r > 2$, separately. It seems difficult to extend their case-by-case analysis to cover the more general results in our Theorem 4.

The continuous Lyapunov function for the class of high-resolution ODEs (31) is the same as in Shi et al. (2022) when $r > 2$, but involves a technical modification when $r = 2$. The modification helps to establish the convergence bounds in Proposition 5 for both $r > 2$ and $r = 2$, whereas similar bounds are provided in Shi et al. (2022) only explicitly for $r > 2$.

## 6.1 Strongly convex setting

We provide a Lyapunov analysis to establish Theorems 1–3 for the class of algorithms (10). Proof details are presented in Appendix C. Although our Lyapunov function, like most existing ones, is manually designed, our analysis proceeds in several structured steps.

**Step 1. Bounding the differencing of an auxiliary energy.** The sequence $\{z_k\}$ plays a key role in our formulation of (10) in a three-variable form. We identify $\frac{\mu}{2}\|z_k - x^\star\|^2$ as an auxiliary-energy term and bound its differencing, $\frac{1}{1-\nu\sqrt{q}}\frac{\mu}{2}\|z_{k+1} - x^\star\|^2 - \frac{\mu}{2}\|z_k - x^\star\|^2$.

**Lemma 2.** *Let $f\colon \mathbb{R}^n \to \mathbb{R}$ be a function in $\mathcal{S}^1_{\mu,L}$. For any $s > 0$ such that $0 \leq \eta s \leq 1/L$, $\nu \geq 0$, $1 - \nu\sqrt{q} > 0$, $\tau > 0$ and $\zeta = 1 + (1 - \tau)\sqrt{q} \geq 0$, the iterates of (10) satisfy that for $k \geq 1$,*

$$\left(\frac{\zeta\nu}{\tau^2}(\tau + \zeta\eta\sqrt{q}) + \frac{\nu\sqrt{q}}{1 - \nu\sqrt{q}}\right)(f(x_k) - f^\star) - \frac{\nu s}{2}\left(\frac{\nu}{1 - \nu\sqrt{q}} - \frac{\zeta\eta}{\tau}\right)\|\nabla f(x_k)\|^2 + \frac{1}{1 - \nu\sqrt{q}}\frac{\mu}{2}\|z_{k+1} - x^\star\|^2$$

$$\leq \frac{\zeta\nu}{\tau^2}(\tau + \zeta\eta\sqrt{q})\left(f(x_{k-1}) - f^\star - \frac{\eta s}{2}\|\nabla f(x_{k-1})\|^2\right) + \frac{\mu}{2}\|z_k - x^\star\|^2. \quad (42)$$

**Step 2. Constructing a discrete Lyapunov function.** We define a Lyapunov function simply from the right-hand-side of (42): for $k \geq 0$,

$$V_{k+1} = \frac{\zeta\nu}{\tau^2}(\tau + \zeta\eta\sqrt{q})\left(f(x_k) - f^\star - \frac{\eta s}{2}\|\nabla f(x_k)\|^2\right) + \frac{\mu}{2}\|z_{k+1} - x^\star\|^2. \quad (43)$$

We refer to the first term above as a potential energy, involving both $f(x_k) - f^\star$ and $\|\nabla f(x_k)\|^2$, and the second term $\frac{\mu}{2}\|z_k - x^\star\|^2$ as an auxiliary energy. The negative term $-\frac{\eta s}{2}\|\nabla f(x_k)\|^2$ in (43) is a technical adjustment and can be traced to (42). A similar term can be found in our Lyapunov (46) in the convex setting. By Assumption 1 for smoothness of $\tilde{\eta}(\cdot)$ near 0, a sufficiently small $C_0 \in (0, \frac{1}{4\eta_0}]$ can be picked with $\eta_0 > 0$ such that when $0 < s \leq \frac{C_0}{L}$, we have $0 < q = \mu s \leq C_0$, $\eta = \tilde{\eta}(\sqrt{q}) \leq 2\eta_0$, and $\eta L s \leq 2\eta_0 C_0 \leq \frac{1}{2}$. Then for $0 < s \leq \frac{C_0}{L}$, by the $L$-smoothness of $f$, we have $f(x_k) - f^\star - \frac{\eta s}{2}\|\nabla f(x_k)\|^2 \geq (1 - \eta L s)(f(x_k) - f^\star) \geq \frac{1}{2}(f(x_k) - f^\star)$. More details are given in Appendix C.3. The negative term in (46) is handled similarly in Appendix D.3.

**Step 3. Identifying sufficient conditions for Lyapunov contraction.** As expected from (42), we further bound $V_{k+1} - (1 - \nu\sqrt{q})V_k$ and identify conditions such that a contraction inequality holds for the Lyapunov function: $V_{k+1} - (1 - \nu\sqrt{q})V_k \leq 0$.

**Lemma 3.** *Define $\mathbf{I}$ and $\mathbf{II}$ as polynomials of $\sqrt{q}$, $\eta$, $\nu$ and $\tau$ (hence functions of $\sqrt{q}$) taking the following forms: (recall that $\zeta = 1 + (1 - \tau)\sqrt{q}$)*

$$\begin{aligned}
\mathbf{I} &= \zeta\nu(\tau + \zeta\eta\sqrt{q}) - \tau^2 \\
&= \tau(\nu - \tau) + \nu(\eta - \tau(\tau - 1))\sqrt{q} - 2\eta\nu(\tau - 1)q + \eta\nu(\tau - 1)^2 q^{\frac{3}{2}}, \\
\mathbf{II} &= \tau(\nu\tau - 2\zeta\eta) + \zeta\eta(\nu\tau - \zeta\eta)\sqrt{q} \\
&= \tau(\nu\tau - 2\eta) + \eta(-\eta + \nu\tau + 2\tau(\tau - 1))\sqrt{q} + \eta(\tau - 1)(2\eta - \nu\tau)q - \eta^2(\tau - 1)^2 q^{\frac{3}{2}}.
\end{aligned} \quad (44)$$

*Under the condition in Lemma 2, if one of the following (mutually exclusive) conditions holds: (i) $\mathbf{I} > 0$ and $\mathbf{I} + \sqrt{q}\mathbf{II} \leq 0$, or (ii) $\mathbf{II} > 0$ and $(1/\kappa)\mathbf{I} + \sqrt{q}\mathbf{II} \leq 0$, or (iii) $\mathbf{I} \leq 0$ and $\mathbf{II} \leq 0$, then we have the contraction inequality $V_{k+1} \leq (1 - \nu\sqrt{q})V_k$ for $k \geq 1$.*

**Step 4. Verifying the contraction conditions and completing the analysis.** For completing the analysis, the final step is to show that the sufficient conditions for Lyapunov contraction in Lemma 3 are satisfied under the conditions included in Theorems 1–3. This step can be algebraically tedious, and the details are left to Appendix C.

The contraction inequality, if verified, directly leads to a convergence bound $O((1 - C_1\sqrt{\mu s})^k)$ as stated in (12). However, to fulfill the conditions in Lemma 3, we find that two ranges of stepsize $s$ (or $q$) are allowed: $0 < s \lesssim \mu/L^2$ (or $0 < q \lesssim 1/\kappa^2$) and $0 < s \lesssim 1/L$ (or $0 < q \lesssim 1/\kappa$). As discussed after Theorem 1, the upper bound of feasible $s$ determines whether the vanilla or accelerated convergence is achieved, in terms of the dependency on $\kappa$.

## 6.2 Convex setting

We provide a Lyapunov analysis to establish Theorem 4 for the class of algorithms (20) or equivalently (22). Proof details are presented in Appendix D. Our analysis proceeds in several structured steps, similarly as in the strongly convex setting (Section 6.1).

**Step 1. Bounding the differencing of an auxiliary energy.** From our formulation of the three-variable form (22), we identify $\frac{1}{2}\|z_k - x^\star\|^2$ as an auxiliary-energy term and bound its differencing, $\frac{1}{2}\|z_{k+1} - x^\star\|^2 - \frac{1}{2}\|z_k - x^\star\|^2$.

**Lemma 4.** *Let $f \colon \mathbb{R}^n \to \mathbb{R}$ be a function in $\mathcal{F}_L^1$. Then for each $k \geq 1$ such that $\alpha_k \geq 1$ and $\widetilde{\alpha}_k \geq 0$, the iterates of (22) satisfy*

$$\alpha_k \widetilde{\alpha}_k s(f(x_k) - f^\star) - \frac{\widetilde{\alpha}_k^2 s^2}{2}\|\nabla f(x_k)\|^2 + \frac{1}{2}\|z_{k+1} - x^\star\|^2$$
$$\leq \widetilde{\alpha}_k(\alpha_k - 1)s\left((f(x_{k-1}) - f^\star) - \frac{(2 - \beta_{k-1}Ls)\beta_{k-1}s}{2}\|\nabla f(x_{k-1})\|^2\right) + \frac{1}{2}\|z_k - x^\star\|^2. \quad (45)$$

**Step 2. Constructing a discrete Lyapunov function.** We define a Lyapunov function simply as the left-hand-side of (45) up to a scalar sequence $\{\omega_k\}$: for $k \geq 0$,

$$V_{k+1} = \omega_{k+1}\left(\alpha_k \widetilde{\alpha}_k s(f(x_k) - f^\star) - \frac{\widetilde{\alpha}_k^2 s^2}{2}\|\nabla f(x_k)\|^2 + \frac{1}{2}\|z_{k+1} - x^\star\|^2\right). \quad (46)$$

The sequence $\{\omega_k\}$ is introduced to later deal with the mismatching of coefficients on two sides of (45). Similarly as in Section 6.1, we refer to the term involving $f(x_k) - f^\star$ and $\|\nabla f(x_k)\|^2$ as a potential energy, and the term $\frac{1}{2}\|z_k - x^\star\|^2$ as an auxiliary energy.

**Step 3. Identifying sufficient conditions for Lyapunov contraction.** As expected from (45), we further bound $V_{k+1} - V_k$ and identify conditions such that a contraction inequality holds for the Lyapunov function: $V_{k+1} - V_k \leq 0$ or $V_{k+1} - V_k \lesssim -k^2 s^2 \|\nabla f(x_{k-1})\|^2$.

**Lemma 5.** *Define $\mathbf{I}$ and $\mathbf{II}$ as follows:*

$$\mathbf{I} = \omega_k \alpha_{k-1}\widetilde{\alpha}_{k-1} - \omega_{k+1}\widetilde{\alpha}_k(\alpha_k - 1),$$
$$\mathbf{II} = \omega_{k+1}\widetilde{\alpha}_k(\alpha_k - 1)\beta_{k-1}(2 - \beta_{k-1}Ls) - \omega_k\widetilde{\alpha}_{k-1}^2.$$

*For any $k \geq 1$ such that $\alpha_k \geq 1$, $\widetilde{\alpha}_k \geq 0$, $\omega_k \geq \omega_{k+1}$, $\mathbf{I} \geq 0$, and $\mathbf{II} \geq 0$, the Lyapunov function (46) satisfies $V_{k+1} \leq V_k$. If further $\mathbf{II} \geq Ck^2$ for a constant $C > 0$, then $V_{k+1} - V_k \leq -\frac{C}{2}k^2 s^2 \|\nabla f(x_{k-1})\|^2$.*

**Step 4. Verifying the contraction conditions and completing the analysis.** The final step is to show that the sufficient conditions for Lyapunov contraction in Lemma 5 are satisfied under the conditions included in Theorem 4. The details are left to Appendix D.

The contraction inequality $V_{k+1} - V_k \leq 0$ leads to the convergence bound (16) for the objective gap, whereas the contraction inequality $V_{k+1} - V_k \lesssim -k^2 s^2 \|\nabla f(x_{k-1})\|^2$ leads to the inverse cubic rate (18) for the squared gradient norm. Unlike in the strongly convex setting, the stepsize $s$ can be simply set in the range $0 < s \lesssim 1/L$ to fulfill the conditions in Lemma 5.

## 7 Conclusion

Our work contributes to understanding the acceleration of first-order algorithms for convex optimization. We directly formulate discrete algorithms as general as we can and establish sufficient conditions for accelerated convergence using discrete Lyapunov functions. We point out currently notable gaps between the convergence properties of the corresponding algorithms and ODEs. We propose the Hamiltonian-assisted gradient method, HAG, and demonstrate unified interpretations of our acceleration conditions, especially those for gradient correction. Future work is needed to address various open questions, including the extent to which our sufficient conditions are necessary, improving analyses using the LMI/SDP framework, and further understanding the gradient correction and its relationship with existing explanations, such as symplectic integrations.

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

# A  Comparison with the literature

## A.1  Comparison with the LMI/SDP techniques

We compare our methodology with the LMI/SDP techniques as in Lessard et al. (2016), Fazlyab et al. (2018), Taylor et al. (2018), Taylor & Bach (2019), and Upadhyaya et al. (2025).

- As mentioned in Section 1, the LMI/SDP techniques, including the PEP methods and the IQC methods, are powerful tools for constructing Lyapunov functions systematically. Due to an inherent optimization procedure, the LMI/SDP techniques generally produce sharper bounds for optimization algorithms. When applied to existing algorithms like NAG-SC and TMM for minimizing strongly convex objectives, the LMI/SDP techniques may produce a larger constant $C$ in the convergence bound $O((1 - C/\sqrt{\kappa})^k)$ than our hand-crafted Lyapunov functions in Section 6.1. Nonetheless, as mentioned in Section 2.2, our goal is not to find the sharpest convergence bound. Instead, all algorithms with a bound $O((1 - C/\sqrt{\kappa})^k)$ for *any* $C > 0$ are considered to be accelerated, and are not further differentiated in our work.

- The LMI/SDP techniques may rely on numerical solvers to find the Lyapunov function leading to the sharpest bound, whose results are less interpretable. However, our goal is to understand the mechanism behind the acceleration phenomenon. To that end, we analyze general and representative classes of algorithms as in (10) and (20). Although our Lyapunov functions are manually constructed, they lead to concrete sufficient conditions on the gradient correction for acceleration that have interesting interpretations.

## A.2 Comparison with Sanz Serna & Zygalakis (2021)

Sanz Serna & Zygalakis (2021) leveraged the IQC methods in Fazlyab et al. (2018) to analyze a family of Nesterov optimization methods, which is a subclass of the algorithm (14). Although their analyses yield sharper bounds due to the control-theoretic framework, the subclass of algorithms has a fixed gradient correction term, and thus do not inform the effect of gradient correction on acceleration.

To be more specific, Sanz Serna & Zygalakis (2021) studied algorithm (3), whose single-variable form is (4). With a momentum coefficient $\sigma = 1 - c_1\sqrt{q} + R_2$, algorithm (3) is a special case of (14) with $c_0 = 1$, $R_1 = 0$, $c_2 = 3/2$, and $R_3 = -c_1\sqrt{q} + R_2$. Notice that the leading coefficient of gradient correction is fixed at $c_2\sqrt{c_0} - c_0/2 = 1$. Sanz Serna & Zygalakis (2021) shows that, for $0 < s \leq \frac{1}{L}$, the iterates of (3) or (4) satisfy the convergence bound $f(x_k) - f^\star = O\left((1 - C_1\sqrt{\mu s})^k\right)$ for $0 < C_1 \leq 1$ as long as $\sigma(q)$ is bounded by

$$\sigma_-(q, C_1) \leq \sigma(q) = 1 - c_1\sqrt{q} + R_2 \leq \sigma_+(q, C_1),$$

where

$$\sigma_\pm(q, C_1) = 1 - \frac{1 \mp \sqrt{1 - C_1^2} + C_1^2}{C_1}\sqrt{q} + O(q).$$

As $C_1 \to 0$, the linear term coefficients $\frac{1 \mp \sqrt{1 - C_1^2} + C_1^2}{C_1}$ goes to 0 and $\infty$ respectively, which implies that $\sigma(q) = 1 - c_1\sqrt{q} + R_2$ with any $c_1 > 0$ is always bounded by $\sigma_-$ and $\sigma_+$ for some $C_1 > 0$ and $q$ small enough. Therefore, for any $c_1 > 0$, the subclass (3) or (4) achieves an accelerated convergence rate $O\left((1 - O(1/\sqrt{\kappa}))^k\right)$, which is weaker than our condition $c_1^2 > 4c_0$.

The subclass (4), however, uses a fixed gradient correction so that its convergence rate does not inform the impact of gradient correction on acceleration. The later work by Dobson et al. (2025) established sharper results on $C_1$ under more complicated conditions on the momentum coefficient $\sigma$. However, because the same class of algorithm (3) or (4) is studied, a sharper constant $C_1$ does not lead to qualitatively broader implications on acceleration.

In terms of the ODE part, the limiting ODE of (4) under $\sigma = 1 - c_1\sqrt{q} + R_2$ is

$$\ddot{X}_t + c_1\sqrt{\mu}\dot{X}_t + \nabla f(X_t) = 0,$$

which is the same as equation (26) with $c_0 = 1$. Interestingly, if $c_1$ depends on $s$, then the algorithm (4) with an accelerated convergence may not have a limiting ODE as $c_1$ is allowed to oscillate between $\left(\frac{1 - \sqrt{1 - C_1^2} + C_1^2}{C_1}, \frac{1 + \sqrt{1 - C_1^2} + C_1^2}{C_1}\right)$. A similar observation is made in the convex case. See Lemma 1 for details. When $c_1$ is constant, Sanz Serna & Zygalakis (2021) used the IQC methods to prove that $f(X_t) - f^\star = O(\exp\{-C\sqrt{\mu}t(f(x_0) - f^\star)\})$ for a sharper constant $C$ than that in our Proposition 3. If $f \in \mathcal{S}^1_{\mu,L}$ with a bounded Lipschitz constant $L$, Dobson et al. (2025) shows that the constant $C$ can be further improved.

Unlike Sanz Serna & Zygalakis (2021) and Dobson et al. (2025), our goal is not to find the sharpest constant in convergence bounds. Instead, we would like to characterize algorithms with accelerated convergence. Nonetheless, the IQC methods remain powerful tools that can be leveraged to refine our analysis on gradient correction in future work.

## A.3 Comparison with Shi et al. (2022)

We compare our results with those in Shi et al. (2022) below.

- In the strongly convex setting, Shi et al. (2022) leveraged the high-resolution ODEs to establish an accelerated convergence for NAG-SC and a non-accelerated convergence for HB. Their analysis links the acceleration to the gradient correction term, which is present in NAG-SC but absent in HB. However, it remains unclear in what scope the gradient correction leads to acceleration. Our work bridges the gap between HB's non-acceleration and NAG-SC's acceleration by showing that as long as the gradient correction coefficient surpasses an explicit threshold, the acceleration is guaranteed. Remarkably, the same condition is also observed for the convex setting. Therefore, for the first time, our analysis establishes a unified condition for acceleration in strongly convex and convex settings.

- In the convex (not necessarily strongly convex) setting, our results are mainly technical refinements of those in Shi et al. (2022). The class of algorithms in (20) is more general than the one studied in Shi et al. (2022). See Table 2 for details.

- Another question left by Shi et al. (2022) is whether the high-resolution ODE that explicitly retains the gradient correction can explain the difference between discrete algorithms. We show that the high-resolution ODE perspective, albeit remaining a useful analytic tool, fails to explain the difference between accelerated and non-accelerated algorithms (especially regarding the gradient correction term) in either the strongly convex or the convex settings.

To summarize, our work substantially extends Shi et al. (2022) in understanding the effect of gradient correction on acceleration.

## B Notation

The notation used in the main paper and the remaining appendices are summarized in Table 3.

## C Technical details in Section 2

### C.1 Reformulation of NAG-SC and TMM

First, we show that NAG-SC (3) with (7) can be reformulated as (8), i.e., (10) with $\eta = \nu = \tau = 1$. Apparently (3a) is the same as (8a). It suffices to derive (8b) and (8c) from (3). Because $x_0 = y_0 = z_0$, (7) also holds for $z_0$. Therefore, for $k \geq 0$ we have

$$z_k = \frac{1 + \sqrt{q}}{\sqrt{q}} x_k + \left(1 - \frac{1 + \sqrt{q}}{\sqrt{q}}\right) y_k,$$

which implies

$$x_k = \frac{\sqrt{q}}{1 + \sqrt{q}} z_k + \frac{1}{1 + \sqrt{q}} y_k, \tag{47}$$

and

$$y_k = (1 + \sqrt{q}) x_k - \sqrt{q} z_k. \tag{48}$$

Eq. (47) with $k$ replaced by $k + 1$ becomes (8c). Substituting $x_{k+1}$ from (47) into (3b), we have

$$\frac{\sqrt{q}}{1 + \sqrt{q}} z_{k+1} + \frac{1}{1 + \sqrt{q}} y_{k+1} = y_{k+1} + \frac{1 - \sqrt{q}}{1 + \sqrt{q}} (y_{k+1} - y_k),$$

which yields

$$z_{k+1} = \frac{1}{\sqrt{q}} y_{k+1} - \frac{1 - \sqrt{q}}{\sqrt{q}} y_k.$$

Substituting $y_{k+1}$ from (3a) into the above display, we have

$$z_{k+1} = \frac{1}{\sqrt{q}} (x_k - s\nabla f(x_k)) - \frac{1 - \sqrt{q}}{\sqrt{q}} y_k,$$

which together with (48) gives

$$z_{k+1} = \frac{1}{\sqrt{q}} (x_k - s\nabla f(x_k)) - \frac{1 - \sqrt{q}}{\sqrt{q}} ((1 + \sqrt{q}) x_k - \sqrt{q} z_k) = \sqrt{q} \left(x_k - \frac{1}{\mu} \nabla f(x_k)\right) + (1 - \sqrt{q}) z_k,$$

which is (8b).

Table 3: Notation

| Section | Notation | Meaning |
|---------|----------|---------|
| General | $f$ | Objective function to be minimized |
| | $x^\star$ | One of the minimizers of $f$ |
| | $f^\star$ | Minimum of $f$ |
| | $L$ | Smoothness parameter of $f$ |
| | $\mu$ | Strong convexity parameter of $f$ |
| | $\kappa$ | Condition number $L/\mu$ |
| | $k$ | Iteration number of gradient methods |
| | $s$ | Step size in discrete algorithms |
| | $\mathcal{F}^m$ | Convex and $m$ times continuously differentiable functions from $\mathbb{R}^n$ to $\mathbb{R}$ |
| | $\mathcal{F}_L^m$ | Functions in $\mathcal{F}^m$ that are additionally $L$-smooth |
| | $\mathcal{S}_\mu^m$ | Functions in $\mathcal{F}^m$ that are additionally $\mu$-strongly convex |
| | $\mathcal{S}_{\mu,L}^m$ | Functions in $\mathcal{F}_L^m \cap \mathcal{S}_\mu^m$ |
| Strongly convex | $q$ | Shorthand for $\mu s$ |
| | $\sigma$ | Momentum coefficient |
| | $\eta, \nu$, and $\tau$ | Three parameters in algorithms (10) and (11) that may depend on $q$ |
| | $\tilde{\eta}, \tilde{\nu}$, and $\tilde{\tau}$ | Non-negative, analytic scalar functions in Assumption 1 such that $\eta = \tilde{\eta}(\sqrt{q})$, $\nu = \tilde{\nu}(\sqrt{q})$, and $\tau = \tilde{\tau}(\sqrt{q})$ |
| | $\{\eta_i\}, \{\nu_i\}, \{\tau_i\}$ | Coefficients in the Taylor expansions (13) of $\eta$, $\nu$, $\tau$ in Assumption 1 |
| | $\zeta$ | Shorthand for $1 + (1-\tau)q$ |
| | $c_0, c_1$, and $c_2$ | Parameters in algorithm (14), consistent with the parameters in the limiting ODE (26) of (14) |
| Convex | $\sigma_{k+1}$ | Momentum coefficient (possibly varying in $k$) |
| | $\alpha_{k+1}$ | A scalar sequence for specifying momentum coefficient as $\sigma_{k+1} = (\alpha_k - 1)/\alpha_{k+1}$ |
| | $r$ | A scalar for specifying $\alpha_k = (k+r)/r$, consistent with the parameters in the limiting ODEs (30) and (31) of (21) |
| | $\beta_k$ and $\gamma_k$ | Two parameters in algorithms (20) and (21) that may vary with $k$ |
| | $\beta$ and $\gamma$ | Limits of $\beta_k$ and $\gamma_k$, respectively, consistent with the parameters in the limiting ODE (31) of (21) |
| ODE | $X_t$ | Solution $X(t)$ for ODEs |
| HAG | $u$ and $u_k$ | Momentum variables in the Hamiltonian function and the associated discrete algorithms, respectively |
| | $\delta$ and $\rho$ | The linear constraint parameter and the extrapolation parameter, respectively, in deriving the HAG method |
| | $a_k, b_k$, and $\phi_k$ | Three parameters in the HAG methods (32) and (34) |
| | $a, b$, and $\phi$ | Constant $a_k, b_k$, and $\phi_k$, respectively |

Second, we show that TMM (5), i.e., (10) with $\eta = \nu = 1$ and $\tau = 2$, can be reformulated as follows, with (49c) same as (6) for $z_{k+1}$:

$$y_{k+1} = x_k - s\nabla f(x_k), \tag{49a}$$

$$x_{k+1} = x_k - (2 - \sqrt{q})s\nabla f(x_k) + \frac{(1-\sqrt{q})^2}{1+\sqrt{q}}(y_{k+1} - y_k), \tag{49b}$$

$$z_{k+1} = \frac{1+\sqrt{q}}{2\sqrt{q}}x_{k+1} + \left(1 - \frac{1+\sqrt{q}}{2\sqrt{q}}\right)y_{k+1}. \tag{49c}$$

Apparently (5a) is the same as (49a). It suffices to derive (49b) and (49c). Solving (5c) for $z_{k+1}$ gives (49c). Because $x_0 = y_0 = z_0$, (49c) also holds for $z_0$. That is, for all $k \geq 0$,

$$z_k = \frac{1 + \sqrt{q}}{2\sqrt{q}} x_k + \left(1 - \frac{1 + \sqrt{q}}{2\sqrt{q}}\right) y_k,$$

which together with (5b) yields

$$z_{k+1} = \sqrt{q}\left(x_k - \frac{1}{\mu}\nabla f(x_k)\right) + (1 - \sqrt{q})\left[\frac{1 + \sqrt{q}}{2\sqrt{q}} x_k + \left(1 - \frac{1 + \sqrt{q}}{2\sqrt{q}}\right) y_k\right]$$

$$= \frac{1 + q}{2\sqrt{q}} x_k - \frac{\sqrt{q}}{\mu}\nabla f(x_k) - \frac{(1 - \sqrt{q})^2}{2\sqrt{q}} y_k.$$

To derive (49b), substituting the above display into (5c) and using (5a) yields

$$x_{k+1} = \frac{2\sqrt{q}}{1 + \sqrt{q}}\left(\frac{1 + q}{2\sqrt{q}} x_k - \frac{\sqrt{q}}{\mu}\nabla f(x_k) - \frac{(1 - \sqrt{q})^2}{2\sqrt{q}} y_k\right) + \frac{1 - \sqrt{q}}{1 + \sqrt{q}} y_{k+1}$$

$$= \frac{1 + q}{1 + \sqrt{q}} x_k - \frac{2s}{1 + \sqrt{q}}\nabla f(x_k) + \frac{(1 - \sqrt{q})^2}{1 + \sqrt{q}}(y_{k+1} - y_k) + \frac{\sqrt{q}(1 - \sqrt{q})}{1 + \sqrt{q}} y_{k+1}$$

$$= \frac{1 + q}{1 + \sqrt{q}} x_k - \frac{2s}{1 + \sqrt{q}}\nabla f(x_k) + \frac{(1 - \sqrt{q})^2}{1 + \sqrt{q}}(y_{k+1} - y_k) + \frac{\sqrt{q}(1 - \sqrt{q})}{1 + \sqrt{q}}(x_k - s\nabla f(x_k))$$

$$= x_k - (2 - \sqrt{q})s\nabla f(x_k) + \frac{(1 - \sqrt{q})^2}{1 + \sqrt{q}}(y_{k+1} - y_k),$$

which is (49b).

### C.2  Proofs of Lemmas 2–3 in Section 6.1

**Proof of Lemma 2.** By (10b), we have

$$\frac{1}{1 - \nu\sqrt{q}}\frac{\mu}{2}\|z_{k+1} - x^\star\|^2 - \frac{\mu}{2}\|z_k - x^\star\|^2$$

$$= \frac{1}{1 - \nu\sqrt{q}}\frac{\mu}{2}\left\|\nu\sqrt{q}(x_k - \frac{1}{\mu}\nabla f(x_k) - x^\star) + (1 - \nu\sqrt{q})(z_k - x^\star)\right\|^2 - \frac{\mu}{2}\|z_k - x^\star\|^2$$

$$= -\frac{\mu\nu\sqrt{q}}{2}\|z_k - x^\star\|^2 + \mu\nu\sqrt{q}\langle z_k - x^\star, x_k - \frac{1}{\mu}\nabla f(x_k) - x^\star\rangle + \frac{\nu^2 q}{1 - \nu\sqrt{q}}\frac{\mu}{2}\left\|x_k - \frac{1}{\mu}\nabla f(x_k) - x^\star\right\|^2$$

$$= -\frac{\mu\nu\sqrt{q}}{2}\left\|(z_k - x^\star) - (x_k - \frac{1}{\mu}\nabla f(x_k) - x^\star)\right\|^2 + \frac{\nu\sqrt{q}}{1 - \nu\sqrt{q}}\frac{\mu}{2}\left\|x_k - \frac{1}{\mu}\nabla f(x_k) - x^\star\right\|^2$$

$$= -\frac{\mu\nu\sqrt{q}}{2}\left\|z_k - x_k + \frac{1}{\mu}\nabla f(x_k)\right\|^2 + \frac{\nu\sqrt{q}}{1 - \nu\sqrt{q}}\frac{\mu}{2}\left\|x_k - \frac{1}{\mu}\nabla f(x_k) - x^\star\right\|^2. \tag{50}$$

Next, we bound the two terms in the above display separately.

For the second term, by the $\mu$-strong convexity of $f$,

$$f^\star - f(x_k) \geq \langle\nabla f(x_k), x^\star - x_k\rangle + \frac{\mu}{2}\|x_k - x^\star\|^2,$$

from which we have

$$\frac{\nu\sqrt{q}}{1 - \nu\sqrt{q}}\frac{\mu}{2}\left\|x_k - \frac{1}{\mu}\nabla f(x_k) - x^\star\right\|^2$$

$$= \frac{\nu\sqrt{q}}{1 - \nu\sqrt{q}}\left(\frac{\mu}{2}\|x_k - x^\star\|^2 - \langle x_k - x^\star, \nabla f(x_k)\rangle + \frac{1}{2\mu}\|\nabla f(x_k)\|^2\right) \tag{51}$$

$$\leq \frac{\nu\sqrt{q}}{1 - \nu\sqrt{q}}\left(f^\star - f(x_k) + \frac{1}{2\mu}\|\nabla f(x_k)\|^2\right).$$

For the first term, solving (10c) for $z_{k+1}$ we obtain

$$z_{k+1} = x_{k+1} + \frac{1 + (1-\tau)\sqrt{q}}{\tau\sqrt{q}}(x_{k+1} - y_{k+1}) = x_{k+1} + \frac{\zeta}{\tau\sqrt{q}}(x_{k+1} - y_{k+1}),$$

which together with (10a) yields

$$z_{k+1} = x_{k+1} + \frac{\zeta}{\tau\sqrt{q}}\left(x_{k+1} - x_k + \eta s\nabla f(x_k)\right).$$

Hence for $k \geq 1$ we have

$$
\begin{aligned}
&-\frac{\mu\nu\sqrt{q}}{2}\left\|z_k - x_k + \frac{1}{\mu}\nabla f(x_k)\right\|^2 \\
&= -\frac{\mu\nu\sqrt{q}}{2}\left\|\frac{\zeta}{\tau\sqrt{q}}(x_k - x_{k-1}) + \frac{\zeta\eta s}{\tau\sqrt{q}}\nabla f(x_{k-1}) + \frac{1}{\mu}\nabla f(x_k)\right\|^2 \\
&= -\frac{\zeta^2\nu}{\tau^2\sqrt{q}}\frac{\mu}{2}\|x_k - x_{k-1}\|^2 - \frac{\zeta^2\eta^2\nu\sqrt{q}}{\tau^2}\frac{s}{2}\|\nabla f(x_{k-1})\|^2 - \frac{\nu\sqrt{q}}{2\mu}\|\nabla f(x_k)\|^2 \\
&\quad - \frac{\zeta^2\eta\nu\sqrt{q}}{\tau^2}\langle\nabla f(x_{k-1}), x_k - x_{k-1}\rangle - \frac{\zeta\nu}{\tau}\langle\nabla f(x_k), x_k - x_{k-1}\rangle - \frac{\zeta\eta\nu}{\tau}s\langle\nabla f(x_{k-1}), \nabla f(x_k)\rangle.
\end{aligned}
\tag{52}
$$

By the $L$-smoothness of $f$, we have

$$
\begin{cases}
f(x_{k-1}) \geq f(x_k) + \langle\nabla f(x_k), x_{k-1} - x_k\rangle + \frac{1}{2L}\|\nabla f(x_k) - \nabla f(x_{k-1})\|^2, \\
f(x_k) \leq f(x_{k-1}) + \langle\nabla f(x_{k-1}), x_k - x_{k-1}\rangle + \frac{L}{2}\|x_k - x_{k-1}\|^2.
\end{cases}
$$

Therefore,

$$
\begin{aligned}
&-\frac{\zeta^2\eta\nu\sqrt{q}}{\tau^2}\langle\nabla f(x_{k-1}), x_k - x_{k-1}\rangle - \frac{\zeta\nu}{\tau}\langle\nabla f(x_k), x_k - x_{k-1}\rangle \\
&\leq -\frac{\zeta^2\eta\nu\sqrt{q}}{\tau^2}\left(f(x_k) - f(x_{k-1}) - \frac{L}{2}\|x_k - x_{k-1}\|^2\right) - \frac{\zeta\nu}{\tau}\left(f(x_k) - f(x_{k-1}) + \frac{1}{2L}\|\nabla f(x_k) - \nabla f(x_{k-1})\|^2\right) \\
&= \frac{\zeta\nu}{\tau^2}(\tau + \zeta\eta\sqrt{q})(f(x_{k-1}) - f(x_k)) + \frac{\zeta^2\eta\nu\sqrt{q}}{\tau^2}\frac{L}{2}\|x_k - x_{k-1}\|^2 - \frac{\zeta\nu}{\tau}\frac{1}{2L}\|\nabla f(x_k) - \nabla f(x_{k-1})\|^2.
\end{aligned}
\tag{53}
$$

By the cosine rule,

$$-\frac{\zeta\eta\nu s}{\tau}\langle\nabla f(x_{k-1}), \nabla f(x_k)\rangle = -\frac{\zeta\eta\nu s}{2\tau}\left(\|\nabla f(x_{k-1})\|^2 + \|\nabla f(x_k)\|^2 - \|\nabla f(x_k) - \nabla f(x_{k-1})\|^2\right). \tag{54}$$

Combining (52), (53) and (54) and noting $0 \leq \eta s \leq 1/L$ by assumption, we obtain

$$
\begin{aligned}
&-\frac{\mu\nu\sqrt{q}}{2}\left\|z_k - x_k + \frac{1}{\mu}\nabla f(x_k)\right\|^2 \\
&\leq -\frac{\zeta^2\mu\nu}{2\tau^2\sqrt{q}}(1 - \eta L s)\|x_k - x_{k-1}\|^2 - \frac{\zeta\nu}{2\tau L}(1 - \eta L s)\|\nabla f(x_k) - \nabla f(x_{k-1})\|^2 \\
&\quad + \frac{\zeta\nu}{\tau^2}(\tau + \zeta\eta\sqrt{q})(f(x_{k-1}) - f(x_k)) - \frac{\zeta\eta\nu s}{2\tau^2}(\tau + \zeta\eta\sqrt{q})\|\nabla f(x_{k-1})\|^2 - \frac{\nu\sqrt{q}}{2\mu\tau}(\tau + \zeta\eta\sqrt{q})\|\nabla f(x_k)\|^2 \\
&\leq \frac{\zeta\nu}{\tau^2}(\tau + \zeta\eta\sqrt{q})(f(x_{k-1}) - f(x_k)) - \frac{\zeta\eta\nu s}{2\tau^2}(\tau + \zeta\eta\sqrt{q})\|\nabla f(x_{k-1})\|^2 - \frac{\nu\sqrt{q}}{2\mu\tau}(\tau + \zeta\eta\sqrt{q})\|\nabla f(x_k)\|^2.
\end{aligned}
\tag{55}
$$

Collecting (50), (51) and (55) completes the proof.

**Proof of Lemma 3.** The three sets of conditions are mutually exclusive because if $\mathbf{I} > 0$ and $\mathbf{II} > 0$, then $\mathbf{I} + \sqrt{q}\mathbf{II} > 0$ and $\frac{\mu}{L}\mathbf{I} + \sqrt{q}\mathbf{II} > 0$. Applying (42) to the Lyapunov function (43) and using the fact that $\frac{1}{2L}\|\nabla f(x_k)\|^2 \le f(x_k) - f^\star \le \frac{1}{2\mu}\|\nabla f(x_k)\|^2$, we have

$$
V_{k+1} - (1 - \nu\sqrt{q})V_k
$$
$$
\le \frac{\nu}{2\tau^2}\Big(2\sqrt{q}(\mathbf{I})(f(x_k) - f^\star) + s(\mathbf{II})\|\nabla f(x_k)\|^2\Big)
$$
$$
\le
\begin{cases}
\dfrac{\nu}{2\tau^2}\dfrac{\sqrt{q}}{\mu}\Big(\mathbf{I} + \sqrt{q}\mathbf{II}\Big)\|\nabla f(x_k)\|^2, & \text{if } \mathbf{I} \ge 0; \\[2ex]
\dfrac{\nu}{2\tau^2}\dfrac{\sqrt{q}}{\mu}\Big(\dfrac{\mu}{L}\mathbf{I} + \sqrt{q}\mathbf{II}\Big)\|\nabla f(x_k)\|^2, & \text{if } \mathbf{I} \le 0; \\[2ex]
\dfrac{\nu}{\tau^2}\dfrac{\sqrt{q}L}{\mu}\Big(\dfrac{\mu}{L}\mathbf{I} + \sqrt{q}\mathbf{II}\Big)(f(x_k) - f^\star), & \text{if } \mathbf{II} \ge 0; \\[2ex]
\dfrac{\nu}{\tau^2}\sqrt{q}\Big(\mathbf{I} + \sqrt{q}\mathbf{II}\Big)(f(x_k) - f^\star), & \text{if } \mathbf{II} \le 0.
\end{cases}
$$

The rest is straightforward.

### C.3   Proofs of Theorems 1–3

To prepare for the proofs of Theorems 1–3, we show that under Assumption 1, the leading coefficients in the Taylor expansions of $\mathbf{I}$ and $\mathbf{II}$ in $\sqrt{q}$ can be used to verify the conditions in Lemma 3. Throughout, the range $0 < q \lesssim \mu^2/L^2$ or $0 < q \lesssim \mu/L$ is interpreted as, respectively, $0 < q \le C_0\mu^2/L^2$ or $0 < q \le C_0\mu/L$ for a constant $C_0 > 0$.

**Lemma 6.** *Under Assumption 1, denote the Taylor expansions of $\mathbf{I}$ and $\mathbf{II}$ as*

$$
\mathbf{I} = \sum_{n=0}^{\infty} a_n(\sqrt{q})^n, \quad \mathbf{II} = \sum_{m=0}^{\infty} b_m(\sqrt{q})^m,
$$

*where $\{a_n\}_{n\ge 0}$ and $\{b_m\}_{m\ge 0}$ are real sequences. When $\mathbf{I}$ is not constant $0$, define $N$ as the minimal of $n$ such that $a_n \ne 0$. Define $M$ in a similar manner for $\mathbf{II}$. Then*
*(i-a) If $a_N > 0$, $b_M < 0$ and $M \le N - 2$, then $\mathbf{I} > 0$ and $\mathbf{I} + \sqrt{q}\mathbf{II} \le 0$ for $0 < q \lesssim \mu/L$.*
*(i-b) If $a_N > 0$, $b_M < 0$, $M = N - 1$, and the first nonzero element of $\{a_n + b_{n-1}\}_{n\ge N}$ (i.e., the first nonzero coefficient in the expansion of $\mathbf{I} + \sqrt{q}\mathbf{II}$) is negative or the entire sequence is $0$ (i.e., $\mathbf{I} + \sqrt{q}\mathbf{II} \equiv 0$), then $\mathbf{I} > 0$ and $\mathbf{I} + \sqrt{q}\mathbf{II} \le 0$ for $0 < q \lesssim \mu/L$.*
*(ii-a) If $a_N < 0$, $b_M > 0$ and $M = N$, then $\mathbf{II} > 0$ and $\frac{\mu}{L}\mathbf{I} + \sqrt{q}\mathbf{II} \le 0$ for $0 < q \lesssim \mu^2/L^2$.*
*(ii-b) If $a_N < 0$, $b_M > 0$ and $M \ge N + 1$, then $\mathbf{II} > 0$ and $\frac{\mu}{L}\mathbf{I} + \sqrt{q}\mathbf{II} \le 0$ for $0 < q \lesssim \mu/L$.*
*(iii) If $a_N < 0$ (or $N$ does not exist) and $b_M < 0$ (or $M$ does not exist), then $\mathbf{I} \le 0$ and $\mathbf{II} \le 0$ for $0 < q \lesssim \mu/L$.*

*Proof.* With $0 < \mu/L \le 1$, $q$ can be made sufficiently small by picking $C_0$ in the range $0 < q \le C_0\mu^2/L^2$ or $0 < q \le C_0\mu/L$. Hence it suffices to study the leading terms of $\mathbf{I}$ and $\mathbf{II}$.

For (i-a) and (i-b), $a_N > 0$ ensures that $\mathbf{I} \sim a_N(\sqrt{q})^N > 0$. For (i-a), with $M + 1 < N$ and $b_M < 0$, we have $\mathbf{I} + \sqrt{q}\mathbf{II} \sim b_M(\sqrt{q})^{M+1} < 0$. For (i-b), with $M + 1 = N$, we have $\mathbf{I} + \sqrt{q}\mathbf{II} = \sum_{n=N}(a_n + b_{n-1})(\sqrt{q})^n \le 0$ if the first nonzero element of $\{a_n + b_{n-1}\}_{n\ge N}$ is negative or the entire sequence is $0$.

For (ii-a) and (ii-b), $b_M > 0$ ensures that $\mathbf{II} \sim b_M(\sqrt{q})^M > 0$. With $a_N < 0$ and $b_M > 0$, we have that for sufficiently small $q > 0$, $\mathbf{I} < \frac{a_N}{2}(\sqrt{q})^N$ and $\mathbf{II} < 2b_M(\sqrt{q})^M$. Hence, $\frac{\mu}{L}\mathbf{I} + \sqrt{q}\mathbf{II} < \frac{\mu a_N}{2L}(\sqrt{q})^N + 2b_M(\sqrt{q})^{M+1} = (\sqrt{q})^N(\frac{\mu a_N}{2L} + 2b_M(\sqrt{q})^{M+1-N})$. For (ii-a), $M = N$ and $0 < q \le C_0^2\mu^2/L^2$ imply that $\frac{\mu}{L}\mathbf{I} + \sqrt{q}\mathbf{II} < (\sqrt{q})^N\frac{\mu}{2L}(a_N + 4C_0b_M) < 0$ by picking sufficiently small $C_0$ with $a_N < 0$. For (ii-b), $M \ge N + 1$ and $0 < q \le C_0\mu/L$ imply $\frac{\mu}{L}\mathbf{I} + \sqrt{q}\mathbf{II} < (\sqrt{q})^N(\frac{\mu a_N}{2L} + 2b_M(\sqrt{q})^2) \le (\sqrt{q})^N\frac{\mu}{2L}(a_N + 4C_0b_M) < 0$ again by picking sufficiently small $C_0$ with $a_N < 0$.

The case (iii) is straightforward to verify. If $N$ (or $M$) does not exist, then $\mathbf{I} \equiv 0$ (or $\mathbf{II} \equiv 0$).

We notice that the constant $C_0$ in the range of $q$ is picked, depending only on $\{a_n\}$ and $\{b_m\}$, which are determined by the algorithm parameters $\tilde{\eta}$, $\tilde{\nu}$ and $\tilde{\tau}$. $\qquad\square$

Next, we show Theorems $1^*$, $2^*$, and 3, where Theorems $1^*$ and $2^*$ are the same as Theorems 1 and 2 except with conditions (i-a) and (ii-a) replaced by (i-a$^*$) and (ii-a$^*$) as follows:

(i-a$^*$) $0 < \nu_0 < \tau_0$, and $0 \leq \eta_0 < \nu_0\tau_0/2$;

(ii-a$^*$) $0 < \nu_0 < \tau_0$, and $\eta_0 \geq \nu_0\tau_0/2$.

Conditions (i-a) and (ii-a) are the symmetrized (hence weaker) versions of (i-a$^*$) and (ii-a$^*$), by allowing either $0 < \nu_0 < \tau_0$ or $0 < \tau_0 < \nu_0$. To show Theorems $1^*$, $2^*$, and 3, it suffices to verify that the conditions in Lemma 3 are satisfied. In fact, the contraction inequality in Lemma 3 directly implies that that for $k \geq 1$,

$$C(f(x_k) - f^\star) \leq \frac{\zeta\nu}{\tau^2}(\tau + \zeta\eta\sqrt{q})(1 - \eta L s)(f(x_k) - f^\star) \leq V_{k+1} \leq (1 - \nu\sqrt{q})^k V_1 \leq (1 - \frac{\nu_0}{2}\sqrt{q})^k V_1,$$

for some constant $C > 0$, where the first inequality holds by noting $\nu_0, \tau_0 > 0$ in each condition of Theorems $1^*$, $2^*$, and 3 and picking sufficiently small $C_0$ in $0 < q \leq C_0\mu/L$ such that, for example, $\nu \geq \nu_0/2$, $\tau_0/2 \leq \tau \leq 2\tau_0$, $\zeta \geq 1/2$, $0 \leq \eta s \leq 1/(2L)$ (i.e., $0 \leq \eta q \leq \mu/(2L)$) with $C = \nu_0/(16\tau_0)$. Moreover, by the definition of $V_1$,

$$
\begin{aligned}
V_1 &\leq \frac{\zeta\nu}{\tau^2}(\tau + \zeta\eta\sqrt{q})(f(x_0) - f^\star) + \frac{\mu}{2}\|z_1 - x^\star\|^2 \\
&= \frac{\zeta\nu}{\tau^2}(\tau + \zeta\eta\sqrt{q})(f(x_0) - f^\star) + \frac{\mu}{2}\|x_0 - x^\star - \frac{\nu\sqrt{q}}{\mu}\nabla f(x_0)\|^2 \\
&\lesssim L\|x_0 - x^\star\|^2 + \mu\|x_0 - x^\star\|^2 + s\|\nabla f(x_0)\|^2 \\
&\leq (L + \mu + sL^2)\|x_0 - x^\star\|^2 \\
&\lesssim L\|x_0 - x^\star\|^2.
\end{aligned}
\tag{56}
$$

Then $f(x_k) - f^\star \lesssim L(1 - \frac{\nu_0}{2}\sqrt{q})^k\|x_0 - x^\star\|^2$, which is (12). The conditions in Lemma 2, which are required in Lemma 3, can be easily verified by noting $\nu_0, \tau_0 > 0$ and picking sufficiently small $C_0$. Therefore, it remains to verify the conditions involving $\mathbf{I}$ and $\mathbf{II}$ in Lemma 3.

**Proof of Theorems $2^*$ and 3.** We apply Lemma 6 to verify the conditions involving $\mathbf{I}$ and $\mathbf{II}$ in Lemma 3. The Taylor expansions of $\mathbf{I}$ and $\mathbf{II}$ up to $\sqrt{q}$-terms are

$$
\begin{aligned}
\mathbf{I} &= \underbrace{\tau_0(\nu_0 - \tau_0)}_{a_0} + \big[\tau_0(\nu_1 - \tau_1) + \tau_1(\nu_0 - \tau_0) + \nu_0(\eta_0 - \tau_0(\tau_0 - 1))\big]\sqrt{q} + O(q), \\
\mathbf{II} &= \underbrace{\tau_0(\nu_0\tau_0 - 2\eta_0)}_{b_0} + \big[\tau_0(\nu_1\tau_1 - 2\eta_1) + \tau_1(\nu_0\tau_0 - 2\eta_0) + \eta_0\big(2\tau_0(\tau_0 - 1) + \nu_0\tau_0 - \eta_0\big)\big]\sqrt{q} + O(q).
\end{aligned}
\tag{57}
$$

Consider the following scenarios.

**Scenario 1: $\boldsymbol{\eta_0 = 0}$, $\boldsymbol{0 < \nu_0, \tau_0}$.** Then $b_M = b_0 = \nu_0\tau_0^2 > 0$. Because $M = 0$, only case (ii-a) in Lemma 6 is feasible, which holds when $N = 0$ and $a_N = a_0 = \tau_0(\nu_0 - \tau_0) < 0$, i.e., $\nu_0 < \tau_0$. To conclude, if $\eta_0 = 0$, $0 < \nu_0 < \tau_0$, then Lemma 3 holds for $0 < q \lesssim \mu^2/L^2$.

Assume $\eta_0, \nu_0$ and $\tau_0$ are all positive. We notice that $0 < \nu_0 \leq \tau_0$ is necessary. Otherwise, $a_0 = \tau_0(\nu_0 - \tau_0) > 0$ and $N = 0$. Then only cases (i-a) and (i-b) in Lemma 6 are feasible, which require $M \leq N - 1$, contradicting $N = 0$. To proceed, we further split $0 < \nu_0 \leq \tau_0$ into Scenario 2 ($0 < \nu_0 < \tau_0$) and Scenario 3 ($0 < \nu_0 = \tau_0$) as below.

**Scenario 2: $\boldsymbol{\eta_0 > 0}$, $\boldsymbol{0 < \nu_0 < \tau_0}$.** Then $a_0 = \tau_0(\nu_0 - \tau_0) < 0$ and $N = 0$. We notice that either $M$ does not exist (i.e., $\mathbf{II} \equiv 0$), or $M$ exists and has $b_M > 0$ or $b_M < 0$ for some $M \geq N = 0$. Therefore, one of

case (ii-a), case (ii-b), and case (iii) in Lemma 6 is valid. Case (ii-a) holds if and only if $M = N = 0$ and $b_M = b_0 = \tau_0(\nu_0\tau_0 - 2\eta_0) > 0$, i.e., $\eta_0 < \frac{\nu_0\tau_0}{2}$. Therefore, if $\eta_0 \geq \frac{\nu_0\tau_0}{2}$, case (ii-b) or case (iii) holds. To conclude, if $0 < \eta_0 < \frac{\nu_0\tau_0}{2}$ and $0 < \nu_0 < \tau_0$, then Lemma 3 holds for $0 < q \lesssim \mu^2/L^2$. If $\eta_0 \geq \frac{\nu_0\tau_0}{2}$ and $0 < \nu_0 < \tau_0$, then the range of $q$ is relaxed to $0 < q \lesssim \mu/L$.

**Scenario 3: $\eta_0 > 0$, $0 < \nu_0 = \tau_0$.** In this scenario, (57) reduces to

$$\mathbf{I} = \underbrace{\tau_0\big[\nu_1 - \tau_1 + \eta_0 - \tau_0(\tau_0 - 1)\big]}_{a_1} \sqrt{q} + O(q),$$

$$\mathbf{II} = \underbrace{\tau_0(\tau_0^2 - 2\eta_0)}_{b_0} + \underbrace{\big[\tau_0(\nu_1\tau_1 - 2\eta_1) + \tau_1(\tau_0^2 - 2\eta_0) + \eta_0\big(\tau_0(3\tau_0 - 2) - \eta_0\big)\big]}_{b_1} \sqrt{q} + O(q).$$

Then $a_0 = 0$ and hence $N \geq 1$. We point out that $\eta_0 \geq \frac{\tau_0^2}{2}$ is necessary. Otherwise, $b_0 = \tau_0(\tau_0^2 - 2\eta_0) > 0$ and hence $M = 0$. Then only case (ii-a) or (ii-b) in Lemma 6 is possible, which requires $M \geq N$. But this contradicts the fact that $N \geq 1$ and $M = 0$. To proceed, we split $\eta_0 \geq \frac{\tau_0^2}{2}$ into Scenario 3.1 ($\eta_0 > \frac{\tau_0^2}{2}$) and 3.2 ($\eta_0 = \frac{\tau_0^2}{2}$) based on $\eta_0$.

**Scenario 3.1: $\eta_0 > \frac{\tau_0^2}{2}$, $0 < \nu_0 = \tau_0$.** Then $b_0 = \tau_0(\tau_0^2 - 2\eta_0) < 0$ and $M = 0$. Case (i-a), case (i-b) and case (iii) in Lemma 6 are each possible. We consider several special cases involving only $\eta_1$, $\nu_1$ and $\tau_1$.

- For case (i-a) to hold, even higher-order coefficients are needed, and we skip this case.

- For case (i-b) to hold, let $N = M + 1 = 1$ and $a_N = a_1 = \tau_0\big[\nu_1 - \tau_1 + \eta_0 - \tau_0(\tau_0 - 1)\big] > 0$, which is equivalent to $\nu_1 - \tau_1 + \eta_0 - \tau_0(\tau_0 - 1) > 0$. Moreover, let $a_N + b_{N-1} = a_1 + b_0 = \tau_0\big[\nu_1 - \tau_1 + \eta_0 - (\tau_0 - 1)\tau_0\big] + \tau_0(\tau_0^2 - 2\eta_0) = \tau_0[\nu_1 - \tau_1 + \tau_0 - \eta_0] < 0$, which is equivalent to $\nu_1 - \tau_1 < \eta_0 - \tau_0$.

- For case (iii) to hold, let $a_1 = \tau_0\big[\nu_1 - \tau_1 + \eta_0 - \tau_0(\tau_0 - 1)\big] < 0$, which is equivalent to $\nu_1 - \tau_1 < \tau_0(\tau_0 - 1) - \eta_0$.

To conclude, if $\eta_0 > \frac{\tau_0^2}{2}$, $0 < \nu_0 = \tau_0$, and either $\tau_0(\tau_0-1) - \eta_0 < \nu_1 - \tau_1 < \eta_0 - \tau_0$ or $\nu_1 - \tau_1 < \tau_0(\tau_0-1) - \eta_0$, then Lemma 3 holds for $0 < q \lesssim \mu/L$.

**Scenario 3.2: $\eta_0 = \frac{\tau_0^2}{2}$, $0 < \nu_0 = \tau_0$.** Then (57) further reduces to

$$\mathbf{I} = \underbrace{\tau_0\big(\nu_1 - \tau_1 + \tau_0(1 - \frac{\tau_0}{2})\big)}_{a_1} \sqrt{q} + O(q),$$

$$\mathbf{II} = \underbrace{\tau_0\big[\nu_1\tau_1 - 2\eta_1 + \frac{\tau_0^2}{2}(\frac{5}{2}\tau_0 - 2)\big]}_{b_1} \sqrt{q} + O(q).$$

Then $a_0 = b_0 = 0$. We consider several special cases where $(\eta_1, \nu_1, \tau_1)$ are enough to determine the convergence. Let $a_1 = \tau_0\big(\nu_1 - \tau_1 + \tau_0(1 - \frac{\tau_0}{2})\big) < 0$, which is equivalent to $\nu_1 - \tau_1 + \tau_0(1 - \frac{\tau_0}{2}) < 0$. Then $N = 1$. We distinguish three cases by the sign of $b_1 = \tau_0\big[\nu_1\tau_1 - 2\eta_1 + \frac{\tau_0^2}{2}(\frac{5}{2}\tau_0 - 2)\big]$.

- If $b_1 < 0$, then $M = 1$ and $b_M < 0$, and hence case (iii) holds.

- If $b_1 > 0$, then $M = N = 1$ and $b_M > 0$, and hence case (ii-a) holds. Note that the range for $q$ is $0 < q \lesssim \mu^2/L^2$.

- If $b_1 = 0$, then $M \geq 2 = N + 1$ or $M$ does not exist. If $M$ does not exist, case (iii) is valid. If $M$ exists, then either $b_M > 0$ or $b_M < 0$. The former satisfies case (ii-b) and the latter satisfies case (iii).

Collecting the results of Scenario 1, 2 and 3 concludes the proof for Theorem $2^*$ and 3.

**Proof of Theorems $1^*$.** When $(\eta, \nu, \tau)$ are constants, the above analysis still holds, but we can unfold Scenario 3 without imposing strong conditions on $(\eta_1, \nu_1, \tau_1)$. We continue with Scenario 3, that is, assume $\eta \geq \frac{\tau^2}{2}$ and $0 < \nu = \tau$. Then we have the Taylor expansions in finite terms:

$$\mathbf{I} = \tau(\eta - (\tau - 1)\tau)\sqrt{q} - 2\eta\tau(\tau - 1)q + \eta\tau(\tau - 1)^2 q^{\frac{3}{2}},$$

$$\mathbf{II} = \tau(\tau^2 - 2\eta) + \eta[\tau(3\tau - 2) - \eta]\sqrt{q} + \eta(\tau - 1)(2\eta - \tau^2)q - \eta^2(\tau - 1)^2 q^{\frac{3}{2}}, \tag{58}$$

$$\mathbf{I} + \sqrt{q}\mathbf{II} = \tau(\tau - \eta)\sqrt{q} + \eta(\tau^2 - \eta)q + \eta(\tau - 1)(2\eta - \tau)q^{\frac{3}{2}} - \eta^2(\tau - 1)^2 q^2.$$

To proceed, we split Scenario 3 into Scenario $3.1^*$ and $3.2^*$ based on $\eta$.

**Scenario $3.1^*$: $\boldsymbol{\eta > \frac{\tau^2}{2}}$, $\boldsymbol{0 < \nu = \tau}$.** Then $b_0 = \tau(\tau^2 - 2\eta) < 0$, and $M = 0$.

- $\tau > 2$ (hence $\tau < \frac{\tau^2}{2} < \tau(\tau - 1)$).
  - If $\eta > \tau(\tau - 1)$, then $a_1 = \tau(\eta - \tau(\tau - 1)) > 0$ and $N = M + 1 = 1$. Moreover, $a_1 + b_0 = \tau(\tau - \eta) < 0$, and hence case (i-b) in Lemma 6 holds.
  - If $\eta = \tau(\tau - 1)$, then $a_1 = \tau(\eta - \tau(\tau - 1)) = 0$, and $a_2 = -2\eta\tau(\tau - 1) = -2\tau^2(\tau - 1)^2 < 0$. Hence case (iii) in Lemma 6 holds.
  - If $\frac{\tau^2}{2} < \eta < \tau(\tau - 1)$, then $a_1 = \tau(\eta - \tau(\tau - 1)) < 0$. Hence case (iii) in Lemma 6 holds.

- $\tau = 2$ (hence $\tau = \frac{\tau^2}{2} = \tau(\tau - 1) = 2$). Then $\mathbf{I} = 2(\eta - 2)\sqrt{q} + O(q)$ and $\mathbf{I} + \sqrt{q}\mathbf{II} = 2(2 - \eta)\sqrt{q} + O(q)$. By $\eta > \frac{\tau^2}{2} = 2$, we have $a_1 = 2(\eta - 2) > 0$ and $a_1 + b_0 = 2(2 - \eta) < 0$. Case (i-b) holds.

- $0 < \tau < 2$ (hence $\tau > \frac{\tau^2}{2} > \tau(\tau - 1)$). Then $\eta > \frac{\tau^2}{2}$ implies that $\eta > \tau(\tau - 1)$. Hence $a_1 = \tau(\eta - \tau(\tau - 1)) > 0$ and $N = M + 1 = 1$. Only case (i-b) in Lemma 6 is possible, which requires the first non-zero coefficient of $\mathbf{I} + \sqrt{q}\mathbf{II}$ in (58) to be negative, or $\mathbf{I} + \sqrt{q}\mathbf{II} \equiv 0$.
  - If $\eta > \tau$, then $\mathbf{I} + \sqrt{q}\mathbf{II} = \tau(\tau - \eta)\sqrt{q} + O(q) < 0$.
  - If $\eta = \tau$, then $\mathbf{I} + \sqrt{q}\mathbf{II} = \tau^2(\tau - 1)q + \tau^2(\tau - 1)q^{\frac{3}{2}} - \tau^2(\tau - 1)^2 q^2$. Hence for $0 < \tau \leq 1$, $\mathbf{I} + \sqrt{q}\mathbf{II} \leq 0$.
  - If $\frac{\tau^2}{2} < \eta < \tau$, then $\mathbf{I} + \sqrt{q}\mathbf{II} = \tau(\tau - \eta)\sqrt{q} + O(q) > 0$, which violates case (i-b).

**Scenario $3.2^*$: $\boldsymbol{\eta = \frac{\tau^2}{2}}$, $\boldsymbol{0 < \nu = \tau}$.** Then (58) only involves $\tau$: $\mathbf{I} = \tau^2(1 - \frac{\tau}{2})\sqrt{q} - \tau^3(\tau - 1)q + \frac{1}{2}\tau^3(\tau - 1)^2 q^{\frac{3}{2}}$, $\mathbf{II} = \frac{1}{2}\tau^3(\frac{5}{2}\tau - 2)\sqrt{q} - \frac{1}{4}\tau^4(\tau - 1)^2 q^{\frac{3}{2}}$, and $\mathbf{I} + \sqrt{q}\mathbf{II} = \tau^2(1 - \frac{\tau}{2})\sqrt{q} + O(q)$.

- If $\tau > 2$, then $a_N = a_1 = \tau^2(1 - \frac{\tau}{2}) < 0$, and $b_M = b_1 = \frac{1}{2}\tau^3(\frac{5}{2}\tau - 2) > 0$. Hence case (ii-a) in Lemma 6 holds. Note that the range of $q$ is $0 < q \lesssim \mu^2/L^2$.

- If $\tau = 2$, then $\mathbf{I} = -8q + 4q^{\frac{3}{2}}$ and $\mathbf{II} = 12\sqrt{q} - 4q^{\frac{3}{2}}$. Hence $a_N = a_2 = -8 < 0$ and $b_M = b_1 = 12 > 0$, with $1 = M < N = 2$, which contradicts case (ii-a) and (ii-b) requiring $M \geq N$. This case is invalid.

- If $0 < \tau < 2$, then $a_N = a_1 = \tau^2(1 - \frac{\tau}{2}) > 0$. Only case (i-a) or (i-b) in Lemma 6 is possible. But then $\mathbf{I} + \sqrt{q}\mathbf{II} = \tau^2(1 - \frac{\tau}{2})\sqrt{q} + O(q) > 0$, contradicting the conclusion in (i-a) and (i-b). This case is invalid.

Collecting the results in Scenario $3.1^*$ and $3.2^*$ concludes the proof for Theorem $1^*$.

Finally, we derive Theorems 1 and 2 from Theorems $1^*$ and $2^*$ by exploiting symmetrization in the case of $\nu_0 \neq \tau_0$ due to Lemma 7 in Appendix C.4.

**Proof of Theorems 1 and 2.** It suffices to only deal with conditions (i-a) and (ii-a) in Theorem 2, which directly implies the conclusions from conditions (i-a) and (ii-a) in Theorem 1.

For $\eta \sim \eta_0 \geq 0$, $\nu \sim \nu_0 > 0$, $\tau \sim \tau_0 > 0$, and $\nu_0 \neq \tau_0$, by Lemma 7, algorithm (10) can be first put into (59), with $R_1$, $R_2$, $R_3$ and $h_1 = \frac{\zeta \eta + \nu \tau}{1 + \sqrt{q}}$. Next, we keep $\eta_0$ and the remainder terms, but exchange the role of $\nu_0$ and $\tau_0$ by setting $\bar{\nu}_0 = \tau_0$ and $\bar{\tau}_0 = \nu_0$ and then translate (59) back to (10) with new parameters $\bar{\eta} \sim \eta_0$, $\bar{\nu} \sim \bar{\nu}_0 = \tau_0$ and $\bar{\tau} \sim \bar{\tau}_0 = \nu_0$ ($\zeta$ is also translated to the new $\bar{\zeta} = 1 + (1 - \bar{\tau})\sqrt{q}$), and a possibly nonzero $h_2 = \frac{\bar{\zeta}\bar{\eta} + \bar{\nu}\bar{\tau} - (\zeta \eta + \nu \tau)}{\bar{\tau}(1 - \bar{\nu}\sqrt{q})}$ in $z_0$. In other words, the original algorithm (10) can be reformulated such that the leading constants in $\nu$ and $\tau$ are exchanged and the algorithm now starts with $x_0$ and possibly $z_0 \neq x_0$.

For the reformulated algorithm, the proof of Theorem 2* remains valid except for the bound of $V_1$ in (56) with the new $z_0$. Nevertheless, an inspection of (56) reveals that $V_1 \lesssim L\|x_0 - x^\star\|^2$ still holds because $h_2$ can be easily shown to be bounded. Hence, the desired result follows by symmetrizing the conditions (i-a*) and (ii-a*) in Theorem 2*.

## C.4 Proof of Corollary 1

To facilitate interpretation of (10) and prepare for the proof of Corollary 1, we study the single-variable form of (10). The following lemma shows that the two forms can be transformed into each other, provided that the leading constants in $\nu$ and $\tau$ differ from each other. The initial points need to be aligned because (10) starts from $x_0$ and $z_0$ while (59) starts from $x_0$ and $x_1$.

**Lemma 7.** *Let $\zeta = 1 + (1 - \tau)\sqrt{q}$ as in Lemma 2. (i) Algorithm (10) with tuning parameters $\eta$, $\nu$ and $\tau$ under Assumption 1 admits the single-variable form (11), which can be expressed as*

$$x_{k+1} = x_k - (\nu_0\tau_0 + R_1)s\nabla f(x_k) + (1 - (\nu_0 + \tau_0)\sqrt{q} + R_2)(x_k - x_{k-1}) - (\eta_0 + R_3)s(\nabla f(x_k) - \nabla f(x_{k-1})), \tag{59}$$

*with $x_0$, $x_1 = x_0 - h_1 s\nabla f(x_0)$, where $R_1 = O(\sqrt{q})$, $R_2 = O(q)$, $R_3 = O(\sqrt{q})$ and $h_1 = \frac{\zeta \eta + \nu \tau}{1 + \sqrt{q}}$ are analytic functions of $\sqrt{q}$ around 0. (ii) Conversely, given any analytic functions of $\sqrt{q}$: $R_1 = O(\sqrt{q})$, $R_2 = O(q)$, $R_3 = O(\sqrt{q})$, $h_1$, and three scalars $\eta_0 \geq 0$, $\nu_0, \tau_0 > 0$, $\nu_0 \neq \tau_0$, there exist $\eta \sim \eta_0$, $\nu \sim \nu_0$ and $\tau \sim \tau_0$ satisfying Assumption 1 such that (59) starting from $x_0$ and $x_1 = x_0 - h_1 s\nabla f(x_0)$ is equivalent to (10) starting from $x_0$ and $z_0 = x_0 + h_2 \frac{\sqrt{q}}{\mu} \nabla f(x_0)$ where*

$$h_2 = \frac{\zeta \eta + \nu \tau - (1 + \sqrt{q})h_1}{\tau(1 - \nu\sqrt{q})}.$$

*Proof.* First, we show that (10) admits the single-variable form (11), i.e.,

$$x_{k+1} = x_k - \frac{\nu(\tau + \zeta\eta\sqrt{q})}{1 + \sqrt{q}}s\nabla f(x_k) + \frac{\zeta(1 - \nu\sqrt{q})}{1 + \sqrt{q}}(x_k - x_{k-1}) - \frac{\zeta\eta(1 - \nu\sqrt{q})}{1 + \sqrt{q}}s(\nabla f(x_k) - \nabla f(x_{k-1})),$$

for $k \geq 1$ starting from $x_0$ and $x_1 = x_0 - \frac{\zeta\eta + \nu\tau}{1 + \sqrt{q}}s\nabla f(x_0)$. The calculation for $x_1$ is straightforward and hence omitted. To show (11), from (10c) and (10a) we have for $k \geq 0$,

$$z_{k+1} = \frac{1 + \sqrt{q}}{\tau\sqrt{q}}x_{k+1} - \frac{\zeta}{\tau\sqrt{q}}y_{k+1} = \frac{1 + \sqrt{q}}{\tau\sqrt{q}}x_{k+1} - \frac{\zeta}{\tau\sqrt{q}}(x_k - \eta s\nabla f(x_k)).$$

Substituting the above display with subscript $k + 1$ and $k$ for $k \geq 1$ into (10b), we have

$$\frac{1 + \sqrt{q}}{\tau\sqrt{q}}x_{k+1} - \frac{\zeta}{\tau\sqrt{q}}(x_k - \eta s\nabla f(x_k))$$

$$= \nu\sqrt{q}(x_k - \frac{1}{\mu}\nabla f(x_k)) + (1 - \nu\sqrt{q})\left(\frac{1 + \sqrt{q}}{\tau\sqrt{q}}x_k - \frac{\zeta}{\tau\sqrt{q}}(x_{k-1} - \eta s\nabla f(x_{k-1}))\right).$$

After rearrangement we obtain

$$\frac{1+\sqrt{q}}{\tau\sqrt{q}}x_{k+1} = \left(\frac{1+\sqrt{q}}{\tau\sqrt{q}} + \frac{\zeta(1-\nu\sqrt{q})}{\tau\sqrt{q}}\right)x_k - \frac{\zeta(1-\nu\sqrt{q})}{\tau\sqrt{q}}x_{k-1}$$

$$-\frac{\zeta\eta + \nu\tau}{\tau\sqrt{q}}s\nabla f(x_k) + \frac{\zeta\eta(1-\nu\sqrt{q})}{\tau\sqrt{q}}s\nabla f(x_{k-1}).$$

Solving for $x_{k+1}$ yields (11). Expanding the coefficients in (11) yields (59).

Second, we prove the reverse statement. Given $R_1 = O(\sqrt{q})$, $R_2 = O(q)$, $R_3 = O(\sqrt{q})$ and the leading constants $\eta_0$, $\nu_0$ and $\tau_0$, we determine $\eta$, $\nu$ and $\tau$ by solving the following equations

$$\nu(\tau + \zeta\eta\sqrt{q}) = (\nu_0\tau_0 + R_1)(1+\sqrt{q}) = g_1 \sim \nu_0\tau_0 + O(\sqrt{q}),$$
$$\zeta(1-\nu\sqrt{q}) = (1 - (\nu_0 + \tau_0)\sqrt{q} + R_2)(1+\sqrt{q}) = g_2 \sim 1 - (\nu_0 + \tau_0 - 1)\sqrt{q} + O(q),$$
$$\zeta\eta(1-\nu\sqrt{q}) = (\eta_0 + R_3)(1+\sqrt{q}).$$

Note that $g_1$ and $g_2$ above are known. Solving the equations we obtain

$$\eta = \frac{\eta_0 + R_3}{1 - (\nu_0 + \tau_0)\sqrt{q} + R_2} \sim \eta_0,$$

and $\nu$ depending on $\tau$ as

$$\nu = \frac{g_1}{\tau + \zeta\eta\sqrt{q}} \sim \frac{\nu_0\tau_0}{\tau_0} = \nu_0,$$

and $\tau$ as a root for the quadratic equation

$$\alpha_2 \cdot \tau^2 + \alpha_1 \cdot \tau + \alpha_0 = 0,$$

where

$$\alpha_2 = -\sqrt{q}(1 - \eta q),$$
$$\alpha_1 = 1 - g_2 + \sqrt{q} + (g_1 - 2\eta + \eta g_2)q - 2\eta q^{\frac{3}{2}} = \sqrt{q}(\nu_0 + \tau_0 + O(\sqrt{q})),$$
$$\alpha_0 = (\eta - g_1 - \eta g_2)\sqrt{q} + (2\eta - g_1 - \eta g_2)q + \eta q^{\frac{3}{2}} = \sqrt{q}(-\nu_0\tau_0 + O(\sqrt{q})).$$

The discriminant is $\Delta = \alpha_1^2 - 4\alpha_2 \cdot \alpha_0 = q[(\nu_0 - \tau_0)^2 + O(\sqrt{q})]$. For $\nu_0 \neq \tau_0 > 0$, the root

$$\tau = \frac{-\alpha_1 \pm \sqrt{\Delta}}{2 \cdot \alpha_2} = \frac{\alpha_1/\sqrt{q} \pm \sqrt{\Delta/q}}{-2 \cdot \alpha_2/\sqrt{q}} = \frac{\nu_0 + \tau_0 + O(\sqrt{q}) \pm \sqrt{(\nu_0 - \tau_0)^2 + O(\sqrt{q})}}{2(1 - \eta q)} \sim \frac{\nu_0 + \tau_0 \pm |\nu_0 - \tau_0|}{2}$$

is well-defined for small $q$. The sign is determined by the sign of $\nu_0 - \tau_0$ to make $\tau \sim \tau_0$. An inspection of the expressions suggests that $\eta$, $\nu$ and $\tau$ are all analytic functions of $\sqrt{q}$.

Lastly, from the above calculation, the updating formula for $x_k$ has been matched between (10) and (59), starting from $x_2$. The initial points can be aligned by picking suitable $z_0$ such that $x_1$ from (10) is exactly $x_0 - h_1 s\nabla f(x_0)$. The calculation is straightforward and hence omitted. $\qquad\square$

Interestingly, the leading coefficients in the single-variable form (59) for (10) are symmetric with regard to $\nu_0$ and $\tau_0$. This suggests that the convergence properties of (10) should also be symmetric in $\nu_0$ and $\tau_0$. Indeed, such symmetrization in the case of $\nu_0 \neq \tau_0$ is exploited in our derivation of Theorems 1–2 from Theorems 1*–2* in Appendix C.3.

Corollary 1 is derived by translating the conclusions from Theorem 2 in the case of $\nu_0 \neq \tau_0$ in terms of the leading coefficients in a single-variable form.

**Proof of Corollary 1.** Let $c_1 = \nu_0 + \tau_0$, $c_0 = \nu_0\tau_0$, $c_2\sqrt{c_0} - c_0/2 = \eta_0$ and solve for $\eta_0$, $\nu_0$ and $\tau_0$. The conditions $c_1^2 > 4c_0$ and $c_0, c_1 > 0$ ensure that $\nu_0$ and $\tau_0$ exist, satisfying $\nu_0, \tau_0 > 0$ and $\nu_0 \neq \tau_0$. The conditions on $c_2$ can be directly translated into the conditions on $\eta_0$. The desired result follows by applying Lemma 7 and Theorem 2.

# D Technical details in Section 3

## D.1 Derivation of three-variable form (22)

By comparing (22) with (20), the update of $y_{k+1}$ is the same. Rearranging $z_{k+1} = \alpha_{k+1} x_{k+1} + (1-\alpha_{k+1}) y_{k+1}$, we obtain the update of $x_{k+1}$ in (22c). It suffices to prove the update of $z_{k+1}$ in (22b). With (20) in place, we have

$$
\begin{aligned}
z_{k+1} &= \alpha_{k+1} x_{k+1} + (1 - \alpha_{k+1}) y_{k+1} \\
&= \alpha_{k+1} [x_k - \gamma_k s \nabla f(x_k) + \sigma_{k+1}(y_{k+1} - y_k)] + (1 - \alpha_{k+1}) y_{k+1} \\
&= \alpha_{k+1}(x_k - \gamma_k s \nabla f(x_k)) + (\alpha_k - 1)(y_{k+1} - y_k) + (1 - \alpha_{k+1}) y_{k+1} \\
&= \alpha_{k+1}(x_k - \gamma_k s \nabla f(x_k)) + (\alpha_k - \alpha_{k+1}) y_{k+1} + (1 - \alpha_k) y_k \\
&= \alpha_{k+1}(x_k - \gamma_k s \nabla f(x_k)) + (\alpha_k - \alpha_{k+1})(x_k - \beta_k s \nabla f(x_k)) + (1 - \alpha_k) y_k \\
&= \alpha_k x_k + (1 - \alpha_k) y_k - (\gamma_k \alpha_{k+1} + \beta_k(\alpha_k - \alpha_{k+1})) s \nabla f(x_k) \\
&= z_k - \widetilde{\alpha}_k s \nabla f(x_k).
\end{aligned}
$$

## D.2 Proofs of Lemmas 4 and 5 in Section 6.2

**Proof of Lemma 4.** Using (22b), we have

$$
\begin{aligned}
\frac{1}{2}\|z_{k+1} - x^\star\|^2 - \frac{1}{2}\|z_k - x^\star\|^2 &= \frac{1}{2}\|z_k - x^\star - \widetilde{\alpha}_k s \nabla f(x_k)\|^2 - \frac{1}{2}\|z_k - x^\star\|^2 \\
&= -\widetilde{\alpha}_k s \langle z_k - x^\star, \nabla f(x_k) \rangle + \frac{\widetilde{\alpha}_k^2 s^2}{2}\|\nabla f(x_k)\|^2.
\end{aligned}
$$

Substituting $z_k = x_k + (\alpha_k - 1)(x_k - y_k)$ into the above display, we have

$$
\frac{1}{2}\|z_{k+1} - x^\star\|^2 - \frac{1}{2}\|z_k - x^\star\|^2 \tag{60}
$$

$$
= -\widetilde{\alpha}_k s \langle x_k - x^\star, \nabla f(x_k) \rangle - \widetilde{\alpha}_k(\alpha_k - 1) s \langle x_k - y_k, \nabla f(x_k) \rangle + \frac{\widetilde{\alpha}_k^2 s^2}{2}\|\nabla f(x_k)\|^2
$$

$$
\leq -\widetilde{\alpha}_k s(f(x_k) - f^\star) - \widetilde{\alpha}_k(\alpha_k - 1) s(f(x_k) - f(y_k)) + \frac{\widetilde{\alpha}_k^2 s^2}{2}\|\nabla f(x_k)\|^2
$$

$$
= -\alpha_k \widetilde{\alpha}_k s(f(x_k) - f^\star) + \widetilde{\alpha}_k(\alpha_k - 1) s(f(y_k) - f^\star) + \frac{\widetilde{\alpha}_k^2 s^2}{2}\|\nabla f(x_k)\|^2,
$$

where the inequality holds because by the convexity of $f$ and the assumption that $\widetilde{\alpha}_k \geq 0$ and $\alpha_k \geq 1$, we have $\langle x_k - x^\star, \nabla f(x_k) \rangle \geq f(x_k) - f^\star$ and $\langle x_k - y_k, \nabla f(x_k) \rangle \geq f(x_k) - f(y_k)$. When $k \geq 1$, by the $L$-smoothness of $f$ and (22a), we have

$$
\begin{aligned}
f(y_k) - f^\star &\leq f(x_{k-1}) - f^\star + \langle \nabla f(x_{k-1}), y_k - x_{k-1} \rangle + \frac{L}{2}\|y_k - x_{k-1}\|^2 \\
&= f(x_{k-1}) - f^\star - (2 - \beta_{k-1} L s) \frac{\beta_{k-1} s}{2}\|\nabla f(x_{k-1})\|^2.
\end{aligned} \tag{61}
$$

Combining (60) and (61) yields (45), which completes the proof.

**Proof of Lemma 5.** By Lemma 4, for any $k \geq 1$ such that $\alpha_k \geq 1$, $\widetilde{\alpha}_k \geq 0$, we have

$$
\begin{aligned}
V_{k+1} - V_k \leq &-\frac{1}{2}(\omega_k - \omega_{k+1})\|z_k - x^\star\|^2 \\
&- \underbrace{(\omega_k \alpha_{k-1} \widetilde{\alpha}_{k-1} - \omega_{k+1} \widetilde{\alpha}_k(\alpha_k - 1))}_{\text{I}} s(f(x_{k-1}) - f^\star) \\
&- \underbrace{(\omega_{k+1} \widetilde{\alpha}_k(\alpha_k - 1)\beta_{k-1}(2 - \beta_{k-1} L s) - \omega_k \widetilde{\alpha}_{k-1}^2)}_{\text{II}} \frac{s^2}{2}\|\nabla f(x_{k-1})\|^2.
\end{aligned} \tag{62}
$$

If $\omega_k \geq \omega_{k+1}$, then $-(\omega_k - \omega_{k+1})\|z_k - x^\star\|^2/2 \leq 0$. The rest is straightforward.

### D.3 Proof of Theorem 4

To prepare for the proof for Theorem 4, we provide a simple bound which will be used in the last step of the proof.

**Lemma 8.** *Let $f \in \mathcal{F}_L^1$. When $0 < s \leq C_0/L$ for some constant $C_0 > 0$, the iterates of (22) satisfy that for any fixed $K$, there exists a constant $C$ such that $s(f(x_K) - f^\star)$, $s^2\|\nabla f(x_K)\|^2$ and $\|z_{K+1} - x^\star\|^2$ are upper-bounded by $C\|x_0 - x^\star\|^2$. The constant $C$ depends only on $C_0$, $K$ and the algorithm parameters $\{\alpha_k\}$, $\{\beta_k\}$ and $\{\gamma_k\}$.*

*Proof.* For notational simplicity, we assume $C_0 = 1$ so that $0 < Ls \leq 1$. First, by the convexity of $f$, we have

$$s(f(x_K) - f^\star) \leq \frac{Ls}{2}\|x_K - x^\star\|^2 \leq \frac{1}{2}\|x_K - x^\star\|^2.$$

Second, by the $L$-smoothness, we have

$$s^2\|\nabla f(x_K)\|^2 = s^2\|\nabla f(x_K) - \nabla f(x^\star)\|^2 \leq (Ls)^2\|x_K - x^\star\|^2 \leq \|x_K - x^\star\|^2.$$

Third, by (22b) and the Cauchy–Schwarz inequality, we have

$$\frac{1}{2}\|z_{K+1} - x^\star\|^2 = \frac{1}{2}\|z_K - x^\star - \widetilde{\alpha}_K s\nabla f(x_K)\|^2 \leq \|z_K - x^\star\|^2 + \widetilde{\alpha}_K^2 s^2\|\nabla f(x_K)\|^2,$$

which together with the preceding bound on $s^2\|\nabla f(x_K)\|^2$ yields

$$\frac{1}{2}\|z_{K+1} - x^\star\|^2 \leq \|z_K - x^\star\|^2 + \widetilde{\alpha}_K^2\|x_K - x^\star\|^2.$$

Because $z_0 = x_0$, it suffices to bound $\|x_k - x^\star\|^2$ by $\|x_0 - x^\star\|^2$ up to a constant for general $k \geq 1$. The fact that (22) is a first-order method implies that for each $k \geq 1$, there exist scalars $\{c_{k,i}\}$ (depending only on the algorithm parameters) for $i = 0, \ldots, k-1$ such that

$$x_k = x_{k-1} + \sum_{i=0}^{k-1} c_{k,i} s\nabla f(x_i).$$

Then by the Cauchy–Schwarz inequality,

$$\begin{aligned}
\|x_k - x^\star\|^2 &= \|x_{k-1} - x^\star + \sum_{i=0}^{k-1} c_{k,i} s\nabla f(x_i)\|^2 \\
&\leq k\left(\|x_{k-1} - x^\star\|^2 + \sum_{i=0}^{k-1} c_{k,i}^2 s^2\|\nabla f(x_i)\|^2\right) \\
&\leq k\left(\|x_{k-1} - x^\star\|^2 + \sum_{i=0}^{k-1} c_{k,i}^2\|x_i - x^\star\|^2\right).
\end{aligned}$$

Because $K$ is fixed, the proof is completed by applying the preceding bound for $1 \leq k \leq K$. □

With Lemma 5 and Lemma 8 in place, we are ready to prove Theorem 4.

**Proof of Theorem 4.** By the definition of $\widetilde{\alpha}_k$ and condition (i)-(ii), we have

$$\lim_{k \to \infty} \frac{\widetilde{\alpha}_k}{\alpha_k} = \lim_{k \to \infty} \beta_k + (\gamma_k - \beta_k)\frac{\alpha_{k+1}}{\alpha_k} = \gamma > 0.$$

Then $\alpha_k = \Omega(k)$ implies that $\widetilde{\alpha}_k = \Omega(k)$. By Lemma 5, the key is to bound **I** and **II** from below for all $k$ sufficiently large. The beginning $V_k$ can be dealt with by the simple bound in Lemma 8 so they do not affect

the convergence. To proceed, we consider two choices of $\{\omega_k\}$ depending on the monotonicity of $\{\widetilde{\alpha}_k/\alpha_k\}$. Note that $\{\omega_k\}$ needs to be non-increasing, to ensure that the first term $-(\omega_k - \omega_{k+1})\|z_k - x^\star\|^2/2$ on the right-hand-side of (62) is always non-positive.

**Choice 1: $\omega_k \equiv 1$.** We pick $w_k$ simply as constant 1 when $\{\widetilde{\alpha}_k/\alpha_k\}$ is non-increasing. As for **I**, by condition (ii) we have

$$\mathbf{I} = \alpha_{k-1}\widetilde{\alpha}_{k-1} - \widetilde{\alpha}_k(\alpha_k - 1) = \alpha_{k-1}^2 \left( \frac{\widetilde{\alpha}_{k-1}}{\alpha_{k-1}} - \frac{\alpha_k(\alpha_k - 1)}{\alpha_{k-1}^2} \cdot \frac{\widetilde{\alpha}_k}{\alpha_k} \right) \geq \alpha_{k-1}^2 \left( \frac{\widetilde{\alpha}_{k-1}}{\alpha_{k-1}} - \frac{\widetilde{\alpha}_k}{\alpha_k} \right) \geq 0.$$

As for **II**, we have

$$\mathbf{II} = \widetilde{\alpha}_k(\alpha_k - 1)\beta_{k-1}(2 - \beta_{k-1}Ls) - \widetilde{\alpha}_{k-1}^2 = \alpha_k(\alpha_k - 1) \left( \frac{\widetilde{\alpha}_k}{\alpha_k}\beta_{k-1}(2 - \beta_{k-1}Ls) - \frac{\widetilde{\alpha}_{k-1}^2}{\alpha_k(\alpha_k - 1)} \right).$$

For $0 < s \leq (2 - \gamma/\beta)/(2\beta L)$ with $\beta > \gamma/2 > 0$,

$$\mathbf{II} \geq \alpha_k(\alpha_k - 1) \left[ \frac{\widetilde{\alpha}_k}{\alpha_k}\beta_{k-1} \left( 2 - \frac{\beta_{k-1}}{2\beta}(2 - \frac{\gamma}{\beta}) \right) - \frac{\widetilde{\alpha}_{k-1}^2}{\alpha_k(\alpha_k - 1)} \right],$$

where the limit of the term in the square brackets is

$$\lim_{k \to \infty} \frac{\widetilde{\alpha}_k}{\alpha_k}\beta_{k-1} \left( 2 - \frac{\beta_{k-1}}{2\beta}(2 - \frac{\gamma}{\beta}) \right) - \frac{\widetilde{\alpha}_{k-1}^2}{\alpha_k(\alpha_k - 1)} = \gamma(\beta - \frac{\gamma}{2}) > 0.$$

Combining the preceding two displays and $\alpha_k = \Omega(k)$ shows that there exist constants $K$ and $C > 0$ depending only on algorithm parameters such that for $0 < s \leq (2 - \gamma/\beta)/(2\beta L)$ and $k \geq K$,

$$\mathbf{II} \geq \frac{\gamma}{2}(\beta - \frac{\gamma}{2})\alpha_k(\alpha_k - 1) \geq Ck^2.$$

**Choice 2: $\omega_{k+1} = \alpha_k/\widetilde{\alpha}_k$.** When $\{\widetilde{\alpha}_k/\alpha_k\}$ is non-decreasing, $\{\omega_k\}$ is non-increasing. In this case we have $\mathbf{I} = \alpha_{k-1}^2 - \alpha_k(\alpha_k - 1) \geq 0$. For $0 < s \leq (2 - \gamma/\beta)/(2\beta L)$,

$$\mathbf{II} = \alpha_k(\alpha_k - 1)\beta_{k-1}(2 - \beta_{k-1}Ls) - \alpha_{k-1}\widetilde{\alpha}_{k-1}$$

$$= \alpha_k(\alpha_k - 1) \left( \beta_{k-1}(2 - \beta_{k-1}Ls) - \frac{\alpha_{k-1}^2}{\alpha_k(\alpha_k - 1)} \cdot \frac{\widetilde{\alpha}_{k-1}}{\alpha_{k-1}} \right)$$

$$\geq \alpha_k(\alpha_k - 1) \left[ \beta_{k-1} \left( 2 - \frac{\beta_{k-1}}{2\beta}(2 - \frac{\gamma}{\beta}) \right) - \frac{\alpha_{k-1}^2}{\alpha_k(\alpha_k - 1)} \cdot \frac{\widetilde{\alpha}_{k-1}}{\alpha_{k-1}} \right],$$

where the limit of the term in the square brackets is

$$\lim_{k \to \infty} \beta_{k-1} \left( 2 - \frac{\beta_{k-1}}{2\beta}(2 - \frac{\gamma}{\beta}) \right) - \frac{\alpha_{k-1}^2}{\alpha_k(\alpha_k - 1)} \cdot \frac{\widetilde{\alpha}_{k-1}}{\alpha_{k-1}} = \beta - \frac{\gamma}{2} > 0.$$

Similarly as in the first choice, there exist constants $K$ and $C > 0$ depending only on algorithm parameters such that for $0 < s \leq (2 - \gamma/\beta)/(2\beta L)$ and $k \geq K$,

$$\mathbf{II} \geq (\beta - \frac{\gamma}{2})\alpha_k(\alpha_k - 1) \geq Ck^2.$$

Combining the two choices above, we see that when $\{\widetilde{\alpha}_k/\alpha_k\}$ is either non-increasing or non-decreasing in $k$, there exist constants $K$ and $C > 0$ such that for $k \geq K$, we have $\mathbf{I} \geq 0$ and $\inf_{0 < s \leq C_0/L} \mathbf{II} \geq Ck^2$, where $C_0 = (2 - \gamma/\beta)/(2\beta)$. By Lemma 5,

$$V_{k+1} - V_k \leq -\frac{C}{2}k^2 s^2 \|\nabla f(x_{k-1})\|^2 \leq 0.$$

To complete the proof of (16) for the objective gap, it suffices to bound $V_k$ from below.

By convexity of $f$, we have $\|\nabla f(x_k)\|^2 \leq (2L)(f(x_k) - f^\star)$ which together with the definition of $V_{k+1}$ in (46) yields that for $k \geq K$ and $0 < s \leq C_0/L$,

$$V_K \geq V_{k+1} \geq \omega_{k+1}(\alpha_k\widetilde{\alpha}_k - \widetilde{\alpha}_k^2 Ls)s(f(x_k) - f^\star) = \omega_{k+1}\alpha_k\widetilde{\alpha}_k\left(1 - \frac{\widetilde{\alpha}_k}{\alpha_k}Ls\right)s(f(x_k) - f^\star).$$

Because $\lim_k \omega_k$ is 1 or $1/\gamma > 0$ in the Choice 1 or 2 above, and $\lim_k \widetilde{\alpha}_k/\alpha_k = \gamma > 0$, we reset $K$ large enough such that the above is further bounded from below as

$$V_K \geq V_{k+1} \geq \frac{1}{2}(\frac{1}{\gamma} \wedge 1)\alpha_k\widetilde{\alpha}_k(1 - 2\gamma Ls)s(f(x_k) - f^\star).$$

By resetting $C_0 = \frac{2-\gamma/\beta}{2\beta} \wedge \frac{1}{4\gamma}$, we have that for $k \geq K$ and $0 < s \leq C_0/L$,

$$V_K \geq V_{k+1} \geq \frac{1}{4}(\frac{1}{\gamma} \wedge 1)\alpha_k\widetilde{\alpha}_k s(f(x_k) - f^\star).$$

Using Lemma 8, we have $f(x_k) - f^\star = O(V_K/(\alpha_k\widetilde{\alpha}_k s)) = O(V_K/(sk^2)) = O(\|x_0 - x^\star\|^2/(sk^2))$, which is the optimal rate (16).

Finally, we complete the proof of (18) for the squared gradient norm. For $k \geq K$ and $0 < s \leq C_0/L$, using $V_{k+1} - V_k \leq -\frac{C}{2}k^2s^2\|\nabla f(x_{k-1})\|^2$, we have

$$\frac{C}{2}\left(\sum_{i=K}^{k} i^2\right)s^2 \min_{0 \leq i \leq k-1} \|\nabla f(x_i)\|^2 \leq \frac{C}{2}\sum_{i=K}^{k} i^2 s^2\|\nabla f(x_{i-1})\|^2 \leq \sum_{i=K}^{k}(V_i - V_{i+1}) = V_K - V_{k+1} \leq V_K,$$

which together with Lemma 8 implies that for $k \geq 2K$,

$$\min_{0 \leq i \leq k-1} \|\nabla f(x_i)\|^2 \leq \frac{2V_K}{Cs^2 \sum_{i=K}^{k} i^2} \leq \frac{4V_K}{Cs^2 \sum_{i=1}^{k} i^2} = \frac{24V_K}{Cs^2 k(k+1)(2k+1)} = O\left(\frac{\|x_0 - x^\star\|^2}{s^2k^3}\right),$$

which is the desired bound (18).

### D.4    Proof of Lemma 1

First, we show $\alpha_k = \Omega(k)$. When $k$ is odd, $\alpha_k = (1+\sqrt{1 + 4\alpha_{k-1}^2})/2 > (1+2\alpha_{k-1})/2 = (1+2(k+r-1)/r)/2 = (k + r - 1)/r + 1/2$. So for any $r > 0$, $\alpha_k = \Omega(k)$.

Second, we show (19), $\alpha_k(\alpha_k - 1) \leq \alpha_{k-1}^2$ for $k \geq 1$. When $k$ is odd, by construction (19) holds. When $k \geq 2$ and $k$ is even, $k-1$ is odd and $k-2$ is even. Then $\alpha_k = (k+r)/r$ and $\alpha_{k-1} = (1 + \sqrt{1 + 4\alpha_{k-2}^2})/2 = (1 + \sqrt{r^2 + 4(k + r - 2)^2}/r)/2$. Simple algebra yields

$$\alpha_k(\alpha_k - 1) - \alpha_{k-1}^2 = \frac{4-r}{r^2}k - \frac{\sqrt{(k+r-2)^2 + r^2/4}}{r} - \frac{1}{2} - \left(\frac{r-2}{r}\right)^2. \tag{63}$$

When $r \geq 4$, (63) is negative for all $k$. For $0 < r < 4$, we further rearrange (63) as

$$\frac{(4-r)^2k^2 - r^2\left((k+r-2)^2 + \frac{r^2}{4}\right)}{r^2\left((4-r)k + r\sqrt{(k+r-2)^2 + \frac{r^2}{4}}\right)} - \frac{1}{2} - \left(\frac{r-2}{r}\right)^2$$

$$= -\frac{8(r-2)k^2 + 2r^2(r-2)k + r^2\left((r-2)^2 + \frac{r^2}{4}\right)}{r^2\left((4-r)k + r\sqrt{(k+r-2)^2 + \frac{r^2}{4}}\right)} - \frac{1}{2} - \left(\frac{r-2}{r}\right)^2,$$

which remains negative for $2 \le r < 4$ but blows up to $\infty$ as $k \to \infty$ if $0 < r < 2$.

Third, we calculate $\lim_k k(1 - \sigma_{k+1})$ along $\{k'\} = \{2k\}$ and $\{k''\} = \{2k+1\}$. By the definition of $\sigma_{k+1}$ we have

$$\lim_{k \to \infty} k(1 - \sigma_{k+1}) = \lim_{k \to \infty} k \left( 1 - \frac{\alpha_k - 1}{\alpha_{k+1}} \right) = \lim_{k \to \infty} \frac{k}{\alpha_{k+1}} (\alpha_{k+1} - \alpha_k + 1). \tag{64}$$

Along $\{k'\} = \{2k\}$,

$$\lim_{k \to \infty} \frac{2k}{\alpha_{2k+1}} = \lim_{k \to \infty} \frac{4k}{1 + \sqrt{1 + 4\alpha_{2k}^2}} = \lim_{k \to \infty} \frac{4k}{2\alpha_{2k}} = \lim_{k \to \infty} \frac{4k}{2(2k+r)/r} = r,$$

and

$$\begin{aligned}
\lim_{k \to \infty} \alpha_{2k+1} - \alpha_{2k} &= \lim_{k \to \infty} \frac{1 + \sqrt{1 + 4\alpha_{2k}^2}}{2} - \alpha_{2k} \\
&= \lim_{k \to \infty} \frac{1}{2} \left( 1 + 2\alpha_{2k} \sqrt{1 + \frac{1}{4\alpha_{2k}^2}} \right) - \alpha_{2k} \\
&= \lim_{k \to \infty} \frac{1}{2} + \alpha_{2k} \left( 1 + O(\frac{1}{\alpha_{2k}^2}) \right) - \alpha_{2k} \\
&= \frac{1}{2}.
\end{aligned}$$

Using (64) we have

$$\lim_{k' \to \infty} k'(1 - \sigma_{k'+1}) = r(1 + 1/2) = 3r/2. \tag{65}$$

Along $\{k''\} = \{2k+1\}$,

$$\lim_{k \to \infty} \frac{2k+1}{\alpha_{2k+2}} = \lim_{k \to \infty} \frac{2k+1}{(2k+2+r)/r} = r,$$

and

$$\begin{aligned}
\lim_{k \to \infty} \alpha_{2k+2} - \alpha_{2k+1} &= \lim_{k \to \infty} \alpha_{2k+2} - \frac{1 + \sqrt{1 + 4\alpha_{2k}^2}}{2} \\
&= \lim_{k \to \infty} \alpha_{2k+2} - \frac{1}{2} \left( 1 + 2\alpha_{2k} \sqrt{1 + \frac{1}{4\alpha_{2k}^2}} \right) \\
&= \lim_{k \to \infty} \alpha_{2k+2} - \frac{1}{2} - \alpha_{2k} \left( 1 + O(\frac{1}{\alpha_{2k}^2}) \right) \\
&= \lim_{k \to \infty} \frac{2k+2+r}{r} - \frac{2k+r}{r} - \frac{1}{2} \\
&= \frac{2}{r} - \frac{1}{2}.
\end{aligned}$$

Using (64) we have

$$\lim_{k'' \to \infty} k''(1 - \sigma_{k''+1}) = r \left( \frac{2}{r} - \frac{1}{2} + 1 \right) = 2 + \frac{r}{2}. \tag{66}$$

Lastly, combining the limits (65) and (66) gives $\lim_k \sigma_{k+1} = 1$, which implies that $\lim_k \alpha_{k+1}/\alpha_k = 1$. Therefore, condition (ii) holds. The proof of Lemma 1 is completed.

# E   Technical details in Section 4

## E.1   Convergence of ODEs (23) and (25)

In the convex setting where $f \in \mathcal{F}^1$, the convergence rates of gradient flow (23) and the ODE (24) for NAG-C are proved in Su et al. (2016). For completeness, we present the proofs for the convergence rates stated in the strongly convex setting where $f \in \mathcal{S}_\mu^1$.

For gradient flow (23), consider the Lyapunov function $V_t = f(X_t) - f^\star$, i.e., the potential gap itself. Then by (23), we have $\dot{V}_t = \langle \nabla f(X_t), \dot{X}_t \rangle = -\|\nabla f(X_t)\|^2$. By strong convexity, $f(X_t) - f^\star \leq \frac{1}{2\mu} \|\nabla f(X_t)\|^2$. Therefore, $\dot{V}_t \leq -2\mu(f(X_t) - f^\star) = -2\mu V_t$, which implies that $f(X_t) - f^\star = V_t \leq V_0 \cdot e^{-2\mu t} = e^{-2\mu t}(f(x_0) - f^\star)$.

For the ODE (25) corresponding to NAG-SC (4), consider the Lyapunov function $V_t = f(X_t) - f^\star + \frac{1}{2}\|\dot{X}_t + \sqrt{\mu}(X_t - x^\star)\|^2$. Then $V_0 = f(x_0) - f^\star + \frac{\mu}{2}\|x_0 - x^\star\|^2 \leq 2(f(x_0) - f^\star)$ by strong convexity. Using the ODE (25), we have

$$
\begin{aligned}
\frac{\mathrm{d}V_t}{\mathrm{d}t} &= \langle \nabla f(X_t), \dot{X}_t \rangle + \langle \dot{X}_t + \sqrt{\mu}(X_t - x^\star), \ddot{X}_t + \sqrt{\mu}\dot{X}_t \rangle \\
&= \langle \nabla f(X_t), \dot{X}_t \rangle - \langle \dot{X}_t + \sqrt{\mu}(X_t - x^\star), \sqrt{\mu}\dot{X}_t + \nabla f(X_t) \rangle \\
&= -\sqrt{\mu}\|\dot{X}_t\|^2 - \sqrt{\mu}\langle \dot{X}_t, \sqrt{\mu}(X_t - x^\star) \rangle - \sqrt{\mu}\langle X_t - x^\star, \nabla f(X_t) \rangle,
\end{aligned}
$$

which together with the inequality $\langle X_t - x^\star, \nabla f(X_t) \rangle \geq f(X_t) - f^\star + \frac{\mu}{2}\|X_t - x^\star\|^2$ by $\mu$-strong convexity suggests that

$$
\frac{\mathrm{d}V_t}{\mathrm{d}t} \leq -\frac{\sqrt{\mu}}{2}\|\dot{X}_t\|^2 - \sqrt{\mu}\left(f(X_t) - f^\star + \frac{1}{2}\|\dot{X}_t + \sqrt{\mu}(X_t - x^\star)\|^2\right) \leq -\sqrt{\mu}V_t.
$$

Hence, $f(X_t) - f^\star \leq V_t \leq V_0 \cdot e^{-\sqrt{\mu}t} \leq 2e^{-\sqrt{\mu}t}(f(x_0) - f^\star)$.

## E.2   Proof of Proposition 3

For the existence and uniqueness of the solution $X_t$ to the ODE (26), define

$$
Y_t = \begin{bmatrix} Y_{t,1} \\ Y_{t,2} \end{bmatrix} = \begin{bmatrix} X_t \\ \dot{X}_t \end{bmatrix} \in \mathbb{R}^{2n}.
$$

Then the original ODE can be reformulated as

$$
\dot{Y}_t = \begin{bmatrix} Y_{t,2} \\ -c_1\sqrt{\mu}Y_{t,2} - c_0\nabla f(Y_{t,1}) \end{bmatrix} = F(t, Y_t),
$$

with initial condition $Y_0 = (x_0^{\mathrm{T}}, 0)^{\mathrm{T}}$, where

$$
F(t, Y) = \begin{bmatrix} Y_2 \\ -c_1\sqrt{\mu}Y_2 - c_0\nabla f(Y_1) \end{bmatrix}.
$$

Because $f$ is $L$-smooth, $\nabla f$ is $L$-Lipschitz continuous. Hence, $F$ is Lipschitz continuous in $Y$. Because $F$ does not depend on $t$, $F$ is globally Lipschitz continuous in $Y$. By Picard–Lindelöf Theorem, the conclusion follows.

For the convergence rate, consider the Lyapunov function

$$
V_t = c_0(f(X_t) - f^\star) + \frac{1}{2}\|\dot{X}_t + \lambda\sqrt{\mu}(X_t - x^\star)\|^2,
$$

where $\lambda$ is to be chosen later. Using the ODE (26), we have

$$
\frac{\mathrm{d}V_t}{\mathrm{d}t} = c_0\langle \nabla f(X_t), \dot{X}_t \rangle + \langle \dot{X}_t + \lambda\sqrt{\mu}(X_t - x^\star), \ddot{X}_t + \lambda\sqrt{\mu}\dot{X}_t \rangle
$$

$$\begin{aligned}
&= c_0 \langle \nabla f(X_t), \dot{X}_t \rangle + \langle \dot{X}_t + \lambda\sqrt{\mu}(X_t - x^\star), -(c_1 - \lambda)\sqrt{\mu}\dot{X}_t - c_0\nabla f(X_t) \rangle \\
&= -(c_1 - \lambda)\sqrt{\mu}\|\dot{X}_t\|^2 - (c_1 - \lambda)\sqrt{\mu}\langle \dot{X}_t, \lambda\sqrt{\mu}(X_t - x^\star)\rangle - c_0\lambda\sqrt{\mu}\langle X_t - x^\star, \nabla f(X_t)\rangle \\
&= -\frac{c_1 - \lambda}{2}\sqrt{\mu}\|\dot{X}_t\|^2 - \frac{c_1 - \lambda}{2}\sqrt{\mu}\|\dot{X}_t + \lambda\sqrt{\mu}(X_t - x^\star)\|^2 + \frac{c_1 - \lambda}{2}\lambda^2\mu^{\frac{3}{2}}\|X_t - x^\star\|^2 \\
&\quad - c_0\lambda\sqrt{\mu}\langle X_t - x^\star, \nabla f(X_t)\rangle \\
&= -C\sqrt{\mu}V_t - \frac{c_1 - \lambda}{2}\sqrt{\mu}\|\dot{X}_t\|^2 - \left(\frac{c_1 - \lambda}{2} - \frac{C}{2}\right)\sqrt{\mu}\|\dot{X}_t + \lambda\sqrt{\mu}(X_t - x^\star)\|^2 \\
&\quad + Cc_0\sqrt{\mu}(f(X_t) - f^\star) + \frac{c_1 - \lambda}{2}\lambda^2\mu^{\frac{3}{2}}\|X_t - x^\star\|^2 - \lambda c_0\sqrt{\mu}\langle X_t - x^\star, \nabla f(X_t)\rangle,
\end{aligned}$$

which together with the inequality $\langle X_t - x^\star, \nabla f(X_t)\rangle \geq f(X_t) - f^\star + \frac{\mu}{2}\|X_t - x^\star\|^2$ by $\mu$-strong convexity suggests that

$$\begin{aligned}
\frac{\mathrm{d}V_t}{\mathrm{d}t} &\leq -C\sqrt{\mu}V_t - \frac{c_1 - \lambda}{2}\sqrt{\mu}\|\dot{X}_t\|^2 - \left(\frac{c_1 - \lambda}{2} - \frac{C}{2}\right)\sqrt{\mu}\|\dot{X}_t + \lambda\sqrt{\mu}(X_t - x^\star)\|^2 \\
&\quad - (\lambda - C)c_0\sqrt{\mu}(f(X_t) - f^\star) - \frac{\lambda\mu^{\frac{3}{2}}}{2}(c_0 - (c_1 - \lambda)\lambda)\|X_t - x^\star\|^2.
\end{aligned}$$

Because $c_0 > 0$ and $c_1 > 0$, we pick $\lambda$ such that $0 < \lambda < c_1$ and $(c_1 - \lambda)\lambda \leq c_0$. Moreover, we pick $C$ such that $0 < C \leq (c_1 - \lambda) \wedge \lambda$. Then

$$\frac{\mathrm{d}V_t}{\mathrm{d}t} \leq -C\sqrt{\mu}V_t \implies V_t \leq V_0 \cdot e^{-C\sqrt{\mu}t}.$$

In particular, if $c_1^2 \leq 4c_0$, we pick $\lambda = c_1/2$ and $C = c_1/2$. If $c_1^2 > 4c_0$, we pick $\lambda = \frac{c_1 + \sqrt{c_1^2 - 4c_0}}{2}$ and $C = \frac{c_1 - \sqrt{c_1^2 - 4c_0}}{2}$. Equivalently, $C = \frac{c_1 - \sqrt{(c_1^2 - 4c_0)\vee 0}}{2} > 0$. The conclusion then follows by bounding $V_0 = c_0(f(x_0) - f^\star) + \frac{\lambda^2\mu}{2}\|x_0 - x^\star\|^2 \leq (c_0 + \lambda^2)(f(x_0) - f^\star)$ because $\frac{\mu}{2}\|x_0 - x^\star\|^2 \leq f(x_0) - f^\star$.

When $c_0 \leq 0$ or $c_1 \leq 0$, $f(X_t) - f^\star$ is not guaranteed to converge. A simple counterexample is the harmonic oscillator, with $f(x) = \mu x^2/2$ for $x \in \mathbb{R}$,

$$\ddot{X}_t + c_1\sqrt{\mu}\dot{X}_t + c_0\mu X_t = 0, \tag{67}$$

starting from the initial position $X(0) = x_0$ with velocity $\dot{X}(0) = 0$. Since (67) is a second-order linear ODE with constant coefficients, its general solutions admit closed forms. Consider the characteristic equation $w^2 + c_1\sqrt{\mu}w + c_0\mu = 0$ with discriminant $\Delta = (c_1^2 - 4c_0)\mu$.

If $c_1^2 = 4c_0$, then there are two identical real roots $w = w_1 = w_2 = -c_1\sqrt{\mu}/2$, and $X_t = (\alpha_0 + \alpha_1 t)e^{wt}$ for some real numbers $\alpha_0$ and $\alpha_1$. By the initial condition $\dot{X}(0) = 0$, we find $\alpha_1 = -\alpha_0 w$. Then $X_t = \alpha_0(1 - wt)e^{wt}$. To achieve $X_t \to 0$, we need $w < 0$, which means $c_1 > 0$ and $c_0 = c_1^2/4 > 0$ too.

If $c_1^2 > 4c_0$, then there are two distinct real roots $w_1 = \frac{-c_1 + \sqrt{c_1^2 - 4c_0}}{2}\sqrt{\mu}$ and $w_2 = \frac{-c_1 - \sqrt{c_1^2 - 4c_0}}{2}\sqrt{\mu}$, with $w_1 > w_2$, and $X_t = \alpha_0 e^{w_1 t} + \alpha_1 e^{w_2 t}$. By the initial condition $\dot{X}(0) = 0$, we find $\alpha_0 w_1 + \alpha_1 w_2 = 0$. If $w_2 = 0$, then $w_1 > 0$ and hence $\alpha_0 = 0$ and $X_t \equiv \alpha_1 = x_0$, which contradicts $X_t \to 0$ for any $x_0 \neq 0$. If $w_2 \neq 0$, then $\alpha_1 = -\alpha_0 w_1/w_2$, and $X_t = \alpha_0(e^{w_1 t} - \frac{w_1}{w_2}e^{w_2 t})$ with $w_1 \neq w_2$. To achieve $X_t \to 0$, we also need $w_1 < 0$. Then $-c_1 + \sqrt{c_1^2 - 4c_0} < 0$, which implies that $c_0 > 0$ and $c_1 > 0$.

If $c_1^2 < 4c_0$, then there are two complex roots $w_{1,2} = \frac{-c_1 \pm \sqrt{4c_0 - c_1^2}i}{2}\sqrt{\mu}$, and $X_t$ is a linear combination of $e^{-c_1\sqrt{\mu}t/2} \times \sin(\sqrt{\mu(4c_0 - c_1^2)}t/2)$ and $e^{-c_1\sqrt{\mu}t/2} \times \cos(\sqrt{\mu(4c_0 - c_1^2)}t/2)$. We need $c_1 > 0$ to make $X_t \to 0$. Then $c_0 > c_1^2/4 > 0$ too.

### E.3 Proof of Proposition 4

The existence and uniqueness of a solution were proved in Shi et al. (2022), Proposition 1, and thus omitted. Below, we focus on the convergence rates. For (27), we consider the Lyapunov function

$$V_t = (1 + \sqrt{\mu s})(f(X_t) - f^\star) + \frac{1}{2}\|\dot{X}_t + \sqrt{\mu}(X_t - x^\star) + \sqrt{s}\nabla f(X_t)\|^2.$$

Using the ODE (27), we have

$$\frac{dV_t}{dt} = (1 + \sqrt{\mu s})\langle \nabla f(X_t), \dot{X}_t \rangle + \langle \dot{X}_t + \sqrt{\mu}(X_t - x^\star) + \sqrt{s}\nabla f(X_t), \ddot{X}_t + \sqrt{\mu}\dot{X}_t + \sqrt{s}\nabla^2 f(X_t)\dot{X}_t \rangle$$

$$= (1 + \sqrt{\mu s})\langle \nabla f(X_t), \dot{X}_t \rangle + \langle \dot{X}_t + \sqrt{\mu}(X_t - x^\star) + \sqrt{s}\nabla f(X_t), -\sqrt{\mu}\dot{X}_t - (1 + \sqrt{\mu s})\nabla f(X_t) \rangle$$

$$= -\sqrt{\mu}\|\dot{X}_t\|^2 - \sqrt{\mu}\langle \dot{X}_t, \sqrt{\mu}(X_t - x^\star) + \sqrt{s}\nabla f(X_t) \rangle - (1 + \sqrt{\mu s})\langle \nabla f(X_t), \sqrt{\mu}(X_t - x^\star) + \sqrt{s}\nabla f(X_t) \rangle$$

$$= -\frac{\sqrt{\mu}}{2}\|\dot{X}_t\|^2 - \frac{\sqrt{\mu}}{2}\|\dot{X}_t + \sqrt{\mu}(X_t - x^\star) + \sqrt{s}\nabla f(X_t)\|^2 + \frac{\sqrt{\mu}}{2}\|\sqrt{\mu}(X_t - x^\star) + \sqrt{s}\nabla f(X_t)\|^2$$
$$- (1 + \sqrt{\mu s})\sqrt{\mu}\langle X_t - x^\star, \nabla f(X_t) \rangle - (1 + \sqrt{\mu s})\sqrt{s}\|\nabla f(X_t)\|^2$$

$$= -\sqrt{\mu}V_t + \sqrt{\mu}(1 + \sqrt{\mu s})(f(X_t) - f^\star) - \frac{\sqrt{\mu}}{2}\|\dot{X}_t\|^2 + \frac{\mu^{\frac{3}{2}}}{2}\|X_t - x^\star\|^2 - \sqrt{\mu}\langle X_t - x^\star, \nabla f(X_t) \rangle$$
$$- \sqrt{s}(1 + \frac{\sqrt{\mu s}}{2})\|\nabla f(X_t)\|^2.$$

By the strong convexity, $\langle X_t - x^\star, \nabla f(X_t) \rangle \geq f(X_t) - f^\star + \frac{\mu}{2}\|X_t - x^\star\|^2$. Then the above display yields

$$\frac{dV_t}{dt} \leq -\sqrt{\mu}V_t + \sqrt{\mu}\sqrt{\mu s}(f(X_t) - f^\star) - \sqrt{s}(1 + \frac{\sqrt{\mu s}}{2})\|\nabla f(X_t)\|^2.$$

By the strong convexity again, we have $f(X_t) - f^\star \leq \frac{1}{2\mu}\|\nabla f(X_t)\|^2$. Hence

$$\frac{dV_t}{dt} \leq -\sqrt{\mu}V_t - \frac{\sqrt{s}}{2}(1 + \sqrt{\mu s})\|\nabla f(X_t)\|^2 \leq -\sqrt{\mu}V_t.$$

The conclusion follows by integrating with $t$.

For (28), we consider the Lyapunov function

$$V_t = (1 + \sqrt{\mu s})(f(X_t) - f^\star) + \frac{1}{2}\|\dot{X}_t + \sqrt{\mu}(X_t - x^\star)\|^2.$$

Using the ODE (28), we have

$$\frac{dV_t}{dt} = (1 + \sqrt{\mu s})\langle \nabla f(X_t), \dot{X}_t \rangle + \langle \dot{X}_t + \sqrt{\mu}(X_t - x^\star), \ddot{X}_t + \sqrt{\mu}\dot{X}_t \rangle$$

$$= (1 + \sqrt{\mu s})\langle \nabla f(X_t), \dot{X}_t \rangle - \langle \dot{X}_t + \sqrt{\mu}(X_t - x^\star), \sqrt{\mu}\dot{X}_t + (1 + \sqrt{\mu s})\nabla f(X_t) \rangle$$

$$= -\sqrt{\mu}\|\dot{X}_t\|^2 - \sqrt{\mu}\langle \dot{X}_t, \sqrt{\mu}(X_t - x^\star) \rangle - \sqrt{\mu}(1 + \sqrt{\mu s})\langle X_t - x^\star, \nabla f(X_t) \rangle$$

$$= -\frac{\sqrt{\mu}}{2}\|\dot{X}_t\|^2 - \frac{\sqrt{\mu}}{2}\|\dot{X}_t + \sqrt{\mu}(X_t - x^\star)\|^2 + \frac{\mu^{\frac{3}{2}}}{2}\|X_t - x^\star\|^2 - \sqrt{\mu}(1 + \sqrt{\mu s})\langle X_t - x^\star, \nabla f(X_t) \rangle$$

$$\leq -\sqrt{\mu}V_t + \sqrt{\mu}(1 + \sqrt{\mu s})(f(X_t) - f^\star) + \frac{\mu^{\frac{3}{2}}}{2}\|X_t - x^\star\|^2 - \sqrt{\mu}(1 + \sqrt{\mu s})\langle X_t - x^\star, \nabla f(X_t) \rangle.$$

By the strong convexity, $\langle X_t - x^\star, \nabla f(X_t) \rangle \geq f(X_t) - f^\star + \frac{\mu}{2}\|X_t - x^\star\|^2$. Then the above display yields

$$\frac{dV_t}{dt} \leq -\sqrt{\mu}V_t - \frac{\mu^2\sqrt{s}}{2}\|X_t - x^\star\|^2 \leq -\sqrt{\mu}V_t.$$

The conclusion follows by integrating with $t$.

### E.4 Proof of Proposition 5

The construction of our continuous-time Lyapunov function is motivated by Su et al. (2016) and Shi et al. (2022). We consider the auxiliary-energy term defined as

$$V_t^A = \frac{1}{2} \left\| r(X_t - x^\star) + t\left( \dot{X}_t + \frac{\beta\sqrt{s}}{\gamma} \nabla f(X_t) \right) \right\|^2.$$

Using the ODE (31), for $t \geq t_0$ we have

$$\frac{\mathrm{d}V_t^A}{\mathrm{d}t} = \left\langle r(X_t - x^\star) + t\left(\dot{X}_t + \frac{\beta\sqrt{s}}{\gamma}\nabla f(X_t)\right), r\dot{X}_t + \dot{X}_t + \frac{\beta\sqrt{s}}{\gamma}\nabla f(X_t) + t\left(\ddot{X}_t + \frac{\beta\sqrt{s}}{\gamma}\nabla^2 f(X_t)\dot{X}_t\right) \right\rangle$$

$$= \left\langle r(X_t - x^\star) + t\left(\dot{X}_t + \frac{\beta\sqrt{s}}{\gamma}\nabla f(X_t)\right), -\left(t + \left(\frac{r+1}{2} - \frac{\beta}{\gamma}\right)\sqrt{s}\right)\nabla f(X_t) \right\rangle$$

$$= -r\left(t + \left(\frac{r+1}{2} - \frac{\beta}{\gamma}\right)\sqrt{s}\right)\langle X_t - x^\star, \nabla f(X_t)\rangle - t\left(t + \left(\frac{r+1}{2} - \frac{\beta}{\gamma}\right)\sqrt{s}\right)\langle \dot{X}_t, \nabla f(X_t)\rangle$$

$$- \frac{\beta\sqrt{s}}{\gamma}t\left(t + \left(\frac{r+1}{2} - \frac{\beta}{\gamma}\right)\sqrt{s}\right)\|\nabla f(X_t)\|^2.$$

Let $C \geq 0$ be a constant to be chosen later, and introduce a factor of $\frac{t+C\sqrt{s}}{t}$ for technical adjustment. Because $\frac{t+C\sqrt{s}}{t} = 1 + \frac{C\sqrt{s}}{t}$ is decreasing in $t$, we have

$$\frac{\mathrm{d}}{\mathrm{d}t}\left(\frac{t+C\sqrt{s}}{t}V_t^A\right) = \frac{\mathrm{d}}{\mathrm{d}t}\left(\frac{t+C\sqrt{s}}{t}\right) \cdot V_t^A + \frac{t+C\sqrt{s}}{t} \cdot \frac{\mathrm{d}V_t^A}{\mathrm{d}t} \leq \frac{t+C\sqrt{s}}{t} \cdot \frac{\mathrm{d}V_t^A}{\mathrm{d}t}$$

$$= -r\frac{t+C\sqrt{s}}{t}\left(t + \left(\frac{r+1}{2} - \frac{\beta}{\gamma}\right)\sqrt{s}\right)\langle X_t - x^\star, \nabla f(X_t)\rangle - (t + C\sqrt{s})\left(t + \left(\frac{r+1}{2} - \frac{\beta}{\gamma}\right)\sqrt{s}\right)\langle \dot{X}_t, \nabla f(X_t)\rangle \quad (68)$$

$$- \frac{\beta\sqrt{s}}{\gamma}(t + C\sqrt{s})\left(t + \left(\frac{r+1}{2} - \frac{\beta}{\gamma}\right)\sqrt{s}\right)\|\nabla f(X_t)\|^2.$$

To eliminate the term of $\langle \dot{X}_t, \nabla f(X_t)\rangle$, we define the potential-energy term as

$$V_t^P = (t + C\sqrt{s})\left(t + \left(\frac{r+1}{2} - \frac{\beta}{\gamma}\right)\sqrt{s}\right)(f(X_t) - f^\star).$$

Then

$$\frac{\mathrm{d}V_t^P}{\mathrm{d}t} = \left(2t + \left(\frac{r+1}{2} - \frac{\beta}{\gamma} + C\right)\sqrt{s}\right)(f(X_t) - f^\star) + (t + C\sqrt{s})\left(t + \left(\frac{r+1}{2} - \frac{\beta}{\gamma}\right)\sqrt{s}\right)\langle \nabla f(X_t), \dot{X}_t\rangle. \quad (69)$$

Define the continuous Lyapunov function as $V_t = V_t^P + \frac{t+C\sqrt{s}}{t}V_t^A$. Set $t_0^\star = t_0 \vee \{2|C|\sqrt{s}\} \vee \{2|\frac{r+1}{2} - \frac{\beta}{\gamma}|\sqrt{s}\} > 0$. Then for $t \geq t_0^\star$, we have $V_t \geq 0$ and

$$(t + C\sqrt{s})\left(t + \left(\frac{r+1}{2} - \frac{\beta}{\gamma}\right)\sqrt{s}\right) \geq \frac{t}{2} \cdot \frac{t}{2} = \frac{t^2}{4}. \quad (70)$$

Combining (68) and (69), we obtain

$$\frac{\mathrm{d}V_t}{\mathrm{d}t} \leq -r\frac{t+C\sqrt{s}}{t}\left(t + \left(\frac{r+1}{2} - \frac{\beta}{\gamma}\right)\sqrt{s}\right)\langle X_t - x^\star, \nabla f(X_t)\rangle + \left(2t + \left(\frac{r+1}{2} - \frac{\beta}{\gamma} + C\right)\sqrt{s}\right)(f(X_t) - f^\star)$$

$$- \frac{\beta\sqrt{s}}{\gamma}(t + C\sqrt{s})\left(t + \left(\frac{r+1}{2} - \frac{\beta}{\gamma}\right)\sqrt{s}\right)\|\nabla f(X_t)\|^2,$$

which together with the convexity inequality $\langle X_t - x^\star, \nabla f(X_t)\rangle \geq f(X_t) - f^\star$ suggests that for $r \geq 0$ and $t \geq t_0^\star$,

$$\frac{\mathrm{d}V_t}{\mathrm{d}t} \leq -\underbrace{\left((r-2)t + (r-1)C\sqrt{s} + (r-1+\frac{rC\sqrt{s}}{t})(\frac{r+1}{2} - \frac{\beta}{\gamma})\sqrt{s}\right)}_{\mathbf{I}_t} \cdot (f(X_t) - f^\star)$$
$$-\underbrace{\frac{\beta}{\gamma}(t + C\sqrt{s})(t + (\frac{r+1}{2} - \frac{\beta}{\gamma})\sqrt{s})}_{\mathbf{II}_t} \cdot \sqrt{s}\|\nabla f(X_t)\|^2.$$

From (70), if $\frac{\beta}{\gamma} \geq 0$, then $\mathbf{II}_t \geq \frac{\beta}{4\gamma}t^2 \geq 0$ for $t \geq t_0^\star$. It remains to deal with $\mathbf{I}_t$.

When $r > 2$, we set $C = 0$. Then $\mathbf{I}_t = (r-2)t + (r-1)(\frac{r+1}{2} - \frac{\beta}{\gamma})\sqrt{s} \geq 0$ for $t \geq -\frac{r-1}{r-2}(\frac{r+1}{2} - \frac{\beta}{\gamma})\sqrt{s}$. In this case, we pick $t_1 = t_0^\star \vee \{-\frac{r-1}{r-2}(\frac{r+1}{2} - \frac{\beta}{\gamma})\sqrt{s}\}$.

When $r = 2$, $\mathbf{I}_t = C\sqrt{s} + (1 + \frac{2C\sqrt{s}}{t})(\frac{3}{2} - \frac{\beta}{\gamma})\sqrt{s}$. We set the constant $C > \frac{\beta}{\gamma} - \frac{3}{2}$ (e.g., $\frac{\beta}{\gamma}$). Then when $t$ is large (e.g., $t \geq \frac{4\beta}{3\gamma}(\frac{\beta}{\gamma} - \frac{3}{2})\sqrt{s}$), $\mathbf{I}_t \geq 0$ always holds. In this case, we pick $t_1 = t_0^\star \vee \{\frac{4\beta}{3\gamma}(\frac{\beta}{\gamma} - \frac{3}{2})\sqrt{s}\}$.

To summarize, when $r \geq 2$ and $\frac{\beta}{\gamma} \geq 0$, there exist $C \geq 0$ and $t_1$ such that when $t \geq t_1$, $\frac{\mathrm{d}V_t}{\mathrm{d}t} \leq 0$. If further $\frac{\beta}{\gamma} > 0$, then $-\frac{\mathrm{d}V_t}{\mathrm{d}t} \geq \frac{\beta\sqrt{s}}{4\gamma}t^2\|\nabla f(X_t)\|^2$. The remaining proof is similar to the proof of Theorem 4, but in a continuous way. We sketch the reasoning below.

If $\frac{\mathrm{d}V_t}{\mathrm{d}t} \leq 0$, then for $t \geq t_1$,

$$V_{t_1} \geq V_t \geq V_t^P = (t + C\sqrt{s})(t + (\frac{r+1}{2} - \frac{\beta}{\gamma})\sqrt{s})(f(X_t) - f^\star) \geq \frac{t^2}{4}(f(X_t) - f^\star),$$

which implies that $f(X_t) - f^\star = O\left(\frac{V_{t_1}}{t^2}\right)$.

If further $-\frac{\mathrm{d}V_t}{\mathrm{d}t} \geq \frac{\beta\sqrt{s}}{4\gamma}t^2\|\nabla f(X_t)\|^2$ with $\frac{\beta}{\gamma} > 0$, then for $t > t_1$, we have $V_t \geq 0$, and

$$V_{t_1} \geq V_{t_1} - V_t = (-V_t) - (-V_{t_1}) = \int_{t_1}^t \frac{\mathrm{d}}{\mathrm{d}u}(-V_u)\mathrm{d}u \geq \frac{\beta\sqrt{s}}{4\gamma}\int_{t_1}^t u^2\|\nabla f(X_u)\|^2\mathrm{d}u$$
$$\geq \frac{\beta\sqrt{s}}{4\gamma}\inf_{t_1 \leq u \leq t}\|\nabla f(X_u)\|^2\int_{t_1}^t u^2\mathrm{d}u = \frac{\beta\sqrt{s}(t^3 - t_1^3)}{12\gamma}\inf_{t_1 \leq u \leq t}\|\nabla f(X_u)\|^2,$$

which implies that

$$\inf_{t_1 \leq u \leq t}\|\nabla f(X_u)\|^2 \leq \frac{12\gamma V_{t_1}}{\beta\sqrt{s}(t^3 - t_1^3)} = O\left(\frac{\gamma V_{t_1}}{\beta\sqrt{s}(t^3 - t_1^3)}\right).$$

## F  Technical details in Section 5

### F.1  Single-variable form for HAG (32)

We derive the single-variable form (34) for HAG. From (32) solving the first equation for $u_k$, and plugging the expressions of $u_k$ and $u_{k+1}$ into the second equation, we obtain

$$\frac{x_{k+2} - x_{k+1} + a_{k+1}\nabla f(x_{k+1})}{\sqrt{a_{k+1}b_{k+1}}} = (b_k-1)\frac{x_{k+1} - x_k + a_k\nabla f(x_k)}{\sqrt{a_k b_k}} - \sqrt{a_k b_k}\nabla f(x_k) - \phi_k(\nabla f(x_{k+1}) - \nabla f(x_k)).$$

Rearrange the above display to conclude.

### F.2 Verification of monotonicity condition for (39)

We verify that the monotonicity condition in Theorem 4 is satisfied by (39), i.e., HAG under the configuration (38). First, we notice that (39) can be put into (21) with

$$\beta_{k-1} = \frac{1}{\sigma_{k+1}}\left(c_2\sqrt{c_0} - \frac{c_0}{b_{k-1}}\right), \quad \gamma_k = c_0\left(\frac{1}{b_{k-1}} + \frac{1}{b_k}\right) - \sigma_{k+1}(\beta_k - \beta_{k-1}).$$

When $\alpha_k = \frac{k+r}{r}$, we have $\sigma_{k+1} = \frac{\alpha_k - 1}{\alpha_{k+1}} = \frac{k}{k+r+1}$, $b_{k-1} = 1 + \sigma_{k+1} = \frac{2k+r+1}{k+r+1}$, and $\frac{\alpha_{k+1}}{\alpha_k} = \frac{k+r+1}{k+r}$. Without loss of generality, take $c_0 = 1$. Substituting $\sigma_{k+1}$, $b_{k-1}$ and $b_k$ into the display above and after some algebra, we obtain

$$\begin{aligned}
\frac{\widetilde{\alpha}_k}{\alpha_k} &= \gamma_k\frac{\alpha_{k+1}}{\alpha_k} - \beta_k\left(\frac{\alpha_{k+1}}{\alpha_k} - 1\right) = \gamma_k\frac{k+r+1}{k+r} - \beta_k\frac{1}{k+r} \\
&= \frac{2k^2 + (7 - 2c_2 + 4r)k + 6 - 3c_2 + 7r - c_2r + 2r^2}{(k+r)(2k+r+3)},
\end{aligned}$$

which is monotone in $k$ when $k \geq K$ for some $K$ depending only on $c_2$ and $r$.

