# OpenReview forum: "Understanding Accelerated Gradient Methods: Lyapunov Analyses and Hamiltonian-Assisted Interpretations"
_TMLR — Accepted by TMLR_

### Review · Reviewer_X4M3 · 2025-10-28

**Summary Of Contributions:**

This paper investigates first-order accelerated optimization algorithms, focusing on how gradient correction affects acceleration. It introduces: 1. Two general algorithmic classes (for strongly and general convex functions) 2. New discrete Lyapunov analyses giving simple, unified sufficient conditions for acceleration 3. Comparative analysis with limiting ODEs (including high-resolution ones), highlighting mismatches between discrete and continuous behaviors. 4. A Hamiltonian-assisted gradient (HAG) interpretation that unifies the roles of momentum and gradient correction.

The central claim is that a gradient correction coefficient ≥ 1/2 is necessary and sufficient for accelerated convergence—an elegant, interpretable threshold connecting Nesterov’s and Heavy-ball methods.

**Audience:**

Yes

**Audience Explanation:**

This submission provide novel theoretical insights providing the first explicit quantitative condition for acceleration across both convex and strongly convex cases. It identifies the gradient correction term’s magnitude as a key mechanism behind Nesterov acceleration. The authors provided rigorous Lyapunov analysis and the proofs avoid reliance on LMI-based frameworks, showcasing a hand-crafted, conceptually clean Lyapunov construction. The results are broad—covering both constant and analytic parameter cases—and include clear boundary conditions for accelerated vs non-accelerated convergence. The comparison with high-resolution ODEs reveals a discrepancy: ODEs (even high-resolution ones) fail to explain acceleration fully. This observation is theoretically important for researchers relying on continuous analogies to justify discrete algorithmic acceleration. The authors provide the Hamiltonian-assisted Interpretation where the HAG formulation introduces a physically inspired perspective connecting optimization to mechanics. It provides a clear interpretation of gradient correction in terms of momentum updates within Hamiltonian dynamics. This submission is written in clarity and rigor. The exposition is methodical and well-structured, with explicit assumptions, notation, and stepwise derivations. And all the results (Propositions, Theorems, Corollaries) are logically interconnected and self-contained.

**Broader Impact Concerns:**

This is a pure theory submission. There is no broader impact concerns.

**Claims And Evidence:**

Yes

**Claims Explanation:**

The Theorems and propositions in this submission are precise, internally consistent, and compatible with existing theory. The analytical treatment is sophisticated yet careful about assumptions (analyticity, strong convexity, Lipschitz smoothness). The authors has presented all the mathematical proof and analysis in this submission.

**Requested Changes:**

Here is the changes that the authors should provided in the revision:

1. There are some limited practical implications in this submission. While the theoretical contributions are strong, the paper lacks practical algorithmic takeaways—e.g., how to design or tune new methods in practice. The numerical results (in Appendix E) are not discussed in the main text, limiting empirical validation.

2. There are some notation issues in this submission. The paper is notation-heavy, especially in Sections 2.2 and 3.2. It may overwhelm readers without strong mathematical backgrounds. The authors should introduce small motivating examples (e.g., 1D quadratic cases) before the general theorems could enhance readability.

3. There are some missing discussion of related works in this paper. Recent developments in control-theoretic optimization and variational perspectives (e.g., symplectic integrators, stochastic extensions) are mentioned but not deeply compared. The paper could better clarify how its results differ from or extend Shi et al. (2022) and Sanz Serna & Zygalakis (2021) beyond technical refinements.

4. Hamiltonian section should be expanded more in the paper. The HAG interpretation is promising but underdeveloped. A deeper analytical connection between HAG updates and discrete acceleration (perhaps via an energy decay analysis) would make this section more impactful.

5. The authors need to provide more clarity in physical interpretation. While the Hamiltonian framework provides intuition, the mapping between mechanical analogies (momentum, damping, potential) and optimization behavior could be elaborated.

---

> ### Author Response · Authors · 2025-12-30
> **Response to Reviewer X4M3 (Part 1/2)**
>
> We sincerely thank the reviewer for thoughtful and encouraging comments and valuable suggestions. Below we detail our response to the requested changes point by point.
>
> ## Requested Changes
> ### C1. Practical implications and empirical validation
> >There are some limited practical implications in this submission. While the theoretical contributions are strong, the paper lacks practical algorithmic takeaways—e.g., how to design or tune new methods in practice. The numerical results (in Appendix E) are not discussed in the main text, limiting empirical validation.
>
> Our work aims to develop further understanding of the acceleration phenomenon and, in this process, yields some practical implications (although this is not our focus). First, there are general classes of algorithms (not just individual existing ones) which are shown to achieve similar acceleration rates as the popular accelerated algorithms. This allows practitioners more flexibility to tune their algorithms. Moreover, our main theoretical results indicate that the choice of the gradient correction coefficient is important in affecting the performance of the algorithms. This may serve as a theoretical basis for carefully tuning gradient correction when applying gradient methods in practice. For empirical validation, we have moved the numerical parts from the appendix to the main paper, following the main theoretical results in Sections 2.2 and 3.2. The numerical study provides evidence in support of our results.
>
> ### C2. Notation issue and motivating examples
> >There are some notation issues in this submission. The paper is notation-heavy, especially in Sections 2.2 and 3.2. It may overwhelm readers without strong mathematical backgrounds. The authors should introduce small motivating examples (e.g., 1D quadratic cases) before the general theorems could enhance readability.
>
> We thank the reviewer for the constructive suggestion.
>
> - On the notation issues, we have made the following changes in the revised paper:
> 	- We now describe necessary notation before presenting the two featured results, Propositions 1 and 2, in the Introduction section.
> 	- We add two tables at the beginning of Sections 2.2 and 3.2 to summarize the coverage of existing algorithms (e.g., NAG and HB) by our analyses. These tables give a quick overview to readers on the generality of the theoretical analysis proposed in this paper.
> 	- We add a notation table as new Appendix B to describe the notation used in a logical order. The notation table is also referred to at the end of the Introduction section.
>
> - As motivating examples in the Introduction section, we provide numerical illustrations for the effect of choosing the coefficient of gradient correction on convergence with other algorithmic parameters fixed.  See Figure 1 in the revised paper. The numerical results show that in both strongly convex and convex settings, choosing gradient correction as indicated by our theoretical results significantly improves convergence.
> ### C3 Connection with Shi et al. (2022) and Sanz Serna & Zygalakis (2021)
> > There are some missing discussion of related works in this paper. Recent developments in control-theoretic optimization and variational perspectives (e.g., symplectic integrators, stochastic extensions) are mentioned but not deeply compared. The paper could better clarify how its results differ from or extend Shi et al. (2022) and Sanz Serna & Zygalakis (2021) beyond technical refinements.
>
> We thank the reviewer for the constructive suggestions that greatly help to improve the clarity of the paper. In the revised paper, we have added a new section (Appendix A) for comparing our work with Shi et al. (2022), Sanz Serna & Zygalakis (2021), and the subsequent work by Dobson et al. (2025). In the introduction (page 2), Appendix A is pointed to before the summary of our main contributions.
>
> ### References mentioned
> - Sanz Serna, J. M., & Zygalakis, K. C. (2021). The connections between Lyapunov functions for some optimization algorithms and differential equations. _SIAM Journal on Numerical Analysis_, _59_(3), 1542-1565.
> - Shi, B., Du, S. S., Jordan, M. I., & Su, W. J. (2022). Understanding the acceleration phenomenon via high-resolution differential equations. _Mathematical Programming_, _195_(1), 79-148.
> - Dobson, P., Sanz-Serna, J. M., & Zygalakis, K. C. (2025). On the connections between optimization algorithms, Lyapunov functions, and differential equations: Theory and insights. _SIAM Journal on Optimization_, _35_(1), 537-566.

---

> ### Author Response · Authors · 2025-12-30
> **Response to Reviewer X4M3 (Part 2/2)**
>
> ### C4 and C5 Deeper analysis on the Hamiltonian framework
> > 4.  Hamiltonian section should be expanded more in the paper. The HAG interpretation is promising but underdeveloped. A deeper analytical connection between HAG updates and discrete acceleration (perhaps via an energy decay analysis) would make this section more impactful.
> > 5. The authors need to provide more clarity in physical interpretation. While the Hamiltonian framework provides intuition, the mapping between mechanical analogies (momentum, damping, potential) and optimization behavior could be elaborated.
>
> We thank the reviewer for encouraging and insightful comments. We totally agree that the Hamiltonian framework is promising in providing a new perspective for understanding acceleration. Techniques from other fields such as physics and computational mathematics, including the energy decay analysis the reviewer has mentioned, might be useful in fulfilling this goal.
>
> With regard to the physical interpretations, our current work has provided some interpretations related to the Hamiltonian framework, in particular the interpretations of sufficient conditions for achieving acceleration by HAG as summarized at the beginning of Section 5.2. As the reviewer pointed out, there remain interesting questions such as how the mapping between mechanical analogies and optimization behavior can be further clarified. A substantive investigation of these questions is beyond the scope of the current paper and can be pursued in future work.

---

### Review · Reviewer_FgrT · 2025-12-16

**Summary Of Contributions:**

**Summary.**

First, the authors consider two parametric families of algorithms obtained by augmenting gradient descent with two additional terms: (i) a momentum-like term proportional to the difference between the current and previous iterates, and (ii) a correction term proportional to the difference between the current and previous gradients. For the first family, they derive parameter conditions that guarantee acceleration in the smooth, strongly convex setting. For the second family, they establish parameter conditions that yield acceleration in the smooth, convex setting.

Second, they analyze the limiting low- and high-resolution ODEs associated with these algorithm families and compare their convergence bounds with the discrete rates, noting that the ODE viewpoint does not directly determine the stepsize regimes that separate accelerated from non-accelerated behavior. They also observe that gradient-correction terms vanish in low-resolution limits and that, even when retained in high-resolution ODEs, the resulting continuous dynamics can converge at the same rate with or without the gradient-correction term.

Finally, the authors introduce and motivate the Hamiltonian-assisted gradient method (HAG), a parametric family of algorithms in the same spirit as the two families above. They show how HAG connects back to those families through specific parameter choices, and use this connection to translate the earlier acceleration conditions into corresponding constraints on HAG’s parameters.

**Strengths.**

- Common Lyapunov framework across several known schemes.

- ODE discussion usefully stresses what it cannot predict.

**Weaknesses/limitations.**

- Key arguments require substantial rereading/cross-referencing.

- The prose leans promotional rather than descriptive; it should be tightened to match the actual evidence.

- The framework-level Lyapunov treatment seems to trade off tightness: constants/results are weaker than what method-specific Lyapunov analyses typically provide.

**Additional Comments:**

It would be helpful to acknowledge the established line of work that leverages performance estimation problem (PEP) techniques to automatically construct and certify Lyapunov/potential-function analyses by solving convex semidefinite programs, often yielding tight convergence guarantees. This is closely related in spirit to your work and should be cited (and, ideally, briefly differentiated). Suggested references:

   - A. Taylor, B. Van Scoy, and L. Lessard, *“Lyapunov functions for first-order methods: Tight automated convergence guarantees”*, ICML 2018.

   - A. Taylor and F. Bach, *“Stochastic first-order method: Non-asymptotic and computer-aided analyses via potential functions”*, COLT 2019.

   - M. Upadhyaya, S. Banert, A. Taylor, and P. Giselsson, *“Automated tight Lyapunov analysis for first-order methods”*, Mathematical Programming, 2025, doi: 10.1007/s10107-024-02061-8.

**Audience:**

Yes

**Audience Explanation:**

This paper is primarily relevant to the convex optimization theory community, especially readers focused on first-order methods and worst-case complexity. It offers a Lyapunov-based synthesis that casts several classical schemes within a common set of parameter conditions separating accelerated from non-accelerated behavior, and it includes a few negative/limitation observations (notably around continuous-time viewpoints) that can be conceptually useful. For readers already familiar with tight, method-specific Lyapunov analyses and with worst-case optimal schemes, the main value here seems to be the common perspective and clarification, rather than new algorithmic design principles or improved guarantees; correspondingly, the framework-level treatment appears to trade some tightness in constants for generality. Overall, I expect the paper to be of interest to a focused subset of TMLR’s optimization audience.

**Broader Impact Concerns:**

I do not see any concerns regarding the ethical implications of this work that would require adding or expanding a Broader Impact Statement.

**Claims And Evidence:**

Yes

**Claims Explanation:**

The central claims appear supported by correct proofs. However, the presentation is not consistently clear (requiring substantial rereading and cross-referencing), contains minor typos, and would benefit from more measured language to better match the evidence; I further elaborate below.

**Requested Changes:**

1. [Minor] The term “unified” is used excessively throughout the manuscript (e.g., “unified condition”, “unified interpretations”); consider replacing it with more specific descriptors.

2. [Minor] I also recommend avoiding the term “simple”. What counts as “simple” is subjective, and in several places it can read as advertising rather than exposition. Where you intend to convey ease or accessibility, please replace it with objective descriptors or present the argument without a qualifier.

3. [Minor] The adjective “novel” is not necessarily overused here. However, any meaningful contribution should be novel by definition, so explicitly advertising “novelty” is redundant and can read as promotional.

4. [Minor] Please provide a link to the code used to generate the numerical results (including experiment and plotting scripts).

5. [Minor] Consider using “acceleration” (instead of “Nesterov’s acceleration”) throughout for more neutral terminology.

6. [Critical] Please add explicit sign/range conditions for all scalars appearing in formal statements (propositions, theorems, corollaries, lemmas), so each statement is well-posed without relying on surrounding text (e.g., $L>0$ rather than just $L$, and that $\\mu>0$ is such that $\\mu \leq L$).

7. [Critical] In Proposition 1, please define/introduce all symbols before first use (in a logically sequential order), in particular $f^{\\star}$, $x^{\\star}$, $s$, $L$, $\mu$, $c_{1}$, and $\\tilde{c}_{2}$. A similar issue appears in other propositions/theorems/lemmas/corollaries; please ensure each formal statement is self-contained with respect to notation.

8. [Minor] Use $\\star$ (\star) and not $\*$ throughout for optimality, e.g., "$f^{\*}$" $\\rightarrow$ "$f^{\\star}$", etc.

9. [Critical] In the smooth and (non-strongly) convex setting, several formal statements and discussions are phrased in terms of a minimizer $x^{\\star}$, which implicitly assumes that the minimum in Eq. (1) is finite and attained. Please make this explicit (in all relevant formal statements) by assuming
$$
\\mathop{\\mathrm{Argmin}}\\limits_{x\\in\\mathbb{R}^n} f(x) \\neq \\emptyset,
$$
and, for clarity, fixing notation by choosing
$$
x^{\\star} \\in \\mathop{\\mathrm{Argmin}}\\limits_{x\\in\\mathbb{R}^n} f(x),
$$
so that $f^{\\star}=f(x^{\\star})$ is well-defined. In the $\\mu$-strongly convex setting, existence (and uniqueness) of $x^{\\star}$ is implied, so no additional assumption is needed there; however, $x^{\\star}$ and $f^{\\star}$ should still be clearly defined when first introduced.

10. [Minor] Page 1: “In fact, Nesterov (1983) used a potential function or a Lyapunov function to establish the convergence of the first accelerated algorithm”. I recommend avoiding the word “first” here unless it is carefully qualified (as written, it reads as historically incorrect).

11. [Critical] When stating an algorithmic update rule, please explicitly specify the index range for which it is valid, and separately clarify any initial step conventions when they differ. This small bookkeeping detail makes the scheme unambiguous and easier for the reader to follow, especially when the $k=0$ and $k>0$ cases require different handling.

12. [Critical] I strongly recommend that whenever a function is introduced (e.g., in any formal statement), the authors explicitly state its domain and codomain, e.g., write $ f:\\mathbb{R}^{n}\\to\\mathbb{R}$ rather than referring to it only as $f$. This improves precision and makes the notation easier for the reader to parse consistently throughout the paper.

13. [Minor] There is a formatting typo on page 14.

14. [Critical] Appendix B.3: The “uniform” lower bound should take the minimum of the two possible limits of $\\omega_{k+1}$ (since $\\omega_{k+1}\\to 1$ or $\\omega_{k+1}\\to 1/\\gamma$), so replace $1/\\gamma \\vee 1$ with $1/\\gamma \\wedge 1$.

15. [Minor] Page 20, Section 6.2 “Step 3”: “As expected from (45), We further bound ...” → change “We” to “we”.

16. [Critical] Step 1 before Lemma 2, Section 6.1: There is a difference written with a missing square, i.e., $\\frac{1}{1-\\nu\\sqrt q}\\frac{\\mu}{2}\\|z_{k+1}-x^\\star\\| - \\frac{\\mu}{2}\\|z_k-x^\\star\\|^2$. The first norm should be squared, i.e., replace $\\|z_{k+1}-x^\\star\\|$ by $\\|z_{k+1}-x^\\star\\|^2$.

17. [Critical] Well-posedness of the ODEs (existence/uniqueness): In Section 4, multiple results refer to “the solution” of ODEs (23), (24), (26), (27), (28), (30), and (31) under assumptions such as $f\\in \\mathcal{F}^1$ or $f\\in \\mathcal{S}^1_\\mu$. Since $\\mathcal{F}^1$ here does not include Lipschitz continuity of $\\nabla f$, and since (24)/(30) have a $1/t$ singularity at $t=0$, it would help to add a brief remark (or citation) clarifying the existence/uniqueness (and whether global-in-time) of these ODE solutions under the stated assumptions, and how the initial condition at $t=0$ is interpreted for the singular-damping equations. If additional regularity assumptions are in fact needed for well-posedness, please state them explicitly in the corresponding formal statements (propositions, etc.).

18. [Critical] Note that the constant $C$ in Proposition 3 and in its proof differ by a factor $2$, which might confuse readers. Please update to make it consistent.

19. [Minor] Spelling: “Cauchy-Schwartz” should be “Cauchy-Schwarz”.

20. [Critical] In Sections 1–3, several results assert the existence of positive constants $C_0$ and $C_1$. In a number of cases, your proofs appear to yield explicit formulas for $C_0$ and/or $C_1$; where this is possible, it would be helpful to state those expressions explicitly in the corresponding theorem/proposition/corollary (or in a short remark).

21. [Minor] I suggest writing simply “convex” rather than “general convex” (and, if needed for contrast, “convex (not necessarily strongly convex)”). Motivation: “general convex” is largely redundant/nonstandard, since “convex” already covers the non-strongly-convex case.

---

> ### Author Response · Authors · 2025-12-30
> **Response to Reviewer FgrT (Part 1/2)**
>
> We sincerely thank the reviewer for thoughtful comments and valuable suggestions. Below we detail our response to the comments about weaknesses and the requested changes point by point. We only discuss the critical changes. All minor changes have been made in the revised paper.
>
> ## Weaknesses
> ### W1. Prose promotional rather than descriptive
> We have revised the paper throughout to use more descriptive phrases.
>
> ### W2. Tightness in constants/results
> To develop further understanding of the acceleration phenomenon, our work primarily focuses on Lyapunov analyses for broad classes of algorithms and, as a tradeoff, may provide weaker constants than method-specific results.
>
> ## Requested Changes
> ### C6. Explicit sign/range conditions
> >Please add explicit sign/range conditions for all scalars appearing in formal statements (propositions, theorems, corollaries, lemmas), so each statement is well-posed without relying on surrounding text (e.g., $L>0$ rather than just $L$, and that $\mu>0$ is such that $\mu\leq L$).
>
> We appreciate the reviewer’s constructive suggestion. In the revised paper we have added explicit sign/range conditions for all scalars appearing in formal statements.
>
> ### C7. Introduce all symbols before first use
> >In Proposition 1, please define/introduce all symbols before first use (in a logically sequential order), in particular $f^\star$, $x^\star$, $s$, $L$, $\mu$, $c_1$, and $\tilde c_2$. A similar issue appears in other propositions/ theorems/lemmas /corollaries; please ensure each formal statement is self-contained with respect to notation.
>
> We have introduced necessary notation before Propositions 1 and 2 in the revised paper. We have also checked notation in other formal statements.
>
> ### C9. Existence of the global minimizer
> >In the smooth and (non-strongly) convex setting, several formal statements and discussions are phrased in terms of a minimizer $x^\star$, which implicitly assumes that the minimum in Eq. (1) is finite and attained. Please make this explicit (in all relevant formal statements).
>
> As suggested, we have introduced notation before Proposition 1, where we explicitly assume the existence of a global minimizer of $f$ in the smooth and convex setting.
>
> ### C11. Index range in an algorithmic update rule
> >When stating an algorithmic update rule, please explicitly specify the index range for which it is valid, and separately clarify any initial step conventions when they differ. This small bookkeeping detail makes the scheme unambiguous and easier for the reader to follow, especially when the $k=0$ and $k>0$ cases require different handling.
>
> As suggested, we have explicitly specify the index range for $k$ in all algorithmic update rules.
>
> ### C12. Function domain and codomain
> >I strongly recommend that whenever a function is introduced (e.g., in any formal statement), the authors explicitly state its domain and codomain, e.g., write $f\colon \mathbb{R}^n\to\mathbb{R}$ rather than referring to it only as $f$. This improves precision and makes the notation easier for the reader to parse consistently throughout the paper.
>
> As suggested, we have explicitly stated the domain and codomain of the objective function $f$ in all formal statements.
>
> ### C14. The "uniform" lower bound in the proof
> >Appendix B.3: The “uniform” lower bound should take the minimum of the two possible limits of $\omega_{k+1}$
>
> We appreciate the reviewer’s careful reading and have corrected the typo in the revised paper.
>
> ### C16. A missing square in the difference of auxiliary energy
> >Step 1 before Lemma 2, Section 6.1: There is a difference written with a missing square, i.e., $\frac{1}{1-\nu\sqrt{q}}\frac{\mu}{2}\||z_{k+1}-x^\star\||-\frac{\mu}{2}\||z_k-x^\star\||^2$. The first norm should be squared, i.e., replace $\||z_{k+1}-x^\star\||$ by $\||z_{k+1}-x^\star\||^2$.
>
> We appreciate the reviewer’s careful reading and have corrected the typo in the revised paper.

---

> ### Author Response · Authors · 2025-12-30
> **Response to Reviewer FgrT (Part 2/2)**
>
> ### C17. Well-posedness of the ODEs
> >Well-posedness of the ODEs (existence/uniqueness): In Section 4, multiple results refer to “the solution” of ODEs (23), (24), (26), (27), (28), (30), and (31) under assumptions such as $f\in\mathcal{F}^1$ or $f\in\mathcal{S}^1_\mu$. Since $\mathcal{F}^1$ here does not include Lipschitz continuity of $\nabla f$, and since (24)/(30) have a $1/t$ singularity at $t=0$, it would help to add a brief remark (or citation) clarifying the existence/uniqueness (and whether global-in-time) of these ODE solutions under the stated assumptions, and how the initial condition at $t=0$ is interpreted for the singular-damping equations. If additional regularity assumptions are in fact needed for well-posedness, please state them explicitly in the corresponding formal statements (propositions, etc.).
>
> We thank the reviewer for raising this important point. In the revised paper, for all ODEs appearing in Section 4 (including those in text and formal statements), we explicitly state the smoothness condition and discuss the existence and uniqueness of the solution, by either providing proofs (e.g., Proposition 3) or pointing out direct references. The only exception is (31), whose well-posedness was partially established in Shi et al. (2022) and a full analysis is beyond the scope of this paper. Therefore, in Proposition 5 we modify the statement by assuming that a solution exists, and then state the convergence bound.
>
> ### References mentioned:
> - Shi, B., Du, S. S., Jordan, M. I., & Su, W. J. (2022). Understanding the acceleration phenomenon via high-resolution differential equations. _Mathematical Programming_, _195_(1), 79-148.
>
> ### C18. Constant C in Proposition 3 and its proof
> >Note that the constant $C$ in Proposition 3 and in its proof differ by a factor $2$, which might confuse readers. Please update to make it consistent.
>
> Thanks for catching the discrepancy. We have updated the proof of Proposition 3, which is now consistent with the statement in Proposition 3.
>
> ### C20. Explicit formulas for the constants
> >In Sections 1–3, several results assert the existence of positive constants $C_0$ and $C_1$. In a number of cases, your proofs appear to yield explicit formulas for $C_0$ and/or $C_1$; where this is possible, it would be helpful to state those expressions explicitly in the corresponding theorem/proposition/corollary (or in a short remark).
>
> We thank the reviewer for this excellent suggestion that would definitely enhance the paper's clarity.
> - The two constants $C_0$ and $C_1$ appearing in theoretical results for the strongly convex setting, including Proposition 1, Theorem 1~3, and Corollary 1, can be explicitly specified. However, due to the use of Taylor expansion in establishing the rates, such expressions for $C_0$ and $C_1$ are complicated. Therefore, we provide a discussion after Theorem 1 on $C_0$ and $C_1$, and refer interested readers to proofs in the appendix for details.
> - For the convex setting, the constant $C_0$ can be explicitly specified in a clean form, which is given in Proposition 1 and Theorem 4 respectively.
> - For Section 4 on ODEs, Proposition 5 contains $t_1$ whose expression is complicated. Similarly to the strongly convex setting mentioned above, we refer readers to the proofs in the appendix.
> ---
> ## Additional Comments
> ### The PEP (performance estimation problem) line of work
> We thank the reviewer for pointing out these important works.
> - We have now incorporated related references into the Introduction section, which briefly discusses the LMI/SDP techniques for automated Lyapunov analyses, including IQC and PEP methodologies.
> - We have added a new section (Appendix A) for a detailed comparison between our methodology and the LMI/SDP techniques. In the introduction (page 2), Appendix A is pointed to before the summary of our main contributions.

---

### Review · Reviewer_es1t · 2025-12-18

**Summary Of Contributions:**

# Summary Of Contributions

This paper introduces a class of accelerated gradient-type methods for smooth (strongly) convex optimization. The convergence analysis is conducted via a Lyapunov-based approach, yielding sufficient conditions on the algorithmic hyperparameters to achieve standard $\kappa$-dependent rates as well as accelerated $\sqrt{\kappa}$-dependent convergence.

In subsequent sections, the authors demonstrate that ODE-based analyses of accelerated methods fail to fully capture their discrete-time behavior. The paper further establishes connections between the proposed class of accelerated algorithms and Hamiltonian-inspired descent dynamics.

## More detailed summary

- The paper studies accelerated first-order methods for smooth convex and (strongly) convex problems. It introduces two classes of discrete algorithms:

    - In Section 2, the authors introduce a three-point scheme (10) for the smooth, strongly convex setting. This scheme is equivalent to the one-line formulation in (11). It is controlled by three nonnegative parameters $\eta,\nu,\tau$, and different choices of these parameters recover various existing optimization methods. For example, NAG-SC corresponds to (1,1,1), and TMM corresponds to (1,1,2). In general, the choice of those constants affects the ratio between gradient, momentum, and gradient correction terms in (11).

    - In Theorem 1, the authors establish convergence guarantees for optimiality gap $f(x_k)-f^*$ under certain step-size restrictions on $s$, together with a specific relation among the parameters $\eta,\nu,\tau$. Depending on how these conditions are instantiated, the resulting convergence rate can be either accelerated or non-accelerated.
    - Considering a one-line formulation, the relation between parameters $\eta,\nu,\tau$ is summarized in the choice of $c_1 > 2, \tilde{c}_2 \ge 0.5$ in Proposition 1.

    - In Theorem 3, the analysis is then extended to the setting in which $\eta,\nu,\tau$ are allowed to depend on $q=\mu s$. In this case, the constraints are imposed on the coefficients of the Taylor expansions of $\eta,\nu,\tau$ with respect to $q$ (note that under this more general parametrization, the number of admissible choices is smaller than in the previous, constant-parameter setting).

- Next, the authors switch to a smooth convex setting.
    - Similarly, in the general convex case, they define a class of algorithms (22) parameterized by sequences $\{\alpha_k\},\{\beta_k\}, \{\gamma_k\}$ that covers NAG-C, FISTA-type methods, and variants studied by [Shi et al. (2022)].
    - In Theorem 4, the authors provide a cubic rate $O(1/k^3)$ for the squared gradient norm under the restrictions on the step-size and the sequences above. The authors also provide examples when the restrictions on the sequences is satisfied.

- Comparing discrete algorithms to low- and high-resolution ODE limits, the paper argues that even high-resolution ODEs fail to fully explain the discrete dynamics. The authors claim that the gradient correction term is vital for achieving acceleration. However, in continuous time, its effect is neglected. Therefore, such models cannot always be used to inform whether the discrete counterpart achieves accelerated convergence.

- Finally, the authors propose a Hamiltonian-assisted gradient method (HAG) (Section 5), derived from minimizing a Hamiltonian $H(x,u) = f(x) + 0.5\|u\|^2$. They show that previously discussed algorithms can be seen as particular HAG discretizations, giving a more interpretable view of the role of momentum and gradient correction.

- In Appendix E, the authors numerical simulations to support their theoretical claims.


## Strengths

- To some extent, Corollary 1 gives interpretable conditions that are needed for achieving acceleration. The authors claim that if the momentum term ($c_1^2 > 4c_0$) and gradient correlation term ($c_2^2>c_0$) are strong enough, then the algorithm achieves accelerated convergence. If the momentum term is as strong as before, but the gradient correlation term is not ($c_0/4 < c_2^2 <c_0$), then the algorithm achieves only a non-accelerated rate.
- The authors provide a unified framework to study acceleration that also covers previously introduced algorithms as special cases.
- The work clearly shows that limiting ODEs (even high-resolution ones) can be misleading: they may accelerate under very weak constraints (e.g., $\beta/\gamma > 0$ in Proposition 5), while the discrete algorithms require a much stronger gradient-correction condition (e.g., $\beta/\gamma>0.5$ in Theorem 4) to achieve analogous rates.
- The HAG part of the work provides a mechanical/energy-based interpretation of the momentum (updates of $u$ term), gradient correction (some sort of additional "friction" term), and how parameter choices control damping regimes (over/critical/under).
- The proofs look correct to me; I did not find any critical mistakes.

## Weaknesses

- The authors claim to explain the acceleration mechanism, but their analysis only provides sufficient conditions for achieving acceleration. It remains unclear whether other parameter choices might also yield accelerated rates, or whether all remaining configurations always lead to non-accelerated convergence or divergence. The presentation would be stronger if the authors established some form of lower bounds or impossibility results for the other parameter regimes, demonstrating that their conditions indeed characterize all possible choices that enable acceleration. In the current form, I suggest reformulating the text to align with the actual results.
- The authors should also provide numerical evidence that they covered all possible configurations that lead to acceleration.
- I believe it is already known that ODE-based analysis does not recover well the discrete dynamics. What is really important is to show that ODE models are still good at approximating the discrete dynamics in practice or providing changes/modifications to ODE models so that they track discrete dynamics better. For example, [1] make changes to SDEs so that at least step-size restrictions are recovered. Can it be done in this work as well?

    [1] Compagnoni, Enea Monzio, et al. "On the Interaction of Noise, Compression Role, and Adaptivity under $(L_0, L_1) $-Smoothness: An SDE-based Approach." arXiv preprint arXiv:2506.00181 (2025).



- Experimental results in Figure 3 look strange to me. Algorithms fail to recover the original error. Why do we observe such weird dynamics?

- Overall, the paper is not well structured and is difficult to follow. The notation is extensive, and the reader is required to keep track of many symbols throughout the text. I recommend adding a notation table in the appendix to help with clarity. In addition, parts of the introduction should be rewritten to improve readability and correct several grammatical issues.

- In the convex case, the convergence rates are given for the gradient norm, while a more standard convergence measure is the function suboptimality. I encourage the authors to explain why it is the case.

**Audience:**

Yes

**Audience Explanation:**

Yes, this paper examines the acceleration of optimization algorithms, a crucial research topic as these techniques enable faster convergence and improved performance in many practical applications.

**Claims And Evidence:**

Yes

**Claims Explanation:**

The authors provide clear convergence statements and a discussion around the results. The convergence guarantees extend existing ones in the literature in a unified manner, providing more intuition on when the acceleration is possible. The results are made more interpretable using the HAG framework.

**Requested Changes:**

## Comments and Suggestions

- Add a table with the results that are derived in this paper in some special cases, and how they align with prior work. For example, for NAG, the paper recovers acceleration of prior work, but the Heavy-ball method is not covered by the framework at all. Such a table would help to give a quick overview of the generality and limitations of the proposed theoretical framework.

- I suggest moving some numerical results to the main part or at least providing links to them to increase the clarity of the work.

- In the introduction, the authors already use the term "gradient correction" without introducing it. In my opinion, such a term is not standard in the literature, and it should be defined.

- Please also check the suggestions in the previous sections.

- On page 3, the authors use the term "meaningful conditions". I suggest avoiding such wording without giving a precise description. The same applies to the wording "It seems to be an open" question

- Can the authors provide other examples in the convex case, where $\alpha_k$ is not necessarily linear? According to Proposition 2, we should have $\alpha_k = \Omega(k)$. Can we choose something else?

- Can the authors provide a discussion on the results in [2]? Does the current theoretical framework cover the results in [2]? I believe that the authors should include the discussion around [2] in the paper.

    [2] Goujaud, Baptiste, Adrien Taylor, and Aymeric Dieuleveut. "Provable non-accelerations of the heavy-ball method: B. Goujaud et al." Mathematical Programming (2025): 1-59.


- I suggest not using terms $\eta_0$, etc, in Theorem 1, as they are later used in Theorem 2 with a different meaning. I would simply say that $\eta$ is constant

- Is there any reason why the theoretical framework does not cover the original Heavy-ball method? Can the authors elaborate more on this point?

- I suggest adding the definitions of (strong) convexity and smoothness at least in the appendix and referring to them in the theorem statements.

---

> ### Author Response · Authors · 2025-12-30
> **Response to Reviewer es1t (Part 1/3)**
>
> We sincerely thank the reviewer for thoughtful comments and valuable suggestions. Below we detail our response to the comments about weaknesses and the requested changes point by point.
>
> ## Weaknesses
> ### W1. Necessary conditions for acceleration
> We thank the reviewer for raising this point. Our current analysis establishes only sufficient conditions for achieving acceleration. Whether these conditions are also necessary remains to be studied. In the revised paper, we have modified the related text to align with this. Particularly, the following is now stated after Proposition 2:
> >It remains an open question to study whether this condition is also necessary.
> ### W2. Numerical evidence for covering all possible configurations for acceleration
> We presented numerical results to illustrate that algorithms satisfying our sufficient conditions tend to converge faster than otherwise. It remains an open question whether these conditions are also necessary, that is, whether our results cover all possible configurations for acceleration. Both theoretical and numerical studies would need to be addressed in future work.
> ### W3. Modification of ODEs
> We thank the reviewer for pointing out an important line of work. A series of papers by Compagnoni (Compagnoni et al., 2025a, 2025b, 2025c) leverage stochastic differential equations (SDEs) to understand behaviors of distributed stochastic gradient methods. The core idea is to construct specific SDEs as order-$\alpha$ weak approximations for stochastic gradient algorithms. The SDEs faithfully represent the dynamics of the corresponding algorithms so that the convergence properties of SDEs provide illumination on the latter algorithms regarding the interplay between various factors, including adaptive learning rate, gradient noise, gradient compression etc. We have updated the introduction section to include the SDE line of work. Particularly, the following is now stated in Section 1:
> > Recently, stochastic differential equations (SDEs) are leveraged to understand the interplay between the learning rate, gradient noise, and gradient compression in distributed stochastic gradient methods (Compagnoni et al., 2025a; b).
>
> Back to our work, based on our analysis of strongly convex settings, the accelerated and non-accelerated rates of gradient algorithms boil down to the different magnitudes of the step size $s$ (same as the learning rate $\eta$ studied in the SDE line of work), where $s\asymp 1/L$ leads to acceleration and $s\asymp\mu/L^2$ leads to non-acceleration (see the discussion after Theorem 1 for details). One notable gap in using ODEs (including high-resolution ODEs) to explain acceleration is that the current ODEs fail to capture the magnitude of the step size. Motivated by the work of Compagnoni et al, it seems an interesting direction to exploit the idea of order-$\alpha$ weak approximations to modify the ODEs, such that its property is directly linked to step size. However, there are nontrivial differences between the SDE line of work and the relevant ODE analysis which need to be tackled in further investigation.
>
> 1. The SDE line of work is typically focused on the convergence of stochastic gradient methods. However, our ODE analysis is conducted to study accelerated convergence (not just convergence) of deterministic gradient methods.
>
> 2. In the SDE line of work, even a first-order SDE approximation of stochastic gradient methods explicitly contains the step size. We are not sure whether it is due to the inherent randomness of stochastic algorithms or the definition of order-$\alpha$ weak approximation. However, in analyzing deterministic gradient methods, the low-resolution ODEs do not explicitly contain the step size (unless a high-resolution ODE is considered).
>
> 3. The existing SDE framework successfully explains the step size restrictions (for stochastic gradient algorithms). However, the acceleration for minimizing convex (not necessarily strongly convex) objectives, unlike in the strongly convex setting, is not due to the magnitude of step size (for deterministic gradient algorithms).
>
> ### References mentioned:
> - Compagnoni, E. M., Islamov, R., Proske, F. N., & Lucchi, A. (2025a). Unbiased and Sign Compression in Distributed Learning: Comparing Noise Resilience via SDEs. In _The 28th International Conference on Artificial Intelligence and Statistics_.
> - Compagnoni, E. M., Liu, T., Islamov, R., Proske, F. N., Orvieto, A., & Lucchi, A. (2025b). Adaptive Methods through the Lens of SDEs: Theoretical Insights on the Role of Noise. In _The Thirteenth International Conference on Learning Representations_.
> - Compagnoni, E. M., Islamov, R., Orvieto, A., & Gorbunov, E. (2025c). On the Interaction of Noise, Compression Role, and Adaptivity under $(L_0, L_1)$-Smoothness: An SDE-based Approach. _arXiv preprint arXiv:2506.00181_.

---

> ### Author Response · Authors · 2025-12-30
> **Response to Reviewer es1t (Part 2/3)**
>
> ### W4. Explanation for the experimental results in Figure 3
> Experiments results in Figures 3 and 4 are in terms of the scaled optimiality gap $sk^2(f(x_k)-f^\star)$ and the scaled minimal squared gradient norm $s^2(k+1)^3\min_{0\leq i \leq k}\|\nabla f(x_i)\|^2$. The scaled error bounds imply $O(1/k^2)$ and $O(1/(k+1)^3)$ rates for the original optimaility gap $f(x_k)-f^\star$ and the minimal squared gradient norm $\min_{0\leq i \leq k}\|\nabla f(x_i)\|^2$, respectively. We are not sure what the reviewer means by "algorithms failing to recover the original error and having a weird dynamics".
>
> ###  W5. Readability and notation issues
> Motivated by the reviewer's comments, we have made the following changes in the revised paper:
> - We now describe necessary notation before presenting the two featured results, Propositions 1 and 2, in the Introduction section.
> - In the Introduction section, we provide numerical illustrations for the effect of choosing the coefficient of gradient correction on convergence with other algorithmic parameters fixed.  See Figure 1 in the revised paper. The numerical results show that in both strongly convex and convex settings, choosing gradient correction as indicated by our theoretical results significantly improves convergence.
> - We add two tables at the beginning of Sections 2.2 and 3.2 to summarize the coverage of existing algorithms (e.g., NAG and HB) by our analyses. These tables give a quick overview to readers on the generality of the theoretical analysis proposed in this paper.
> - We add a notation table as new Appendix B to describe the notation used in a logical order. The notation table is referred to at the end of the introduction section.
>
> ### W6. Convergence rates in terms of the gradient norm in the convex case
> In the convex case, our result (Theorem 4) for the discrete algorithms (20)-(22) actually covers both types of convergence rates: the inverse quadratic rate for the optimality gap in (16) and the inverse cubic rate for the gradient norm in (18).
> ***
> ## Requested Changes
> ### C1. Adding a table
> >Add a table with the results that are derived in this paper in some special cases, and how they align with prior work. For example, for NAG, the paper recovers acceleration of prior work, but the Heavy-ball method is not covered by the framework at all. Such a table would help to give a quick overview of the generality and limitations of the proposed theoretical framework.
>
> We thank the reviewer for this constructive suggestion. We have added two tables in Sections 2.2 and 3.2 before presenting main results for strongly convex and convex settings respectively. These tables provide a quick overview of how prior work on accelerations is covered by our analyses.
>
> ### C2. Moving numerical results to the main paper
> >I suggest moving some numerical results to the main part or at least providing links to them to increase the clarity of the work.
>
> We appreciate the reviewer’s constructive suggestion. We have moved the numerical results to Sections 2.2 and 3.2 in the main paper, following the main results for strongly convex and convex cases respectively.
> ### C3.Defining gradient correction before first use
> >In the introduction, the authors already use the term "gradient correction" without introducing it. In my opinion, such a term is not standard in the literature, and it should be defined.
>
> In the revised paper, we have defined the term "gradient correction" in the Introduction section where it is first mentioned. Specifically, on page 2 we write
> > In particular, the high-resolution ODE for NAG-SC includes an additional Hessian term, which is absent in the ODE for HB. The Hessian term stems from the gradient correction term in the NAG-SC algorithm, which is defined as linear in the difference between the current and previous gradients.
>
> ### C4. Wording adjustment
> >On page 3, the authors use the term "meaningful conditions". I suggest avoiding such wording without giving a precise description. The same applies to the wording "It seems to be an open" question.
>
> We have modified the related text to be more precise in the revised paper.
> ### C5. Linearity of $\alpha_k$
> >Can the authors provide other examples in the convex case, where $\alpha_k$ is not necessarily linear? According to Proposition 2, we should have $\alpha_k=\Omega(k)$. Can we choose something else?
>
> Theorem 4 and Proposition 2 assume a recursive condition $\alpha_{k+1}(\alpha_{k+1}-1)\leq \alpha^2_k$ on $\alpha_k$, which implies that $\alpha_{k+1}\leq (1+\sqrt{1+4\alpha_k^2})/{2}\leq \alpha_k+1$, and hence $\alpha_k=\Theta(k)$. This is explained in our Footnote 8 on page 12.

---

> ### Author Response · Authors · 2025-12-30
> **Response to Reviewer es1t (Part 3/3)**
>
> ### C6. Discussion on the non-accelerations of HB
> >Can the authors provide a discussion on the results in [2]? Does the current theoretical framework cover the results in [2]? I believe that the authors should include the discussion around [2] in the paper. [2] Goujaud, Baptiste, Adrien Taylor, and Aymeric Dieuleveut. "Provable non-accelerations of the heavy-ball method: B. Goujaud et al." Mathematical Programming (2025): 1-59.
>
> We thank the reviewer for pointing out this important work. We agree that it is highly relevant. We have revised the paper to reflect the latest results on HB in Section 2.1 on page 6. Please also see our response later to your comment about the coverage of HB by our analysis. In more details below, we discuss the paper and its relation with our work.
>
> - Regarding HB, previously it was only known that HB with the configuration $s=4/(\sqrt{\mu}+\sqrt{L})^2$ and $\sigma=(1-\sqrt{\mu s})^2$ cycles and fails to converge for some $f\in\mathcal{S}^1_{\mu, L}$ (Lessard et al., 2016), and that HB with $\sigma=(1-\sqrt{\mu s})/(1+\sqrt{\mu s})$ achieves a non-accelerated rate $O((1-O(1/\kappa))^k)$ with $s\asymp \mu/L^2$ (Shi et al. 2022). Goujaud et al. (2025) proves that the worst-case convergence rate of HB on $\mathcal{S}^1_{\mu, L}$ is no better than a non-accelerated rate, for any combinations of $s$ and $\sigma$. Goujaud et al. (2025) further shows that adding more Lipschitz-type regularity conditions on $f$ (e.g., restricting $f$ with Lipschitz continuous Hessians) still does not help HB to achieve acceleration.
> - The results of Goujaud et al. (2025) are in alignment with our work. Since HB fails to converge at an accelerated rate for any $s$, $\sigma$, and Lipschitzness in higher-order derivatives of $f$, the key that NAG-SC achieves acceleration must be due to the gradient correction term, which is the **only** difference between HB and NAG-SC.
> - Goujaud et al. (2025) obtained the non-acceleration results of HB by analyzing the cycling patterns of HB, which can be extended to more general first-order algorithms in a systematic and constructive manner. The techniques can be used to find counterexamples and potentailly help establish lower bounds and necessary conditions for acceleration.
>
> ### References mentioned:
> - Lessard, L., Recht, B., & Packard, A. (2016). Analysis and design of optimization algorithms via integral 	quadratic constraints. _SIAM Journal on Optimization_, _26_(1), 57-95.
> - Goujaud, B., Taylor, A., & Dieuleveut, A. (2025). Provable non-accelerations of the heavy-ball method. _Mathematical Programming_, 1-59.
> ### C7. Notation $\eta_0$
> >I suggest not using terms $\eta_0$, etc, in Theorem 1, as they are later used in Theorem 2 with a different meaning. I would simply say that $\eta$ is constant
>
> The terms $\eta_0$, $\nu_0$, and $\tau_0$ in Theorem 1 are consistent with those in Theorems 2 and 3.  They can be understood as the leading constants in the Taylor expansions of $\eta$, $\nu$, and $\tau$ as a function of $\sqrt{q}$, as stated in Assumption 1. In this way, Theorem 1 is notationally consistent with Theorems 2 and 3, and can be understood as a special case where all terms  with order greater than or equal to $1$ vanish in the Taylor expansion of $\eta$, $\nu$, and $\tau$. This notation is convenient when we explain the relationship between Theorems 1, 2, and 3 in Section 2.2.
> ### C8. On the coverage of HB
> >Is there any reason why the theoretical framework does not cover the original Heavy-ball method? Can the authors elaborate more on this point?
>
> In Section 2, we mainly study the three-variable form (10), which is formulated for the technical convenience in constructing Lyapunov functions. However, it turns out that the three-variable form (10) has inherent structural constraints on the coefficients in its equivalent single-variable form (11), which makes it fail to include HB as a special case. Such limitation also results in the technical condition $c_1^2>4c_0$ in our Corollary 1, which is basically a translation of the results in Theorem 2 for (11) to the re-parametrized (14). Nonetheless, HB can be viewed as a limiting case of (14) with $c_0=1$, $c_1\downarrow 2$, and $c_2=1/2$, which, by Corollary 1 part (i), has a non-accelerated convergence rate. These points are now clarified after Corollary 1 in the revised paper.
> ### C9. Definition of strong convexity and smoothness
> >I suggest adding the definitions of (strong) convexity and smoothness at least in the appendix and referring to them in the theorem statements.
>
> We have described necessary notation, including the definitions of (strong) convexity and smoothness, before the highlighted Propositions 1 and 2 in the Introduction section.

---

> > ### Comment · Reviewer_es1t · 2026-02-17
> > **Response to the authors**
> >
> > Dear Authors,
> >
> > I thank you for the detailed care you have taken of my concerns. You addressed all the raised concerns carefully. I am satisfied with the way the changes are incorporated into the revision. This work presents an interesting framework for analyzing acceleration in optimization. I hope that in future work, sufficient conditions for acceleration will be further studied.

---

### Author Response · Authors · 2025-12-30
**Revised paper**

We thank all the reviewers for their thoughtful and constructive comments and suggestions that greatly help to improve the paper. Below, we summarize the major revisions made to the paper. All changes are highlighted in blue in the revised paper.
- On literature review and comparison
	- We add literature review on the LMI/SDP techniques (with multiple references) & the SDE line of work (Compagnoni et al., 2025a; b) in the Introduction section.
	- We incorporate the latest results on HB by Goujaud et al. (2025) in Section 2.1 on page 6.
	- We add a new section (Appendix A) for comparing our work with the LMI/SDP techniques, Sanz Serna & Zygalakis (2021),  and Shi et al. (2022).
- On readability and notation issues
	- In the Introduction section, we provide numerical illustrations (see Figure 1) for the effect of choosing the coefficient of gradient correction on convergence with other algorithmic parameters fixed.
	- For further empirical validation, we move the numerical parts from the appendix to the main paper (see Figures 2, 3, and 4), following the main theoretical results in Sections 2.2 and 3.2.
	- We add two tables at the beginning of Sections 2.2 and 3.2 to summarize the coverage of existing algorithms by our analyses. The coverage of NAG and HB as limiting cases of the algorithm (14) is now clarified after Corollary 1.
	- We now describe necessary notation before presenting the two featured results, Propositions 1 and 2, in the Introduction section.
	- We add a notation table as new Appendix B to describe the notation used in the whole paper in a logical order.
- On well-posedness of ODEs
	- In the revised paper, for all ODEs appearing in Section 4 (including those in text and formal statements), we explicitly state the smoothness condition and discuss the existence and uniqueness of the solution, except for the high-resolution ODE (31) for which we assume a solution exists in Proposition 5.

We welcome and look forward to any further feedback from the reviewers and Action Editor regarding the revised manuscript.

### References mentioned:
- Compagnoni, E. M., Islamov, R., Proske, F. N., & Lucchi, A. (2025a). Unbiased and Sign Compression in Distributed Learning: Comparing Noise Resilience via SDEs. In _The 28th International Conference on Artificial Intelligence and Statistics_.
- Compagnoni, E. M., Liu, T., Islamov, R., Proske, F. N., Orvieto, A., & Lucchi, A. (2025b). Adaptive Methods through the Lens of SDEs: Theoretical Insights on the Role of Noise. In _The Thirteenth International Conference on Learning Representations_.
- Goujaud, B., Taylor, A., & Dieuleveut, A. (2025). Provable non-accelerations of the heavy-ball method. _Mathematical Programming_, 1-59.
- Lessard, L., Recht, B., & Packard, A. (2016). Analysis and design of optimization algorithms via integral 	quadratic constraints. _SIAM Journal on Optimization_, _26_(1), 57-95.
- Sanz Serna, J. M., & Zygalakis, K. C. (2021). The connections between Lyapunov functions for some optimization algorithms and differential equations. _SIAM Journal on Numerical Analysis_, _59_(3), 1542-1565.
- Shi, B., Du, S. S., Jordan, M. I., & Su, W. J. (2022). Understanding the acceleration phenomenon via high-resolution differential equations. _Mathematical Programming_, _195_(1), 79-148.

---

### Decision · Action_Editor_4p3M · 2026-03-09

**Recommendation:** Accept as is

**Audience:**

Yes

**Audience Explanation:**

Yes: a noticeable part of TMLR's audience works on optimization and closely-related topics. Since acceleration is a fundamental topic in optimization, these readers will be most likely interested in learning about the findings of this work closer.

**Claims And Evidence:**

Yes

**Claims Explanation:**

As all reviewers pointed out, all claims are supported by complete mathematical proofs. The original submission required an improvement of the presentation of the results (as noted by som reviewers) and the authors incorporated the required changes in the final version.